# Hierarchical Reinforcement Learning with Targeted Causal Interventions

Sadegh Khorasani [1]  Saber Salehkaleybar [2]  Negar Kiyavash [3]  Matthias Grossglauser [1]

## Abstract

Hierarchical reinforcement learning (HRL) improves the efficiency of long-horizon reinforcement-learning tasks with sparse rewards by decomposing the task into a hierarchy of subgoals. The main challenge of HRL is efficient discovery of the hierarchical structure among subgoals and utilizing this structure to achieve the final goal. We address this challenge by modeling the subgoal structure as a causal graph and propose a causal discovery algorithm to learn it. Additionally, rather than intervening on the subgoals at random during exploration, we harness the discovered causal model to prioritize subgoal interventions based on their importance in attaining the final goal. These targeted interventions result in a significantly more efficient policy in terms of the training cost. Unlike previous work on causal HRL, which lacked theoretical analysis, we provide a formal analysis of the problem. Specifically, for tree structures and, for a variant of Erdős-Rényi random graphs, our approach results in remarkable improvements. Our experimental results on HRL tasks also illustrate that our proposed framework outperforms existing work in terms of training cost.

## 1. Introduction

In traditional reinforcement learning (RL), an agent is typically required to solve a specific task based on immediate feedback from the environment (Sutton, 2018). However, in many real-world applications, the agent faces tasks where rewards are sparse and delayed. This poses a significant challenge to traditional RL since the agent must take many actions without receiving an immediate reward. For instance, in maze-solving tasks, the agent might only receive a reward once it finds a path to a specific exit or a location within the maze. Hierarchical reinforcement learning (HRL) allows the agent to break down the problem into subtasks, which is helpful in environments where achieving goals requires a long horizon (Bacon et al., 2017; Barto & Mahadevan, 2003; Eysenbach et al., 2019; Le et al., 2018).

Hierarchical policies are widely used in HRL to manage the complexity of long-horizon and sparse-reward tasks. However, learning hierarchical policies introduces significant challenges. To address these issues, many methods are proposed to improve learning efficiency (Kulkarni et al., 2016; Levy et al., 2017; Vezhnevets et al., 2017). The high-level policy does not deal with primitive actions but instead sets subgoals or options (Sutton et al., 1999) that the lower-level policies must achieve. Lower-level policies are trained to achieve subgoals assigned by the high-level policy. In order to discover meaningful subgoals and their relationships, it is essential to ensure that both high-level and low-level policies explore the environment efficiently. However, naive exploration of the entire goal space is not sample efficient in high-dimensional goal spaces. Some studies in the literature aim to improve sample efficiency and scalability by employing off-policy techniques (Nachum et al., 2018b), utilizing representation learning (Nachum et al., 2018a), or restricting the high-level policy to explore only a subset of the goal space (Zhang et al., 2020).

In order to improve exploration efficiency, recently, Hu et al. (2022); Nguyen et al. (2024) proposed using an off-the-shelf causal discovery algorithm (Ke et al., 2019) to infer the causal structure among subgoals. This learned causal structure is subsequently used to select a subset of subgoals, from which one is chosen *uniformly at random* for exploration. In Nguyen et al. (2024), the agent initially takes random actions until it serendipitously achieves the final goal a few times–a scenario that is highly unlikely in long-horizon tasks with a huge state space. After that, the agent infers the causal structure among state-action pairs which is more challenging than recovering the causal structure among subgoals when the state space is very large. It

[1]School of Computer and Communication Sciences, EPFL, Lausanne, Switzerland [2]Leiden Institute of Advanced Computer Science (LIACS), Leiden University, Leiden, The Netherlands [3]College of Management of Technology, EPFL, Lausanne, Switzerland. Correspondence to: Sadegh Khorasani <sadegh.khorasani@epfl.ch>, Saber Salehkaleybar <s.salehkaleybar@liacs.leidenuniv.nl>, Negar Kiyavash <negar.kiyavash@epfl.ch>, Matthias Grossglauser <matthias.grossglauser@epfl.ch>.

*Proceedings of the $42^{nd}$ International Conference on Machine Learning*, Vancouver, Canada. PMLR 267, 2025. Copyright 2025 by the author(s).

is noteworthy that the causal discovery algorithm used in both works was applied without any adaptation to the HRL setting, and no theoretical guarantee was provided regarding the quality of its output. Additionally, they did not include theoretical analysis for their proposed framework.

In this paper, we introduce the Hierarchical Reinforcement Learning via Causality (HRC) framework where achieving a subgoal by a multi-level policy is formulated as an intervention in the environment. The HRC framework enables us to discover the causal relationships among subgoals (so-called subgoal structure) through targeted interventions. The agent is fully guided by the recovered subgoal structure in the sense that the acquired causal knowledge is utilized in both designing targeted interventions and also in training the multi-level policy. Our main contributions are as follows:

- We introduce a general framework called HRC (Section 4) that enables the agent to exploit the subgoal structure of the problem to achieve the final goal more efficiently. In particular, we design three causally-guided rules (sections 6 and F.9) to prioritize the subgoals in order to explore those with higher causal impact on achieving the final goal (see Section 6).
- Although subgoal discovery in the HRC framework could be carried out using off-the-shelf causal discovery algorithms, we propose a new causal discovery algorithm (Section 8) specifically tailored to subgoal discovery in the HRL setting, and we provide theoretical guarantees for it.
- Our HRC framework allows us to formulate the cost of training for any choice of exploration strategy (Section 5). In particular, we theoretically bound the training cost for our proposed causally-guided rules, which are derived harnessing the subgoal structure, compared to an agent that performs random subgoal exploration (Section 7).
- The experimental results in Section 9 demonstrate the clear advantage of the proposed approach compared to the state of the art both in synthetic datasets and in a real-world scenario (the Minecraft environment).

## 2. Related Work

Hierarchical policies enable multilevel decision-making (see Appendix A for related work). Moreover, causal relationships can be utilized to improve exploration, generalization, and explanation (see Appendix A). In the following, we review the related work at the intersection of HRL and causality.

**Causal hierarchical reinforcement learning**: In Corcoll & Vicente (2020), a notion of subgoal is defined based on the causal effect of an action on the changes in the state of the environment. A high-level policy is trained to select from the defined subgoals. However, in the exploration phase, the subgoals are chosen uniformly at random without exploiting the causal relationships among subgoals. Recent work Hu et al. (2022); Nguyen et al. (2024) used the causal discovery algorithm in Ke et al. (2019) to infer causal structures among subgoals. In Hu et al. (2022), this learned causal structure is subsequently used to select a subset of subgoals, from which one is chosen uniformly at random (without any prioritization among the selected subgoals) for exploration. As a result, it made limited use of the causal structure during exploration. In Nguyen et al. (2024), the agent initially takes random actions until it accidently achieves the final subgoal a few times (which is highly improbable in long-horizon tasks with a large state space). Following this, the agent infers the causal structure among state-action pairs, which is more challenging than inferring the causal structure among subgoals when the state space is large. Moreover, both work directly applied the causal discovery algorithm in Ke et al. (2019) without providing any theoretical guarantee on its performance.

## 3. Problem Statement and Notations

In this section, we review the concepts and define the notations necessary to formulate our proposed work on HRL. In many real-world applications, an agent must perform a sequence of actions before receiving any reward signal from the environment. We focus on this specific setting, where the agent must achieve intermediate objectives before receiving a reward. Figure 1 shows a simple example where a craftsman must obtain wood and stone to build a pickaxe. The craftsman receives a reward only if he builds a pickaxe; otherwise, he receives no reward. That is, while obtaining both wood and stone are intermediate subgoals that the craftsman needs to plan for, it does not result in a reward. The action space (denoted by $\mathcal{A}$) for this game includes moving "right", "left", "up", and "down", " pick" and "craft", which enables the craftsman to interact with the environment.

In our setup, the agent perceives the environment in the form of "disentangled factors" (Hu et al., 2022). For instance, in a visual observation of a robotic environment, the disentangled factors may include elements such as object quantity, object position, and velocity. We define Environment Variables (EVs) as the disentangled factors of the environment observations. We assume that the agent has access to all EVs. We denote the set of EVs by $\mathcal{X}_{EV} = \{X_1, \cdots, X_l\}$. The vector of these EVs

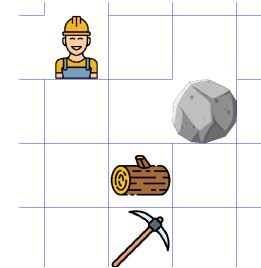

*Figure 1.* Craftsman in the mini-craft game.

at time step $t$, denoted by $\mathbf{X}_{EV}^t = (X_1^t, \cdots, X_l^t)$, is the *state of the system*. In the craftsman game of Figure 1, the craftsman observes his position on the map and the quantity of each object (wood or stone) in his backpack. Therefore, in this game, $\mathcal{X}_{EV} = \{X_1, X_2, X_3, X_4\} = \{$"number of wood", " number of stone", " number of pickaxe", "craftsman position"$\}$.

We define a subset $\mathcal{X} \subseteq \mathcal{X}_{EV}$, where each variable $X_i \in \mathcal{X}$ represents a resource environment variable. Resource environment variables refer to items, tools, or abilities that the agent attains throughout the game and are key factors in achieving the final objective. Without loss of generality, we assume that the first $n$ EVs are resource EVs, hence $\mathcal{X} = \{X_1, \cdots, X_n\}$, where $n \leq l$. We denote the vector of $\mathcal{X}$ at time step $t$ by $\mathbf{X}^t = (X_1^t, \cdots, X_n^t)$. In our example, the objects wood, stone, and pickaxe are resource EVs. Therefore, $\mathcal{X} = \{X_1, X_2, X_3\}$.

Furthermore, we define a set of subgoals $\Phi = \{g_1, \cdots, g_n\}$, associated with a set of values $e = \{e_1, \cdots, e_n\}$, where each $g_i$ corresponds to resource EV $X_i$. We say subgoal $g_i$ is achieved at time $t$ if and only if $X_i^t = e_i$. Without loss of generality, we consider $g_n$ as the only desired main goal that the agent is supposed to achieve and refer to it as the "final subgoal" or "final goal". In our example, for instance, the subgoals $g_1$, $g_2$, and $g_3$ with the corresponding values $e_1 = 1$, $e_2 = 1$, and $e_3 = 1$ correspond to obtaining one wood, one stone, and one pickaxe, respectively [1].

In many real-world applications, resource variables (such as a skill, tool, or access to certain objects) only need to be attained once. For instance, once a skill is attained by an agent in an environment, it remains permanently available. Motivated by this observation but mainly for ease of presentation, we suppose that the domain of each resource EV is $\{0, 1\}$, and a subgoal $X_i$ is achieved when $e_i = 1$ [2].

We model the HRL problem of our interest through a subgoal-based Markov decision process. The subgoal-based MDP is formalized as a tuple $(\mathcal{S}, \Phi, \mathcal{A}, T, R, \gamma)$, where $\mathcal{S}$ is the set of all possible states (in our setup, all possible values of $\mathbf{X}_{EV}^t$), $\Phi$ is the subgoal space, which contains intermediary objectives that guide the learning of the policy, $\mathcal{A}$ is the set of actions available to the agent, $T : \mathcal{S} \times \mathcal{A} \times \mathcal{S} \to [0, 1]$ is the transition probability function that denotes the probability of the transition from state $s$ to state $s'$ after taking an action $a$, and $R : \mathcal{S} \times \mathcal{A} \times \Phi \to \mathbb{R}$ is the reward indicator function of the agent pertaining to whether a subgoal $g$ was achieved in state $s$ after taking action $a$. More specifically, $R(s, a, g) = 1$, if the subgoal $g$ is achieved, otherwise it is zero. $\gamma$ is the discount factor that quantifies the diminishing value of future rewards. Given a state $s \in \mathcal{S}$ and a subgoal $g \in \Phi$, the objective in a subgoal-based MDP is to learn a subgoal-based policy $\pi(a|s, g) : \mathcal{S} \times \Phi \to \mathcal{A}$ that maximizes value function defined as $V^\pi(s, g) = \mathbb{E}_\pi \left[ \sum_{t=0}^\infty \gamma^t R(s_t, a_t, g) \mid s_0 = s \right]$.

We use the term *system probes* to refer to the observed state-action pairs. For a given time horizon $H$, the agent observes a sequence of state-action pairs $\tau = (s_0, a_0, \cdots, s_{H-1}, a_{H-1})$, called a trajectory. The action variable at time step $t$ is denoted by $A^t \in \mathcal{A}$. We assume that the process $\{A^t, \mathbf{X}_{EV}^t\}_{t \in \mathbb{Z}^+}$ can be described by a structural causal model (SCM) (see the definition of SCM C.1) in the following form:

$$\begin{cases} X_i^{t+1} = f_i(\text{pa}(X_i^{t+1}), A^t, \epsilon_i^{t+1}), & \forall t \geq 1 \text{ and } 1 \leq i \leq l, \\ A^{t+1} = f_0(\mathbf{X}_{EV}^{t+1}, \epsilon_0^{t+1}), & \forall t \geq 0, \end{cases}$$
(1)

where $\text{pa}(X_i^{t+1})$ represents the parent of $X_i^{t+1}$ in $\mathbf{X}_{EV}^t$, and $f_i$ is the causal mechanism showing how $\text{pa}(X_i^{t+1})$ influences $X_i$ at time $t + 1$. Furthermore, $\epsilon_i^{t+1}$ and $\epsilon_0^{t+1}$ represent the corresponding exogenous noises of $X_i$ and $A$ at time $t + 1$, respectively.

The summary graph $\mathsf{G}$ is used to graphically represent the causal relationships in the SCM where there is a node for each variable in $\mathcal{X}_{EV} \cup \{A\}$. Furthermore, an edge from $X_j$ to $X_i$ is drawn in $\mathsf{G}$ if $X_j^t \in \text{pa}(X_i^{t+1})$ for any $t$. Additionally, there exist directed edges from the action variable $A$ to each environment variable $X_i$ and from $X_i$ back to $A$.

**Definition 3.1** (Subgoal Structure). The subgoal structure, denoted as $\mathscr{G}$, is a directed graph (not necessarily acyclic), where the nodes represent subgoals in the set $\Phi$. In $\mathscr{G}$, a directed edge from subgoal $g_j$ to subgoal $g_i$ exists if and only if there is at least one path in the summary graph $\mathsf{G}$ from $X_j$ to $X_i$ such that all intermediate nodes along this path belong to the set $(\mathcal{X}_{EV} \setminus \mathcal{X})$. These edges in $\mathscr{G}$ represent a one-step causal relationship from one subgoal to another by ignoring intermediate environmental variables not designated as subgoals.

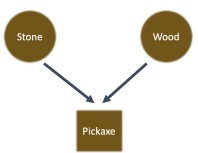

*Figure 2.* Subgoal structure $\mathscr{G}$ of the mini-craft game.

---

[1] Our framework assumes discretized resource EVs, which is a reasonable assumption in domains where acquiring specific resources is required. For environments with continuous variables, one can apply disentangled representation learning methods to high-dimensional continuous observations in order to extract categorical latent variables—such as using categorical VAEs. We view this as a natural extension of our current work.

[2] Our proposed solutions can be extended to the settings with non-binary domains for EVs at the cost of more cumbersome notation. The assumption of binary variables for resource EVs is made solely to facilitate the cost analysis in Section 7 and to provide theoretical guarantees for our causal discovery method in Section 8. Indeed, in our experiments, we evaluate our proposed methods in the non-binary settings.

The subgoal structure of the above mini-craft game is depicted in Figure 2. We use the terms 'subgoal' and 'node' interchangeably. In $\mathscr{G}$, the parent and children sets of a subgoal $g_i$ are denoted by $PA_{g_i}$ and $CH_{g_i}$, respectively.

In our setup, the subgoals are further categorized into two types, based on their corresponding causal mechanisms. $g_i$ is an "AND" subgoal if it can be achieved at time step $t$ if and only if all its parents in the set $PA_{g_i}$ have been achieved prior to time step $t$, indicating a strict conjunctive requirement for the achievement of $g_i$. $g_i$ is an "OR" subgoal if it can be achieved at time step $t$ if at least one of its parents in the set $PA_{g_i}$ has been achieved before time step $t$. This indicates a disjunctive requirement for the activation of $g_i$, where the achievement of any single parent is sufficient to achieve $g_i$. In the rest of the paper, we represent an AND subgoal with a square and an OR subgoal with a circle in the figures.

We denote the estimate of $\mathscr{G}$ by $\hat{\mathscr{G}}$. Additionally, the parent and children sets of a subgoal $g_i$ in the graph $\hat{\mathscr{G}}$ are denoted by $PA_{g_i}^{\hat{\mathscr{G}}}$ and $CH_{g_i}^{\hat{\mathscr{G}}}$, respectively. For a given set of subgoals $L$, $CH_L^{\hat{\mathscr{G}}} = \bigcup_{g_i \in L} CH_{g_i}^{\hat{\mathscr{G}}}$.

**Definition 3.2** (Hierarchical Structure). The hierarchical structure is denoted by $\mathcal{H} = (\mathscr{H}, \mathcal{L})$, where $\mathscr{H}$ is a directed acyclic graph (DAG) which is a subgraph of $\hat{\mathscr{G}}$, and $\mathcal{L} : \Phi \to \mathbb{N}$ is a level assignment function, where $\mathcal{L}(g_i)$ denotes the hierarchy level of subgoal $g_i$. In $\mathcal{H}$, each node is assigned to a hierarchy level such that for each subgoal $g_i$, $\mathcal{L}(g_i) \geq \max_{g_j \in PA_{g_i}^{\hat{\mathscr{G}}}} \mathcal{L}(g_j) + 1$ (See an example in C.1).

**Definition 3.3** (Intervention, Explorational Data). An intervention on a subgoal $g_i$ at time $t^*$ is considered as replacing the structural assignment (1) of the corresponding environment variable $X_i$ with $X_i^t = \alpha$ for $t \geq t^*$, where $\alpha \in \{0, 1\}$. This intervention is denoted by $\mathrm{do}(X_i^{t^*} = \alpha)$. Moreover, the explorational data of the subgoal $g_i$ is shown by $D_{\mathrm{do}(X_i^{t^*}=1)}$ and consists of the state-action pairs that the agent collects (the actions are taken randomly) from the time step $t^*$ until $\min(H, t^* + \Delta)$, where $\Delta$ is a positive integer and $H$ is the horizon.

Furthermore, for a given set of subgoals $B$ and any time step $t$, $X_B^t = a$ implies that $X_i^t = a$ for all $g_i \in B$.

**Definition 3.4.** Let $A$ and $B$ be two subsets of subgoals $\Phi$, such that $g_n \notin (A \cup B)$. The expected causal effect ($ECE$) of $A$ on the final goal $g_n$ at some time step $t^* + \Delta$ (where $\Delta$ is a positive integer) conditioned on $\mathrm{do}(X_B^{t^*} = 0)$ is defined as:

$$ECE_{t^*}^{\Delta}(A, B, g_n) = \mathbb{E}[X_n^{t^*+\Delta} \mid \mathrm{do}(X_A^{t^*} = 1), \mathrm{do}(X_B^{t^*} = 0)]$$
$$- \mathbb{E}[X_n^{t^*+\Delta} \mid \mathrm{do}(X_A^{t^*} = 0), \mathrm{do}(X_B^{t^*} = 0)].$$

# 4. Hierarchical Reinforcement Learning via Causality (HRC)

In this section, we introduce Hierarchical Reinforcement Learning via Causality (HRC) framework which guides the learning policy by incorporating information pertaining to hierarchical subgoal structure (see Definition 3.2). In order to describe the HRC framework, we first define the notion of "controllable subgoal."

**Definition 4.1.** For a given $\epsilon > 0$, a subgoal $g_i$ is $(1 - \epsilon)$-**controllable** by the subgoal-based policy $\pi(a \mid s, g_i)$, if there exists a time step $t^* < H$ such that $g_i$ is achieved with probability at least $1 - \epsilon$ (i.e., $\mathbb{P}(X_i^{t^*} = 1) \geq 1 - \epsilon$), when actions are taken according to $\pi$.

**Assumption 4.2.** We assume that if a subgoal $g_i$ is achieved at a time step $t^*$ (i.e., $X_i^{t^*} = 1$), it will remain achieved for all future time steps. Formally, if $X_i^{t^*} = 1$, then $X_i^t = 1$ for all $t \geq t^*$.

*Remark* 4.3. Suppose a subgoal $g_i$ is $(1 - \epsilon)$-controllable. In this case, based on Definition 4.1 and Assumption 4.2, $X_i^t = 1$ for all $t \geq t^*$. According to Definition 3.3, this is equivalent to an intervention $\mathrm{do}(X_i^{t^*} = 1)$ with probability at least $1 - \epsilon$, where the intervention can be performed by taking actions based on the subgoal-based policy $\pi(a \mid s, g_i)$. For the sake of brevity, from hereon we say a subgoal is controllable and drop $(1 - \epsilon)$.

Before introducing the algorithm, we give a high-level overview using the example 1, explained in Section 3. The craftsman has no knowledge of the subgoal structure (i.e., Figure 2) but he knows which objects exist on the map (call them resources). Initially, he considers obtaining each resource as a subgoal and divides subgoals into two sets: controllable (CS) and intervention (IS) as shown in the figures with blue and green, respectively. Controllable Subgoals set are those that he has learnt how to achieve (see Definition 4.1). Initially, both sets are empty (Figure 3(a)). The craftsman plays the game several times and learns how to gather "stone" and "wood". Thus, he can add "obtaining stone" and "obtaining wood" (call them root subgoals) to the controllable set, meaning he can from now on acquire them whenever needed (Figure 3(b)). At this stage, he can still be unsure how to craft a pickaxe. Now, the craftsman starts a loop and at each iteration of this loop, he chooses one of the items in the controllable set and adds it to the intervention set to explore whether it is a requirement for obtaining other resources (i.e., subgoals). Let us assume he chooses "obtaining stone" (denote it as $g_{\mathrm{sel}}$) and adds it to the intervention set (Figure 3(c)). Then he plays the game multiple times, and observes all the trajectories (so-called interventional data) with the goal of discovering all reachable subgoals, which are the direct children of the subgoals in the intervention set. This is done using a causal discovery subroutine in HRC algorithm (will be discussed

in Algorithm 1). In the example in Figure 1, "obtaining pickaxe" is an AND subgoal (see Figure 2), therefore it is highly likely that the causal discovery algorithm would not detect it as the child of "obtaining stone" alone. However, in the next iteration of the loop, when the craftsman also adds the subgoal "obtaining wood" to the intervention set (Figure 3(d)), he will discover that the subgoal "obtaining pickaxe" is the child of "obtaining stone" and "obtaining wood". He can now play the game multiple times to hone in on how to efficiently obtain a pickaxe (Figure 3(e)). Note that the main phases of the algorithm are learning the subgoal structure and then exploiting it to train the policy. To learn the subgoal structure, the algorithm can benefit from purposeful "interventions" on the resources instead of naive random play (see Section 6 for details).

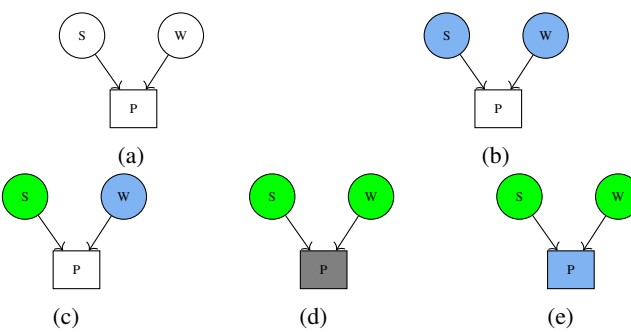

(a)            (b)

(c)            (d)            (e)

*Figure 3.* Discovering the subgoal structure in the mini-craft (Figure 1). S, W, and P represent stone, wood, and pickaxe. The sets IS, and CS, are illustrated with green, and blue respectively.

### 4.1. HRC Framework

The pseudocode of HRC is given in Algorithm 1. HRC algorithm at each iteration, chooses a subgoal from the controllable set (line 4) and adds it to the intervention set (line 5), then it collects interventional data (line 6) and uses causal discovery (line 7) to identify the reachable subgoals (line 8) from the current intervention set. These reachable subgoals are added to the hierarchy, assigned levels (line 9), and trained (line 10). Successfully trained subgoals are added to the controllable set (line 11), and the process repeats until the final subgoal is added to the intervention set or no further controllable subgoals remain. In particular, HRC is comprised of the following key steps:

1. **Initialization:** We initialize intervention ($IS_0$) and controllable sets ($CS_0$) with empty sets. The hierarchical structure, $\mathcal{H} = (\mathscr{H}, \mathcal{L})$, initially contains a graph $\mathscr{H}$ with the subgoals as nodes, but with no edges between them, and all the entries in $\mathcal{L}$ are set to 0.
2. **Pre-train:** In line 2, we call SubgoalTraining on all the subgoals ($\Phi$) to train the ones with no parent in $\mathscr{G}$. These subgoals are called root subgoals (see Appendix D.4 for more details about the SubgoalTraining sub-

routine). Controllable set ($CS_0$), is populated with root subgoals using SubgoalTraining subroutine.
3. **Expand intervention set:** In this step, a subgoal from $CS_{t-1}$ is selected and added to the current intervention set $IS_{t-1}$ to form the new intervention set $IS_t$, and then removed from the controllable set (lines 4-5).
4. **Intervention Sampling:** InterventionSampling subroutine collects $T$ trajectories. In each trajectory, it randomly intervenes on the subgoals in $IS_t$ until the horizon $H$ is reached or all the subgoals in $IS_t$ are achieved. We call the entire data collected in this subroutine "interventional data" and denote it as $D_I$ (see Appendix D.1 for more details).
5. **Causal Discovery:** After collecting interventional data ($D_I$), the algorithm estimates the causal model and $\hat{\mathscr{G}}_t$ using a causal discovery method. We propose our causal discovery method in Section 8.
6. **Candidate Controllabe Set:** From $\hat{\mathscr{G}}_t$, we identify the set of subgoals that are not in $IS_t \cup CS_t$, but are children of subgoals in $IS_t$, and all their parents are in $IS_t$. We call this set the candidate controllable set $CCS_t$ and its elements reachable subgoals.
7. **Update-$\mathcal{H}$:** To update the graph $\mathscr{H}$, for each subgoal $g_i \in CCS_t$, we add an edge from each parent $g_j \in PA_{g_i}^{\hat{\mathscr{G}}_t}$ to $g_i$. Additionally, the level of $g_i$ is updated to:

$$\mathcal{L}(g_i) = \begin{cases} 0, & \text{if } |PA_{g_i}^{\hat{\mathscr{G}}_t}| = 0, \\ \max_{g_j \in PA_{g_i}^{\hat{\mathscr{G}}_t}} \mathcal{L}(g_j) + 1, & \text{otherwise.} \end{cases}$$

8. **Subgoal Training:** SubgoalTraining subroutine trains the subgoal-based policy for achieving the subgoals in $CCS_t$ [3] (see Appendix D.4 for more details).
9. **Update Controllable Set:** This step adds the trained subgoals to the controllable set $CS_t$.

Steps 3 to 6, aim of which is to find the reachable subgoals, form **Stage 1**, and steps 7 to 9, which train the reachable subgoals to make them controllable, form **Stage 2**. HRC can use different strategies to select a controllable subgoal from $CS_{t-1}$ in line 4 . If HRC randomly selects a controllable subgoal from $CS_{t-1}$, we call it "random strategy".

Figure 4 illustrates a simple example of how strategically selecting controllable subgoals can enhance the performance of HRC. Figure 4 shows different stages of the underlying subgoal structure. In this example, we assume that every subgoal is of OR type. The final subgoal is node $F$. Reachable subgoals are depicted in gray. Assume we have reached the stage where $S$ and $W$ are controllable (Figure 4(a)).

---

[3]If a subgoal becomes reachable, it becomes controllable after Subgoal Training step. However, in practice, this may not always be the case. Therefore, we consider a threshold and remove the ones that are not trained well both from the set $CCS_t$ and hierarchical structure $\mathcal{H}$ (see Appendix D.4 for more details).

---

**Algorithm 1** Hierarchical Reinforcement Learning via Causality (HRC)

1: Initialize subgoal-based policy $\pi$; intervention set $\text{IS}_0 = \{\}$; controllable set $\text{CS}_0 = \{\}$; hierarchical structure $\mathcal{H} = (\mathscr{H}, \mathcal{L})$; iteration counter $t = 1$.
2: $\text{CS}_0 \leftarrow \text{SubgoalTraining}(\pi, \mathcal{H}, \text{IS}_0, \Phi)$ {Train the root subgoals}
3: **while** $g_n \notin \text{IS}_{t-1}$ and $\text{CS}_{t-1} \neq \emptyset$ **do**
4:     $g_{\text{sel, t}} \leftarrow$ Choose a controllable subgoal from $\text{CS}_{t-1}$   ⎱ Expand the intervention set by adding
5:     $\text{IS}_t \leftarrow \text{IS}_{t-1} \cup \{g_{\text{sel, t}}\}, \text{CS}_t \leftarrow \text{CS}_{t-1} \setminus \{g_{\text{sel, t}}\}$   ⎰ a potentially required subgoal
6:     $D_I \leftarrow \text{InterventionSampling}(\pi, \text{IS}_t)$
7:     $\hat{\mathscr{G}}_t \leftarrow \text{CausalDiscovery}(D_I, \text{IS}_t)$   Find reachable subgoals
8:     $\text{CCS}_t \leftarrow \{g_i \mid g_i \notin (\text{IS}_t \cup \text{CS}_t) \text{ and } g_i \in CH_{\text{IS}_t}^{\hat{\mathscr{G}}_t} \text{ and } PA_{g_i}^{\hat{\mathscr{G}}_t} \subset \text{IS}_t\}$

    **Stage 1** (line 4 to 8)

9:     $\text{Update-}\mathcal{H}(\hat{\mathscr{G}}_t, \text{IS}_t, \text{CCS}_t)$   ⎱ Train reachable subgoals
10:    $\text{CCS}_t \leftarrow \text{SubgoalTraining}(\pi, \mathcal{H}, \text{IS}_t, \text{CCS}_t)$   ⎰ and add them to the controllable set
11:    $\text{CS}_t \leftarrow \text{CS}_t \cup \text{CCS}_t$
12:    $t \leftarrow t + 1$

    **Stage 2** (line 9 to 13)

13: **end while**

---

HRC will terminate earlier if it selects subgoal $S$ as $g_{\text{sel, 1}}$ (Figure 4(c)), rather than $W$ (Figure 4(b)). Therefore, strategic selection of controllable variables from CS in line 4 of Algorithm 1 can significantly improve the performance. In Section 6, we propose causally-guided selection mechanisms to pick $g_{\text{sel, t}}$ in order to improve the performance. Before introducing these mechanisms, we formulate the cost of HRC algorithm in the next section.

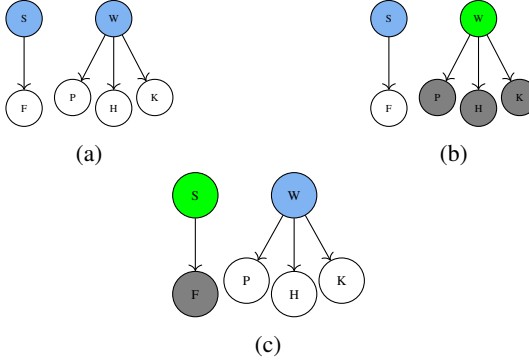

(a)             (b)

(c)

*Figure 4.* A showcase of HRC performing more efficiently under a better selection of controllable subgoal. The sets $\text{IS}, \text{CS}$, and CCS are illustrated with green, blue, and gray colors, respectively.

## 5. Formulating the Cost

In order to measure the effectiveness of any strategy used for selecting $g_{\text{sel, t}}$ in line 4 of Algorithm 1, we need to formalize the training cost of the HRC algorithm, which is the expected system probes during Intervention Sampling and Subgoal Training accumulated in all iterations. To compute the expected total cost of Algorithm 1, we model its execution as an MDP, denoted as $\mathcal{M}$. The state space of $\mathcal{M}$ is the power set of $\Phi$. A state in this MDP, denoted by $\mathcal{I}$, represents the intervention set IS in Algorithm 1. The

action space is $\Phi$, and the action is determined by $g_{\text{sel}}$. If we are in state $\mathcal{I}$ and choose $g_{\text{sel}}$, then we transit to state $\mathcal{I} \cup \{g_{\text{sel}}\}$ with a probability determined by the strategy in line 4 of Algorithm 1. Let $C_{g_n}(\mathcal{I})$ be the expected cost of adding the final subgoal $g_n$ to the intervention set when the current state is $\mathcal{I}$. Then, we have:

$$C_{g_n}(\mathcal{I}) = \sum_{g_{\text{sel}} \in \Phi} p_{\mathcal{I}, \mathcal{I} \cup \{g_{\text{sel}}\}} \Big[ C_{trans}(\mathcal{I}, \mathcal{I} \cup \{g_{\text{sel}}\})$$
$$+ C_{g_n}(\mathcal{I} \cup \{g_{\text{sel}}\}) \Big], \qquad (2)$$

where the transition cost $C_{trans}(\mathcal{I}, \mathcal{I} \cup \{g_{\text{sel}}\})$ is the cost of transition from the current intervention set $\mathcal{I}$ to a new set by adding $g_{\text{sel}}$ to it ($\mathcal{I} \cup \{g_{\text{sel}}\}$). Moreover, the cost-to-go, $C_{g_n}(\mathcal{I} \cup \{g_{\text{sel}}\})$, accounts for the future costs from the new state onwards. Both terms are weighted by the transition probability $p_{\mathcal{I}, \mathcal{I} \cup \{g_{\text{sel}}\}}$, the probability of transition from state $\mathcal{I}$ to $\mathcal{I} \cup \{g_{\text{sel}}\}$ in $\mathcal{M}$. This transition probability is determined by the strategy in line 4 of Algorithm 1. Based on the above definitions, the total expected cost is equal to $C_{g_n}(\{\})$. See the detailed derivation of $C_{trans}(\mathcal{I}, \mathcal{I} \cup \{g_{\text{sel}}\})$ and $C_{g_n}(\{\})$ in Appendix E.1.

## 6. Subgoal Discovery with Targeted Strategy

In this section, we propose two causally-guided ranking rules (a third rule, which is a combination of these two rules is discussed in Appendix F.9) that utilize $\hat{\mathscr{G}}$ and the estimated causal model to prioritize among controllable subgoals candidates for picking $g_{\text{sel, t}}$ at iteration $t$ in line 4 of Algorithm 1.

**Causal Effect Ranking Rule** An intuitive rule is to estimate the expected causal effect of every $g_i \in \text{CS}_{t-1}$ on the final subgoal $g_n$ and select the subgoal with the largest

effect. Specifically,

$$g_{\text{sel, t}} = \arg\max_{g_i \in \text{CS}_{t-1}} \left( \widehat{ECE}_{t^*}^{\Delta}(\{g_i\}, \{\}, g_n) \right),$$

where $\widehat{ECE}_{t^*}^{\Delta}$ is the estimate of the expected causal effect defined in 3.4 (See Appendix H.3 for details on how to estimate the expected causal effect from the recovered causal model and also a discussion on the values of $\Delta$ and $t^*$). When all subgoals are of AND type, under the conditions explained in Appendix F.1, this rule yields the minimum training cost as only subgoals with a path to final subgoal are added to IS.

**Shortest Path Ranking Rule**  In this section, we utilize $A^*$-based search approach in designing a shortest path ranking rule. $A^*$ algorithm finds a weighted shortest path in a graph by using a heuristic function $h(g_i)$ to estimate the cost from a node $g_i$ to the final subgoal (Hart et al., 1968). Additionally, $A^*$ uses a cost function $g(g_i)$ that determines the actual cost from the start or the root node to $g_i$. At each iteration, it selects the node with the lowest combined cost $f(g_i) = g(g_i) + h(g_i)$ until it reaches the final subgoal (see Appendix F.2 for details of $A^*$ algorithm).

Inspired by $A^*$ search, at each iteration of Algorithm 1, we choose $g_{\text{sel, t}}$ as the subgoal with the lowest combined cost $f(g_i)$, that is, $g_{\text{sel, t}} = \arg\min_{g_i \in \text{CS}_{t-1}} f(g_i)$.

In Appendix F.7, we explain how to calculate functions $g$ and $h$ in our setting. When the subgoal structure is a DAG and every subgoal is of the OR type [4], this rule adds precisely the subgoals that are on the shortest path to the final subgoal to IS and hence incurs a minimum training cost. In Appendix F.9, we also propose a **Hybrid ranking rule** that is a combination of the two ranking rules presented here.

## 7. Cost Analysis

In this section, we analyze the cost of Algorithm 1 for a random strategy, denoted by $\text{HRC}_\text{b}$ (baseline), and targeted strategy, denoted by $\text{HRC}_\text{h}$, which uses any of the ranking rules in Section 6. We consider two subgoal structures: a tree graph $G(n, b)$, where $b$ is the branching factor; and a semi-Erdős–Rényi graph $G(n, p)$, with $p = \frac{c \log(n)}{n-1}$ and $0 < c < 1$ (see definition in Appendix E.3).

Recall that a subgoal becomes reachable once it is added to the candidate controllable set CCS, which is defined based on $\mathscr{G}$ (see line 8 of Algorithm 1). If there is an estimation error in the causal discovery part, it is possible that a subgoal $g_i$ is not added to CCS despite all its parents being in the intervention set IS for an AND subgoal (resp. at least one of

*Table 1.* Comparison of the cost between $\text{HRC}_\text{h}$ and $\text{HRC}_\text{b}$ in tree and semi-Erdős–Rényi graphs.

| | **TREE** $G(n, b)$ | **SEMI-ERDŐS–RÉNYI** $G(n, p)$ |
|---|---|---|
| $\text{HRC}_\text{h}$ | $O(\log^2(n)b)$ | $O(n^{\frac{4}{3} + \frac{2}{3}c} \log(n))$ |
| $\text{HRC}_\text{b}$ | $\Omega(n^2 b)$ | $\Omega(n^2)$ |

its parents has been added to IS in the case of OR subgoal). For the ease of analysis, we exclude this error event by making the next assumption and add two more assumptions.

**Assumption 7.1.**  In the Causal Discovery part, we assume that for every $g \in \text{IS}$, and $g_i \in CH_g$, $g_i$ is also in $CH_g^{\hat{\mathscr{G}}}$ iff $g_i$ is an OR subgoal or all $PA_{g_i} \in \text{IS}$. Moreover, we assume if a subgoal becomes reachable in line 8, it will become controllable after Subgoal Training step in line 10.

**Assumption 7.2.**  We assume that the estimated causal effect of a subgoal $g_i$ on the final subgoal $g_n$ is not zero if $g_i$ is an ancestor of $g_n$ in $\mathscr{G}$.

**Assumption 7.3.**  We assume that the expected system probes between achieving any two consecutive subgoals in $\text{IS}_t$ are equal (see the exact statement in E.1).

**Theorem 7.4.**  *Let $G(n, b)$ be a tree structure with branching factor $b > 1$, and $G(n, p)$ a semi-Erdős–Rényi graph where $p = \frac{c \log(n)}{n-1}$, $0 < c < 1$ and $n > 4$. Under Assumptions 7.1, 7.2, and 7.3, the cost complexities of the $\text{HRC}_b$ and $\text{HRC}_h$ are given in Table 1.*[5]

Thus, the worst-case cost of targeted strategy $\text{HRC}_\text{h}$, is significantly lower than the lower bound on the cost of random strategy $\text{HRC}_\text{b}$. In fact, in semi-Erdős–Rényi graphs, the gap widens for sparser subgoal structures.

## 8. Causal Subgoal Structure Discovery (SSD)

In this section, we focus on the Causal Discovery part of Algorithm 1 by proposing a new causal discovery algorithm that improves the efficiency of learning the subgoal structure. First, we need the following definitions below:

**Definition 8.1** (One-sided valid assignment).  Let $\mathbf{X} \in \{0, 1\}^n$. We say $\mathbf{X}$ is a *one-sided valid assignment* if it satisfies the following conditions for every subgoal indexed by $i$: (1) For an OR subgoal $g_i$, $(X_i = 1) \implies \exists g_j \in PA_{g_i}, X_j = 1$; (2) For an AND subgoal $g_i$, $(X_i = 1) \implies \forall g_j \in PA_{g_i}, X_j = 1$.

In general, the algorithm cannot discover all parents of a goal, but only the so-called discoverable parents as defined below.

**Definition 8.2** (Discoverable Parent).  For any subgoal $g_i$, a parent $g_j \in PA_{g_i}$ is called *discoverable* if the following

---

[4]There are some further mild conditions given in Appendix F.1

[5]All proofs are provided in the Appendix.

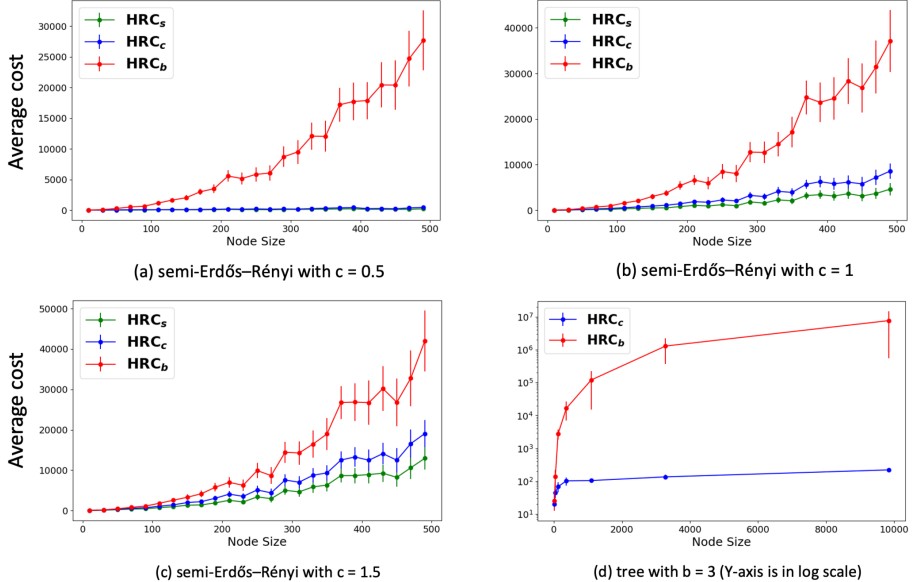

*Figure 5.* Training cost in semi-Erdős–Rényi and tree graphs.

holds: (1) For an OR subgoal, there exist one-sided valid assignments $\mathbf{X}, \mathbf{X}' \in \{0,1\}^n$ such that $X_j = 1$ and $X'_j = 0$, and $\forall g_k \in PA_{g_i} \setminus g_j, X_k = X'_k = 0$, implying $X_i$ changes from 0 to 1 due to $X_j$; (2) For an AND subgoal, there exist one-sided valid assignments $\mathbf{X}, \mathbf{X}' \in \{0,1\}^n$ such that $X_j = 1$ and $X'_j = 0$, and $\forall g_k \in PA_{g_i} \setminus g_j, X_k = X'_k = 1$, indicating $X_i$ becomes 1 only when $X_j = 1$ and all other parents are equal to one.

**Proposition 8.3.** *Under Assumption 4.2, the subgoal structure is identifiable up to discoverable parents.*

In Appendix G.2, we give an example where all the parents of a subgoal cannot be recovered from the interventional data collected in Algorithm 1.

To recover the relationship between subgoals, we model the changes in the EVs corresponding to subgoals by an abstracted structural causal model (A-SCM) between the variables in $\mathcal{X}$. This abstraction enables us to focus on the relationships between subgoals while skipping intermediate environment variables ($\mathcal{X}_{EV} \setminus \mathcal{X}$), operating on a different time scale compared to the original SCM. In A-SCM, the value of $X_i^{t+1}$ is determined as follows:

$$X_i^{t+1} = \theta_i(\mathbf{X}^t) \oplus \epsilon_i^{t+1}, \qquad 1 \le i \le n, \qquad (3)$$

where

$$\theta_i(\mathbf{X}^t) = \begin{cases} \bigwedge_{g_j \in PA_{g_i}} X_j^t & \text{if } g_i \text{ is AND subgoal,} \\ \bigvee_{g_j \in PA_{g_i}} X_j^t & \text{if } g_i \text{ is OR subgoal,} \end{cases}$$

and $\oplus$ denotes the XOR operation. Moreover, the error term $\epsilon_i^{t+1}$ is Bernoulli with parameter $\rho < 1/2$.

**Theorem 8.4.** *Given the A-SCM defined in* (3)*, for a vector $\boldsymbol{\beta} \in \mathbb{R}^n$, consider*

$$S(\mathbf{X}^t, \boldsymbol{\beta}) = \sum_j \beta_j X_j^t + \beta_0.$$

*Let $\hat{X}_i^{t+1} = \mathbb{1}\{S(\mathbf{X}^t, \boldsymbol{\beta}) > 0\}$ [6] be an estimate of $X_i^{t+1}$. Define the loss function:*

$$\mathcal{L}(\boldsymbol{\beta}) = \mathbb{E}[(\hat{X}_i^{t+1} - X_i^{t+1})^2] + \lambda \|\boldsymbol{\beta}\|_0. \qquad (4)$$

*There exists a $\lambda > 0$ such that for any optimal solution $\boldsymbol{\beta}^*$ minimizing the loss function in* (4)*, the positive coefficients in $\boldsymbol{\beta}^*$ correspond to the parents of $X_i$.*

*Remark 8.5.* In our experiments, for the causal discovery algorithm, we use the $l_1$ norm instead of the $l_0$ norm in the above loss function.

## 9. Experimental Results

In this section, we compare our proposed methods with previous work [7]. First, we compare our proposed ranking rules with a random strategy using synthetic data. Herein, $HRC_c$ and $HRC_s$ represent the versions of HRC algorithm using the causal effect ranking rule and the shortest path ranking rule, respectively. When HRC algorithm uses the random strategy, it is denoted as a baseline or $HRC_b$.

The cost depicted in Figures 5(a), 5(b), 5(c), and 5(d) represents the training cost of the HRC algorithm, as formulated

---

[6] $\mathbb{1}\{A\} = 1$ if $A$ is true, and $\mathbb{1}\{A\} = 0$ otherwise.

[7] The code for all experiments is available at https://github.com/sadegh16/HRC.

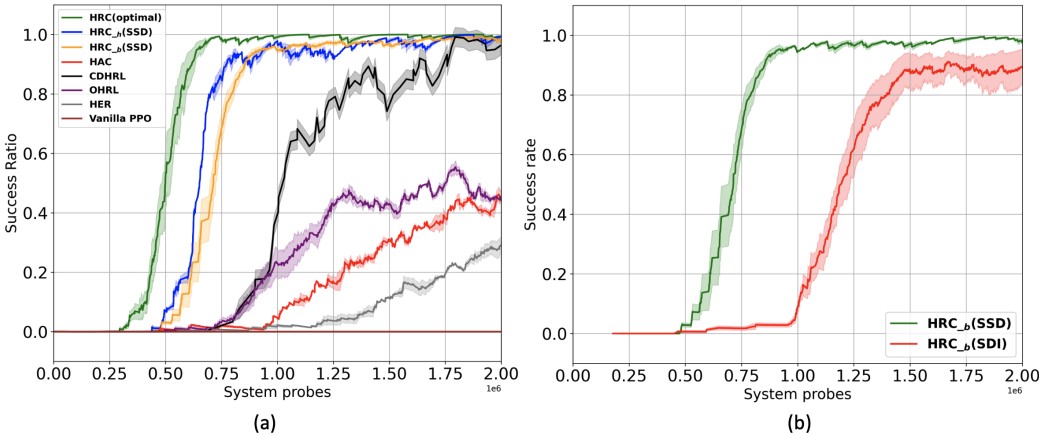

*Figure 6.* Comparison of versions of HRC algorithm (ours) with other methods in a complex environment of 2D-Minecraft.

in (11), as a function of the number of nodes. Specifically, Figures 5(a), 5(b), and 5(c) illustrate the costs for semi-Erdős–Rényi graph $G(n, p)$ for different values of $c$ (see Section 7 for the definition of $c$). Figure 5(d) illustrates the cost of $HRC_c$ for a tree $G(n, b)$ with the branching factor 3. In tree structures, there is no difference between using the shortest path versus the causal effect rule, as there is only one path to the final subgoal. In all the cases, the proposed ranking rules outperform the baseline, especially in sparse graphs (See Appendix H for more details).

To evaluate the effectiveness of our proposed methods in a realistic setting, we conducted experiments using a complex long-horizon game called **2D-Minecraft** (Sohn et al., 2018). This environment is a simplified two-dimensional adaptation of the well-known game Minecraft (Guss et al., 2019), where the agent must accomplish various hierarchical subgoals (such as collecting resources and crafting items) to reach the final objective. Note that in this game the agent receives a positive reward only upon achieving the final subgoal [8]. We compared our algorithm with several HRL algorithms, namely CDHRL, HAC, HER, OHRL, and Vanilla PPO. See Appendix H for more details about these algorithms.

Figure 6(a) compares three versions of HRC algorithm with previous work. In Figure 6(a), $HRC_h$ (SSD) denotes the version of HRC algorithm that uses the causal effect ranking rule as the targeted strategy, and the proposed causal discovery method (SSD). $HRC_b$ (SSD) represents the version of HRC algorithm which uses the same causal discovery method (SSD) but with a random strategy instead. HRC(optimal) denotes the HRC algorithm where both the causal discovery algorithm and the ranking rules operate without error, representing an upper bound on the performance that cannot be surpassed. The y-axis represents the

[8]For implementation details, see Section H in the Appendix.

*Table 2.* Comparison of SHD for causal discovery methods

| METHOD | SHD | MISSING | EXTRA |
|---|---|---|---|
| SSD (OURS) | 12.3 | 6.0 | 6.3 |
| SDI (KE ET AL., 2019) | 19.8 | 4.2 | 15.6 |

success ratio, indicating the percentage of trajectories in which the final subgoal was achieved, and the x-axis shows system probes. As can be seen, all versions of our HRC algorithm learn faster than the previous methods. This indicates that both our causal discovery method and targeted strategies provide remarkable improvements in achieving the final subgoal (see sensitivity analysis in Appendix H.4).

In Figure 6(b) we compare the success ratio of two versions of the HRC algorithm which are only different in the causal discovery part. In both versions, we use random strategy. In particular, $HRC_b$ (SSD) uses our proposed causal discovery method while $HRC_b$ (SDI) uses the one in Ke et al. (2019), which was used in Hu et al. (2022). This figure shows that our proposed causal discovery method is more effective in having faster learning by providing more accurate causal knowledge.

We compare the recovered causal graphs in both methods and show their accuracy in Table 2, measured in terms of SHD (Tsamardinos et al., 2006). As can be seen in Table 2, our causal discovery algorithm results in much fewer extra edges, while having a few more missing edges on average.

## 10. Conclusion

We studied the problem of discovering and utilizing hierarchical structures among subgoals in HRL. By utilizing our causal discovery algorithm, the framework enables targeted interventions that prioritize impactful subgoals for achieving the final goal. Experimental results validate its superiority over existing methods in training cost.

## Acknowledgements

This research was supported by the Swiss National Science Foundation through the NCCR Automation program under grant agreement no. 51NF40_180545.

## Impact Statement

This paper presents work whose goal is to advance the field of Machine Learning. There are many potential societal consequences of our work, none which we feel must be specifically highlighted here.

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

# A. Additional Related Work

**Temporal abstractions and hierarchical policy:** In many applications, such as strategy games Vinyals et al. (2019) or robot locomotion Stone et al. (2005), decision-making involves operating across different time scales. Temporal abstraction in RL addresses this by considering abstract actions that represent sequences of primitive actions. One way to implement temporal abstraction is through options, where each option defines a policy that the agent follows until a specific termination condition is met. The option framework allows agents to choose between executing a primitive action or an option. Further studies have been made to generalize the standard reward function or the value function for this framework (Schaul et al., 2015; Sorg & Singh, 2010). Upon the concept of temporal abstractions, researchers have developed several frameworks to learn hierarchical policies (Dietterich, 2000; Kulkarni et al., 2016; Levy et al., 2017; McGovern & Barto, 2001; Nachum et al., 2018b). A recent work by (Kulkarni et al., 2016) proposed a framework that takes decisions in two levels of hierarchy: a higher-level policy that picks a goal, and a lower-level policy that selects primitive actions to achieve that goal. Despite these advancements, training hierarchical policies remains challenging, as changes in a policy at one level of the hierarchy can alter the transition and reward functions at higher levels. This interdependence makes it difficult to learn policies at different levels jointly. Previous work has attempted to solve this issue but they mostly lack generality or require expensive on-policy training (Florensa et al., 2017; Heess et al., 2016; Sigaud & Stulp, 2019; Vezhnevets et al., 2017). To address these challenges, techniques such as Hierarchical Reinforcement Learning with Off-Policy Correction (HIRO) (Nachum et al., 2018b) and Hierarchical Actor-Critic (HAC) (Levy et al., 2017) have been proposed. For instance, HAC accelerates learning by enabling agents to learn multiple levels of policies simultaneously.

**Causal reinforcement learning:** Recently, many studies have focused on exploiting causal relationships in the environment to address some of the main challenges in RL (Ding et al., 2022; Florensa et al., 2018; Forestier et al., 2022; Li et al., 2024; Méndez-Molina, 2024; Méndez-Molina et al., 2020; Pitis et al., 2020; Seitzer et al., 2021; Wang et al., 2022; Yang et al., 2024; Yu et al., 2023; Zhu et al., 2022). For instance, Seitzer et al. (2021) proposes a mutual information measure to detect the causal influence of an action on the subsequent state, given the current state. This measure can be leveraged to enhance exploration and training. Many model-based RL methods learn a world model that generalizes poorly to unseen states. Wang et al. (2022) aims to address this challenge by learning a causal world model that removes unnecessary dependencies between state variables and the action. Ding et al. (2022) leverage causal reasoning to address the challenge of goal-conditioned generalization, where the goal for the training and testing stages will be sampled from different distributions. Recent work Yu et al. (2023) utilized causal discovery to capture the influence of actions on environmental variables and explain the agent's decisions. Zhu et al. (2022) shows theoretically that an agent equipped with a causal world model outperforms the one with a conventional world model in offline RL.

# B. Connections to Related Literature

**Skill Discovery:** The goal of skill discovery (Wang et al., 2024) is to learn a diverse set of skills, which are later used to train a higher-level policy for a downstream task. In that context, a skill is conceptually similar to a subgoal in our work. However, there are key methodological differences in subgoal vs. skill discovery: 1- In skill discovery, the process of learning the skill set is often decoupled from the downstream task. In contrast, our work—framed within the context of HRL—conditions the learning process on a target goal. By leveraging the learned subgoal structure, our approach guides exploration toward only the relevant subgoals that contribute to achieving the target goal. In contrast, skill discovery methods often aim to learn as many diverse skills as possible, regardless of their relevance to the target goal in the downstream task. 2- In our framework, the lower-level policy is itself hierarchical and is trained according to defined hierarchical structure (where we defined it formally based on the discovered subgoal structure). This results in a significant reduction in sample complexity compared to standard policy training, which typically does not utilize such a structure. Beyond these methodological distinctions in subgoal discovery and training policies, to the best of our knowledge, our work is the first to rigorously study HRL within a causal framework. We formally defined the cost formulation, proposed subgoal discovery strategies (with our key Definition ECE 3.4) with performance guarantees (Theorem 7.4), and provided theoretical bounds on the extent to which the subgoal structure can be learned in an HRL setting (Prop. 8.3). Furthermore, we introduce a causal discovery algorithm tailored to this setting, with provable guarantees on its correctness (Theorem 8.4).

**Inductive Logic Programming (ILP):** RL has also been studied in other areas. Inductive Logic Programming (ILP)-based methods (Jiang & Luo, 2019; Kimura et al., 2021) represent policies as logical rules to enhance interpretability. Our work considers a multi-level policy, with a focus on recovering the hierarchy among subgoals (with our explicit hierarchical structure definition) for training the multi-level policy more efficiently. Herein, we do not impose any structural limitations

on our policy. For our theoretical analysis, we assumed that the causal mechanism of each subgoal follows an AND/OR structure. This assumption pertains to the environment rather than for representing the policy using logical rules.

**Reward Machines:** Reward machine (Icarte et al., 2022) can be viewed as an alternative way to represent hierarchical structures in HRL and can be effectively used for training policies.

## C. Additional Preliminaries and Definitions

We model the HRL problem of our interest through a subgoal-based Markov decision process. The subgoal-based MDP is formalized as a tuple $(\mathcal{S}, \Phi, \mathcal{A}, T, R, \gamma)$, where $\mathcal{S}$ is the set of all possible states (in our setup, all possible values of $\mathbf{X}_{EV}^t$), $\Phi$ is the subgoal space, which contains intermediary objectives that guide the learning of the policy, $\mathcal{A}$ is the set of actions available to the agent, $T : \mathcal{S} \times \mathcal{A} \times \mathcal{S} \to [0, 1]$ is the transition probability function that denotes the probability of the transition from state $s$ to state $s'$ after taking an action $a$, and $R : \mathcal{S} \times \mathcal{A} \times \Phi \to \mathbb{R}$ is the reward indicator function of the agent pertaining to whether a subgoal $g$ was achieved in state $s$ after taking action $a$. More specifically, $R(s, a, g) = 1$, if the subgoal $g$ is achieved, otherwise it is zero. $\gamma$ is the discount factor that quantifies the diminishing value of future rewards. Given a state $s \in \mathcal{S}$ and a subgoal $g \in \Phi$, the objective in a subgoal-based MDP is to learn a subgoal-based policy $\pi(a|s, g) : \mathcal{S} \times \Phi \to \mathcal{A}$ that maximizes value function defined as $V^\pi(s, g) = \mathbb{E}_\pi \left[ \sum_{t=0}^\infty \gamma^t R(s_t, a_t, g) \mid s_0 = s \right]$.

### C.1. Introductory Example

For the example provided in Section 3, based on Definition 3.2, we present two hierarchical structures of the subgoal structure in Figure 2 in the following.

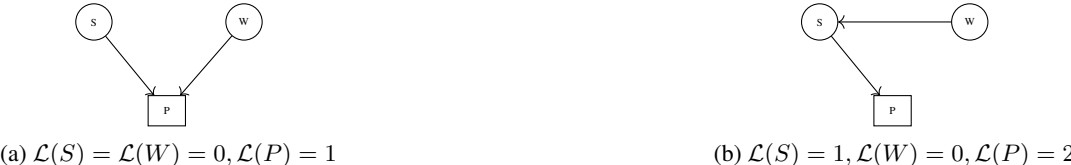

(a) $\mathcal{L}(S) = \mathcal{L}(W) = 0, \mathcal{L}(P) = 1$         (b) $\mathcal{L}(S) = 1, \mathcal{L}(W) = 0, \mathcal{L}(P) = 2$

*Figure 7.* Two hierarchical structures of the subgoal structure given in Figure 2. S, W, and P denote stone, wood, and pickaxe, respectively.

In Figure 7, S, W, and P denote stone, wood, and pickaxe, respectively. Figure 7a represents a hierarchical structure $\mathcal{H} = (\mathscr{H}, \mathcal{L})$ where $\mathscr{H}$ graph is the same as the real subgoal structure $\mathscr{G}$ and level of S and W are zero and the level of P is 1. Figure 7b shows another $\mathcal{H} = (\mathscr{H}, \mathcal{L})$ where $\mathscr{H}$ is not an exact estimate of $\mathscr{G}$. The level of W is zero, S is 1 and P is 2.

### C.2. More Definitions

**Definition C.1** (Structural Causal Model (SCM)(Peters et al., 2017)). A Structural Causal Model is defined as a tuple $M = \langle U, V, F, P(U) \rangle$, where:

- $U$ is a set of exogenous variables whose values are determined by factors outside the model.
- $V$ is a set of endogenous variables whose values are determined by variables within the model.
- $F$ is a set of structural functions $\{f_v\}$, one for each $v \in V$, such that $v = f_v(pa(v), u_v)$, where $pa(v) \subseteq V \setminus \{v\}$ is the set of parents of $v$ and $u_v$ is the exogenous variable affecting $v$.
- $P(U)$ is a probability distribution over the exogenous variables $U$ which are required to be jointly independent.

## D. HRC Algorithm

### D.1. Intevention Sampling

Algorithm 2 collects interventional data $D_I$ for the subgoals in IS in $T$ trajectories. During each trajectory, it randomly intervenes on the subgoals in IS until the horizon $H$ is reached or all the subgoals in IS are achieved. In line 1, it initializes an empty dataset to store the interventional data. Then at each iteration in the "For" loop, it records a trajectory $\tau_j$. The "While" loop in line 4 iterates until the horizon $H$ is reached or all the subgoals in IS are achieved. In line 5, it randomly selects not already achieved subgoal $g_i$ from the set IS and in line 6 it performs intervention on it. This intervention can be accomplished by taking actions based on the subgoal-based policy $\pi(a \mid s, g_i)$ until the subgoal $g_i$ is achieved at some

time $t^*$ (see Remark 4.3) or a predefined $max\_steps$ (the maximum number of steps that the policy interacts with the environment) is reached. Since all the subgoals in set IS are already controllable, with a high probability, the subgoal $g_i$ will be achieved. Next, in line 7, we append all the observed state-action pairs to the $\tau_j$ (along with explorational data if $g_i$ is achieved).

---

**Algorithm 2** Intervention Sampling

---

**Require:** Subgoal-based policy $\pi$, set of subgoals IS, number of trajectories $T$, maximum horizon $H$
**Ensure:** Collected interventional data $D_I$
 1: $D_I \leftarrow \{\}$ {Initialize empty interventional dataset}
 2: **for** $j = 1$ to $T$ **do**
 3:     Initialize empty trajectory $\tau_j = \{\}$
 4:     **while** (len($\tau_j$) < $H$) or (there exists an unachieved subgoal in IS) **do**
 5:         In IS, randomly select a subgoal that is not already achieved, denoted as $g_i$.
 6:         Perform intervention on $g_i$ {This can be accomplished by taking actions based on the subgoal-based policy $\pi(a \mid s, g_i)$ until the subgoal $g_i$ is achieved at some time $t^*$ 4.3}
 7:         Add the observed state-action pairs and explorational data ($D_{\mathrm{do}(X_i^{t^*}=1)}$) (if any) to $\tau_j$
 8:     **end while**
 9:     Append trajectory $\tau_j$ to $D_I$: $D_I \leftarrow D_I \cup \tau_j$
10: **end for**
11: **return** $D_I$

---

## D.2. How to use a hierarchical structure and policy?

The hierarchical structure $\mathcal{H}$ determines which subgoals need to be achieved before achieving a specific subgoal $g_i$. These prerequisite subgoals are the ancestors of $g_i$, and they must be achieved by following a valid causal order. This causal order can be obtained using Algorithm 3, which identifies the ancestors of $g_i$ and then sorts them by their level in the hierarchical structure. However, multiple valid causal orderings may exist, and it is the policy's role to choose a proper order in a given state. This policy can be modeled as a multi-level policy, which is defined in the following subsection.

---

**Algorithm 3** Find Causal Order of Ancestors

---

**Require:** Hierarchical structure $\mathcal{H} = (\mathscr{H}, \mathcal{L})$, node $g_i$
**Ensure:** List of ancestors of $g_i$ sorted by causal order
 1: Initialize empty list $A \leftarrow []$ {List of ancestors}
 2: Initialize stack $S \leftarrow [g_i]$ {DFS stack to traverse ancestors}
 3: **while** $S$ is not empty **do**
 4:     $g_j \leftarrow$ pop element from stack $S$
 5:     **if** $g_j$ is not already in $A$ **then**
 6:         Add $g_j$ to $A$
 7:         Push all parents of $g_j$ from $\mathscr{H}$ into $S$
 8:     **end if**
 9: **end while**
10: Sort $A$ by increasing $\mathcal{L}(g_j)$ for all $g_j \in A$
11: **return** $A$ {List of ancestors in causal order}

---

## D.3. Multi-level Policy

A multi-level policy $\pi_h$ is a collection of $k$ policies: $\pi_h = \{\pi_0, \pi_1, \ldots, \pi_{k-1}\}$, where $\pi_i$ is the policy at level $i$. The lowest-level policy is $\pi_0(a|s, g) : \mathcal{S} \times \Phi \rightarrow \mathcal{A}$ which returns a primitive action given the current state $s$ and some subgoal $g$. Moreover, a policy $\pi_i(g'|s, g) : \mathcal{S} \times \Phi \rightarrow \Phi$ at a higher level ($1 \leq i \leq k$), returns a subgoal $g'$ given the current state $s$ and conditioned subgoal. Please note that the level of a policy is different than the level of a subgoal in the hierarchical structure (Refer to Definition 3.2 for the definition of levels of subgoals in a hierarchical structure).

Algorithm 4 gives a pseudo-code of executing the multi-levelpolicy recursively for a given state $s \in \mathcal{S}$ and subgoal $g \in \Phi$.

The input to the multi-level policy is a tuple of state, conditioned subgoal, and the recovered hierarchical structure $\mathcal{H}$. Initially, in line 2, the algorithm determines the level of the subgoal (denoted by $i$) in the hierarchical structure, $\mathcal{L}(g)$. In line 3, it identifies the parents of this subgoal in the recovered subgoal structure and stores them in the $action\_mask$. The output of policy is forced to be taken from the subgoals in $action\_mask$ in line 5. The maximum number of actions that the policy at each level is allowed to take is equal to $max\_actions$. If the subgoal is a root in the subgoal structure (i.e., $i > 0$), the $action$ of $\pi_i$ is a subgoal, and the algorithm recursively calls the EXECUTE_$\pi_h$ function for this subgoal (line 7). Otherwise, the subgoal is a root, and the agent takes the primitive actions directly in the environment (line 9). After taking each action, the algorithm checks whether the goal has been achieved or if the trajectory is terminated. If either condition is met, the loop is terminated, and the algorithm returns the new state to the upper level of the multi-level policy.

---

**Algorithm 4** Executing Hierarchical Policy

---

**Require:** state $s$, subgoal $g$, hierarchical structure $\mathcal{H}$
 1: **Function** EXECUTE_$\pi_h(s, g, \mathcal{H})$:
 2:    $i \leftarrow$ level of the $g$ in the hierarchical structure
 3:    $action\_mask \leftarrow$ parents of the $g$ from hierarchical structure
 4:    **for** each step in $max\_actions$ **do**
 5:      Select action based on the $action\_mask$: $a \leftarrow \pi_i(a \mid s, g)$
 6:      **if** $i > 0$ **then**
 7:        $next\_state, done, goal\_achieved \leftarrow$ EXECUTE_$\pi_h(s, a, \mathcal{H})$
 8:      **else**
 9:        Execute action in environment:
          $next\_state, reward, done \leftarrow env.step(a)$
10:      **end if**
11:      goal\_achieved $\leftarrow$ Check if $goal$ is achieved
12:      $s \leftarrow next\_state$
13:      **if** $done$ **or** $g$ is achieved **then**
14:        **break**
15:      **end if**
16:    **end for**
17:    **return** $next\_state, done, goal\_achieved$

---

### D.4. Subgoal Training

In Algorithm 5, in line 1, we first determine the maximum level of the reachable subgoals from the set CCS. If the level number of these subgoals exceeds the current number of levels, $k$, in the multi-level policy $\pi_h$, we add a new level to $\pi_h$. For each reachable subgoal $g_i$ in the set CCS, we collect $T'$ trajectories. In each trajectory $\tau$, we iteratively generate a random number between 0 and 1 and randomly pick a subgoal in the set IS if the chosen number is less than $p$. Otherwise, we select $g_i$ for intervention. This ensures that the policy learns how to achieve $g_i$ not only from the initial state but also after having some interventions. In practice, we set $p = 0.1$. Note that if the final subgoal belongs to the reachable set CCS, we only train on the final subgoal $g_n$. Additionally, for each subgoal $g$, we define a parameter called the *success ratio*, which measures the proportion of times the subgoal $g$ is successfully achieved after interventions on it during Algorithm 5. We consider the subgoals that have a success ratio greater than a threshold $\phi_{\text{causal}}$ as controllable.

## E. Subgoal Discovery

### E.1. Cost formulation proof

We start by formulating the cost at each iteration $t$:

$$\underbrace{\sum_{j=1}^{T} \Big( \sum_{g' \in \text{IS}_t} w^{\tau_j}_{\text{int}, g'} + w^{\tau_j}_{\text{exp}} \Big)}_{\text{InterventionSampling}} + \underbrace{\sum_{g'' \in \text{CCS}_t} \sum_{j=1}^{T'} \Big( \sum_{g' \in \text{IS}_t} w^{\tau_j}_{\text{int}, g'} + w^{\tau_j}_{\text{train}, g''} \Big)}_{\text{SubgoalTraining}}, \tag{5}$$

---

**Algorithm 5** Subgoal Training with Probabilistic Subgoal Selection

---

**Require:** subgoal-based policy $\pi_h$ with $k$ levels, hierarchical structure $\mathcal{H}$, intervention set IS, reachable subgoals in CCS, verification threshold $\phi_{\text{causal}}$

  1: **if** final subgoal $g_n$ belongs to the reachable set CCS **then**
  2:     Train only on the final subgoal $g_n$ : CCS $\leftarrow \{g_n\}$
  3: **end if**
  4: Add a new level to $\pi_h$ if $\max_{g_i \in \text{CCS}} \mathcal{L}(g_i) > k$
  5: **for** $g_i$ in CCS **do**
  6:     **for** $j = 1$ to $T'$ **do**
  7:         Initialize empty trajectory $\tau_j = \{\}$
  8:         **while** $\text{len}(\tau_j) < H$ or subgoal $g_i$ is not achieved **do**
  9:             Select $r$ randomly from $[0, 1]$
10:             **if** $r \leq p$ **then**
11:                 In IS, randomly select a not already achieved subgoal (denoted as $g_j$)
12:             **else**
13:                 Set $g_j = g_i$
14:             **end if**
15:             Call EXECUTE_$\pi_h(state, g_j, \mathcal{H})$
16:             Add the observed state-action pairs to $\tau_j$
17:         **end while**
18:         Append trajectory $\tau_j$ to $D_U$: $D_U \leftarrow D_U \cup \tau_j$
19:     **end for**
20:     Train $\pi_h$ using $D_U$
21: **end for**

---

where $T$ and $T'$ are the number of trajectories collected in Intervention Samping and Subgoal Training steps at iteration $t$, respectively. Furthermore, $\tau_j$ represents a trajectory and $w_{\text{int},g'}^{\tau_j}$, $w_{\text{exp}}^{\tau_j}$, and $w_{\text{train},g''}^{\tau_j}$ represent system probes corresponding to the different steps of the algorithm in trajectory $\tau_j$, which are further explained below. In Intervention Sampling (Algorithm 2), in each trajectory $\tau_j$, we randomly select a subgoal $g_i \in \text{IS}_t$ and perform an intervention on it. After each intervention, explorational data is gathered as specified in Definition 3.3. A trajectory is terminated when all subgoals in $\text{IS}_t$ have been achieved or when the horizon is reached. Note that, during an intervention on a randomly chosen subgoal $g_i$, it may be necessary to first achieve other subgoals $g' \in \text{IS}_t$. Therefore, for each $\tau_j$, we consider the system probes obtained toward achieving each subgoal $g' \in \text{IS}_t$ regardless of whether it was selected as the intervention subgoal ($g_i$) or not. We denote this cost by $w_{\text{int},g'}^{\tau_j}$ and it is equal to the system probes between achieving two consecutive subgoals, with $g'$, being the second one. Additionally, we need to consider collected explorational data during the trajectory $\tau_j$ to explore the environment and discover reachable subgoals. The number of system probes for exploration in trajectory $\tau_j$ is denoted by $w_{\text{exp}}^{\tau_j}$.

In Subgoal Training, for each reachable subgoal in $\text{CCS}_t$, which is denoted by $g''$, we gather $T'$ trajectories to train the policy for achieving it. For each trajectory $\tau_j$, similar to the reasoning above, we consider the system probes obtained toward achieving each subgoal in $\text{IS}_t$. Note that $w_{\text{train},g''}^{\tau}$ represents the portion of system probes towards achieving the subgoal $g''$, during the intervention on $g''$ by taking actions based on the subgoal-based policy $\pi(a \mid s, g'')$.

As we mentioned in section 5 we define a recursive formula denoted by $C_{g_n}(\mathcal{I})$ that represents the expected cost of achieving the final subgoal $g_n$ when the set $\mathcal{I}$ is intervened on:

$$C_{g_n}(\mathcal{I}) = \sum_{g_{\text{sel}} \in \Phi} p_{\mathcal{I}, \mathcal{I} \cup \{g_{\text{sel}}\}} \left[ C_{trans}(\mathcal{I}, \mathcal{I} \cup \{g_{\text{sel}}\}) + C_{g_n}(\mathcal{I} \cup \{g_{\text{sel}}\}) \right], \tag{6}$$

where it has two terms: 1-the transition cost and 2-the cost-to-go. Specifically, the transition cost $C_{trans}(\mathcal{I}, \mathcal{I} \cup \{g_{\text{sel}}\})$ is the cost of transition from the current interventions set $\mathcal{I}$ to a new set by adding $g_{\text{sel}}$ to it ($\mathcal{I} \cup \{g_{\text{sel}}\}$). The cost-to-go $C_{g_n}(\mathcal{I} \cup \{g_{\text{sel}}\})$ shows the future costs from the new state onwards. Both components are weighted by the transition probability $p_{\mathcal{I}, \mathcal{I} \cup \{g_{\text{sel}}\}}$, which shows the probability of transition from state $\mathcal{I}$ to $\mathcal{I} \cup \{g_{\text{sel}}\}$ which is determined by the strategy. $C_{trans}(\mathcal{I}, \mathcal{I} \cup \{g_{\text{sel}}\})$ is indeed the expected of the cost formulated in (5):

$$\mathbb{E}\left[\sum_{j=1}^{T}\left(\sum_{g'\in\mathcal{I}\cup\{g_{\text{sel}}\}}w^{\tau_j}_{\text{int},g'}+w^{\tau_j}_{\text{exp}}\right)+\sum_{g''\in N(\mathcal{I}\cup\{g_{\text{sel}}\})}\sum_{j=1}^{T'}\left(\sum_{g'\in\mathcal{I}\cup\{g_{\text{sel}}\}}w^{\tau_j}_{\text{int},g'}+w^{\tau_j}_{\text{train},g''}\right)\mid\mathcal{I},g_{\text{sel}}\right],\tag{7}$$

where $N(\mathcal{I}\cup\{g_{\text{sel}}\})$ contains reachable subgoals having interventions on the subgoals in the set $\mathcal{I}\cup\{g_{\text{sel}}\}$. This set is equal to $\text{CCS}_t$ in Algorithm 1. We assume $\mathbb{E}[w^{\tau_j}_{\text{int},g'}]=\mathbb{E}[w^{\tau_j}_{\text{exp}}]=\mathbb{E}[w^{\tau_j}_{\text{train},g''}]=w$. Therefore, $C_{trans}(\mathcal{I},\mathcal{I}\cup\{g_{\text{sel}}\})$ will be simplified to:

$$T(|\mathcal{I}|+2)w+\sum_{g''\in N(\mathcal{I}\cup\{g_{\text{sel}}\})}T'(|\mathcal{I}|+2)w.\tag{8}$$

Equation (6) can be represented in matrix form:

$$\mathbf{C_{g_n}}=(P\circ C_{\text{trans}})\mathbf{1}+P\mathbf{C_{g_n}},\tag{9}$$

where $\mathbf{C_{g_n}}$ is a $2^n\times 1$ column vector, $P$ and $C_{\text{trans}}$ are $2^n\times 2^n$ matrices, and $\mathbf{1}$ is a $2^n\times 1$ column vector of ones.

We define $R=(P\circ C_{\text{trans}})\mathbf{1}$ where $R$ is a $2^n\times 1$ column vector:

$$\begin{bmatrix}R_{\mathcal{I}_1}\\R_{\mathcal{I}_2}\\\vdots\\R_{\mathcal{I}_{2^n}}\end{bmatrix},$$

the elements of which are defined by:

$$R_{\mathcal{I}_j}=\begin{cases}\displaystyle\sum_{g_{\text{sel}}\in\Phi\backslash\mathcal{I}_j}p_{\mathcal{I}_j,\mathcal{I}_j\cup\{g_{\text{sel}}\}}\left[T(|\mathcal{I}_j|+2)w+\sum_{g''\in N(\mathcal{I}_j\cup\{g_{\text{sel}}\})}T'(|\mathcal{I}_j|+2)w\right],&g_n\notin\mathcal{I}_j\\0,&g_n\in\mathcal{I}_j\end{cases}\tag{10}$$

If we expand (9):

$$\mathbf{C_{g_n}}=R+P\mathbf{C_{g_n}}$$
$$=R+P(R+P\mathbf{C_{g_n}})=R+PR+P^2\mathbf{C_{g_n}},$$

which further expands to $(I+P+P^2+\ldots)R$. Since $P^iR=0$ for $i\geq n$, we can truncate the series at $P^{n-1}R$. This truncation is justified because, beyond this point, all subgoals have been added to the set $\mathcal{I}$, including $g_n$, and the algorithm terminates and results in no additional transition costs. Therefore,

$$\mathbf{C_{g_n}}=(I+P+P^2+\ldots+P^{n-1})R$$

$$=\left[\begin{pmatrix}1&0&0&\cdots\\0&1&0&\cdots\\0&0&1&\cdots\\\vdots&\vdots&\vdots&\ddots\end{pmatrix}+\begin{pmatrix}0&q^{(1)}_{12}&q^{(1)}_{13}&\cdots\\0&0&q^{(1)}_{23}&\cdots\\0&0&0&\cdots\\\vdots&\vdots&\vdots&\ddots\end{pmatrix}+\ldots+\begin{pmatrix}0&q^{(n-1)}_{12}&q^{(n-1)}_{13}&\cdots\\0&0&q^{(n-1)}_{23}&\cdots\\0&0&0&\cdots\\\vdots&\vdots&\vdots&\ddots\end{pmatrix}\right]\begin{pmatrix}R_{\mathcal{I}_1}\\R_{\mathcal{I}_2}\\\vdots\\R_{\mathcal{I}_{2^n}}\end{pmatrix}.$$

$q^{(k)}_{ij}$ denote the $ij$-th entry of the $k$-th power of the transition probability matrix $P$. In other words, $(P^k)_{ij}=q^{(k)}_{ij}$. The expected cost of training is equal to the first entry in $\mathbf{C_{g_n}}$, where we start from primitive actions and $\mathcal{I}=\{\}$. Therefore,

$$C_{g_n}[\{\}]=R_{\mathcal{I}_1}+\sum_j q^{(1)}_{1j}R_{\mathcal{I}_j}+\sum_j q^{(2)}_{1j}R_{\mathcal{I}_j}+\sum_j q^{(3)}_{1j}R_{\mathcal{I}_j}\ldots+\sum_j q^{(n-1)}_{1j}R_{\mathcal{I}_j}.\tag{11}$$

*Remark* E.1. The condition $PA^{\hat{\mathcal{G}}_t}_{g_i}\subset\text{IS}_t$ in line 8 of Algorithm 1, is added because we assume that for a given $g_i\in CH^{\hat{\mathcal{G}}_t}_{\text{IS}_t}$, all of its recovered parents (i.e., $PA^{\hat{\mathcal{G}}_t}_{g_i}$), are needed to be achieved before achieving $g_i$. Hence, this condition ensures that all of them are in set $\text{IS}_t$. However, under Assumption 7.1, we can remove this condition, as all necessary parents of $g_i\in CH^{\hat{\mathcal{G}}_t}_{\text{IS}_t}$ (i.e., for AND subgoal all of its parents in $\mathscr{G}$ and for OR subgoal at least one of its parents in $\mathscr{G}$) are already in $\text{IS}_t$.

**E.2. Analysis for the Tree**

In this section, we analyze the cost associated with a subgoal structure modeled as a tree $G(n, b)$, where $n$ represents the number of nodes and $b$ the branching factor. First we rewrite equation (10):

$$
R_{\mathcal{I}_j} = \begin{cases} \displaystyle\sum_{g_{\text{sel}} \in \Phi \setminus \mathcal{I}_j} p_{\mathcal{I}_j, \mathcal{I}_j \cup \{g_{\text{sel}}\}} \left[ T(|\mathcal{I}_j| + 2)w + \sum_{g'' \in N(\mathcal{I}_j \cup \{g_{\text{sel}}\})} T'(|\mathcal{I}_j| + 2)w \right], & g_n \notin \mathcal{I}_j \\ 0, & g_n \in \mathcal{I}_j \end{cases}
$$

In the tree structure, each subgoal has only one parent and only becomes reachable by that. Hence:

$$
R_{\mathcal{I}_j} = \begin{cases} \sum_{g_{\text{sel}} \in (\Phi \setminus \mathcal{I}_j)} p_{\mathcal{I}_j, \mathcal{I}_j \cup \{g_{\text{sel}}\}} \left[ T(|\mathcal{I}_j| + 2)w + bT'(|\mathcal{I}_j| + 2)w \right], & g_n \notin \mathcal{I}_j, \\ 0, & g_n \in \mathcal{I}_j. \end{cases}
$$

Note that $\sum_{g_{\text{sel}} \in (\Phi \setminus \mathcal{I}_j)} p_{\mathcal{I}_j, \mathcal{I}_j \cup \{g_{\text{sel}}\}} = 1$. Therefore,

$$
R_{\mathcal{I}_j} = \begin{cases} (|\mathcal{I}_j| + 2)w(T + bT'), & g_n \notin \mathcal{I}_j, \\ 0, & g_n \in \mathcal{I}_j. \end{cases}
$$

We assume that the final subgoal is positioned at the depth $D$. Consequently, Algorithm 1 is expected to perform at most $D \leq \log_b(n)$ interventions before termination, where $n$ is the total number of nodes and $b$ is the branching factor.

In the (11), $R_{\mathcal{I}_j}$ is zero if $g_n \in \mathcal{I}_j$. This implies that for any $k$, the term $q_{1j}^k R_{\mathcal{I}_j}$ should not be included in the cost calculation if $g_n$ is in $\mathcal{I}_j$. Additionally, $q_{1j}^k$ represents the probability of transition from the initial subgoal to the subset $\mathcal{I}_j$ where $|\mathcal{I}_j| = k$. Please note that the values of $R_{\mathcal{I}_j}$ are the same for each term in the expression $\sum_j q_{1j}^{(k)} R_{\mathcal{I}_j}$ for every $j$ where $g_n \notin \mathcal{I}_j$ as $|\mathcal{I}_j| = k$. Thus:

$$
\sum_j q_{1j}^{(k)} R_{\mathcal{I}_j} = \sum_{j \text{ where } g_n \notin \mathcal{I}_j} q_{1j}^{(k)} R_{\mathcal{I}_j} + \sum_{j \text{ where } g_n \in \mathcal{I}_j} q_{1j}^{(k)} R_{\mathcal{I}_j} \tag{12}
$$

$$
= (k + 2)w(T + bT') \sum_{j \text{ where } g_n \notin \mathcal{I}_j} q_{1j}^{(k)} = Q_{\text{NG}}^{(k)} \times (k + 2)w(T + bT'), \tag{13}
$$

where $Q_{\text{NG}}^{(k)} = \sum_{j \text{ where } g_n \notin \mathcal{I}_j} q_{1j}^{(k)}$ is the probability of reaching a state such as $\mathcal{I}_j$ where $|\mathcal{I}_j| = k$ and $g_n \notin \mathcal{I}_j$ after $k$ transitions from the initial state $\{\}$ in $\mathcal{M}$ (we call this sequence of transitions in $\mathcal{M}$ as a path). Note that for $k < D$, $Q_{\text{NG}}^{(k)} = 1$ since the final subgoal cannot be intervened before all of its ancestors have been intervened.

Now, for $k \geq D$, we derive the probability that the final subgoal is in a set such as $\mathcal{I}_j$ where $|\mathcal{I}_j| = k$. In a tree structure, each node has only one parent and $b$ children. Thus, after adding each node to the intervention set, all of its children become controllable. Let $f(h)$ represent the number of controllable subgoals having intervened on $h$ subgoals in the tree. It can be easily shown that:

$$
f(h) = hb - h + 1 = h(b - 1) + 1,
$$

as $hb$ is the total number of controllable subgoals after intervening on $h$ subgoals, $-h$ accounts for the subgoals that have already been added to the intervention set, and $+1$ represents the root subgoal.

In order to calculate $Q_{\text{NG}}^{(k)}$, since $Q_{\text{NG}}^{(k)} = 1 - Q_{\text{G}}^{(k)}$, we first an upper bound on $Q_{\text{G}}^{(k)}$. We know that there is only one path in the subgoal structure that leads from the root to the final subgoal. To add the final subgoal to the intervention set, we must select exactly one controllable node from the controllable set, at each iteration. The probability of making the correct selection is $\frac{1}{f(h)}$. On the other hand, for $k \geq D$, the final subgoal might be added to the intervention set in any position (iteration). Hence, the probability that the final subgoal is in a set $\mathcal{I}_j$ where $|\mathcal{I}_j| = k$ is as follows:

$$
Q_{\text{G}}^{(k)} = \sum_{d=D}^{k} \prod_{h=1}^{d} \frac{1}{f(h)}. \tag{14}
$$

Note that the outer sum in the (14), is because $\prod_{h=1}^{d} \frac{1}{f(h)}$ is the probability that the final subgoal is added to set $\mathcal{I}_j$ at last (iteration $d$). Therefore, the sum considers all possible positions(iterations) where the final goal can be added to $\mathcal{I}_j$ (from $D$ to $k$). Based on this, we show that there exist a constant $\alpha$ where $Q_{NG}^{(k)} = (1 - Q_G^{(k)}) \geq \alpha > 0$ for $k \geq D$:

$$Q_G^{(k)} = \sum_{d=D}^{k} \prod_{h=1}^{d} \frac{1}{h(b-1)+1} \leq \sum_{d=D}^{k} \prod_{h=1}^{d} \frac{1}{h(b-1)} \tag{15}$$

$$= \sum_{d=D}^{k} \frac{1}{d!(b-1)^d} \tag{16}$$

$$\overset{(a)}{\leq} \sum_{d=D}^{k} \frac{1}{d!} \tag{17}$$

$$\overset{(b)}{\leq} \sum_{d=D}^{k} \frac{1}{2^{(d-1)}} \tag{18}$$

$$\overset{(c)}{=} \frac{2}{2^{D-1}} \left(1 - \frac{1}{2^{k-D+1}}\right) \tag{19}$$

$$\leq \frac{2}{2^{D-1}}. \tag{20}$$

$(a)$ $(b-1)^d \geq 1$ for $b > 1$.
$(b)$ Because for any $d \geq 1$: $\frac{1}{d!} \leq \frac{1}{2^{(d-1)}}$.
$(c)$ Sum of a geometric series with a first term $\frac{1}{2^{(D-1)}}$ and ratio $\frac{1}{2}$.

Therefore for $k \geq D$,

$$Q_{NG}^{(k)} = (1 - Q_G^{(k)}) \geq (1 - \frac{2}{2^{D-1}}) = \alpha. \tag{21}$$

Now, we calculate the total cost using the (11):

$$\mathbf{C_{g_n}}[\{\}] = R_{\mathcal{I}_1} + \underbrace{\sum_j q_{1j}^{(1)} R_{\mathcal{I}_j} + \ldots + \sum_j q_{1j}^{(D-1)} R_{\mathcal{I}_j}}_{\text{"paths in } \mathcal{M} \text{ where the final subgoal cannot be reached"}} + \underbrace{\sum_j q_{1j}^{(D)} R_{\mathcal{I}_j} + \ldots + \sum_j q_{1j}^{(n-1)} R_{\mathcal{I}_j}}_{\text{"paths in } \mathcal{M} \text{ where the final subgoal can be reached"}} \tag{22}$$

$$= \underbrace{2w(T+bT') + \sum_{k=1}^{D-1} Q_{NG}^{(k)}(k+2)w(T+bT')}_{\text{"paths in } \mathcal{M} \text{ where the final subgoal cannot be reached"}} + \underbrace{\sum_{k=D}^{n-1} Q_{NG}^{(k)}(k+2)w(T+bT')}_{\text{"paths in } \mathcal{M} \text{ where the final subgoal can be reached"}} \tag{23}$$

$$\overset{(a)}{\geq} \underbrace{2w(T+bT') + \sum_{k=1}^{D-1} (k+2)w(T+bT')}_{\text{"paths in } \mathcal{M} \text{ where the final subgoal cannot be reached"}} + \underbrace{\sum_{k=D}^{n-1} \alpha(k+2)w(T+bT')}_{\text{"paths in } \mathcal{M} \text{ where the final subgoal can be reached"}} \tag{24}$$

$$\geq 2w(T+bT') + \sum_{k=1}^{n-1} \alpha(k+2)w(T+bT') \tag{25}$$

$$= \frac{\alpha}{2} n(n+3)w(T+bT'). \tag{26}$$

where in $(a)$, we used (21). Therefore for HRC$_b$, the complexity is $\Omega(n^2 b)$, while for the targeted strategies in Section 6, the sum ends on $k \leq D \leq \log_b(n)$ since they add only the ancestors of the final subgoal to the intervention set in the worst-case scenario (based on Assumption 7.2). Therefore, the complexity of our targeted strategies is $O(\log^2(n)b)$.

### E.3. Analysis for Random Graph

In this section, we analyze the cost of training in subgoal structures modeled by a random graph which we call semi-Erdős–Rényi graph. In particular, semi-Erdős–Rényi $G(n, p)$ is a directed graph with the vertex set $V = \{1, 2, \ldots, n\}$ and the directed edge set $E$. Each node $i \in V$, corresponds with a subgoal $g_i$. The edges in the graph are generated as follows: For each node $i \in \{1, 2, \ldots, n-1\}$, iterate over each node $j$ such that $i < j$ and add a directed edge from node $i$ to node $j$ with probability $p$, where $p$ is the edge probability. Formally, for each pair of nodes $(i, j)$ such that $i < j$:

$$\Pr((i, j) \in E) = p,$$

where the existence of an edge $(i, j)$ in $E$ is determined independently from other edges. In our analysis, we consider $p = \frac{c \log(n)}{n-1}$, where is a positive constant $c > 0$.

**Lower bound for semi-Erdős–Rényi $G(n, p)$ for HRC$_b$ algorithm**

From (11), the total expected of cost is:

$$\mathbb{E}\Big[C_{g_n}[\{\}]\Big] = \mathbb{E}\Big[R_{\mathcal{I}_1}\Big] + \mathbb{E}\Big[\sum_j q_{1j}^{(1)} R_{\mathcal{I}_j}\Big] + \mathbb{E}\Big[\sum_j q_{1j}^{(2)} R_{\mathcal{I}_j}\Big] + \ldots + \mathbb{E}\Big[\sum_j q_{1j}^{(n-1)} R_{\mathcal{I}_j}\Big]. \tag{27}$$

Based on the definition of $R_{\mathcal{I}_j}$, we know:

$$\sum_j q_{1j}^{(k)} R_{\mathcal{I}_j} = \sum_{j \text{ where } g_n \notin \mathcal{I}_j} q_{1j}^{(k)} R_{\mathcal{I}_j} + \sum_{j \text{ where } g_n \in \mathcal{I}_j} q_{1j}^{(k)} R_{\mathcal{I}_j} \!\!\!\!\nearrow^{0} \tag{28}$$

Hence, we can rewrite (27) as follows:

$$\mathbb{E}\Big[C_{g_n}[\{\}]\Big] = \mathbb{E}\Big[R_{\mathcal{I}_1}\Big] + \sum_{k=1}^{n-1} \mathbb{E}\Big[\sum_{j \text{ where } g_n \notin \mathcal{I}_j} q_{1j}^{(k)} R_{\mathcal{I}_j}\Big] = \mathbb{E}\Big[R_{\mathcal{I}_1}\Big] + \sum_{k=1}^{n-1} \sum_{j \text{ where } g_n \notin \mathcal{I}_j} \mathbb{E}\Big[q_{1j}^{(k)} R_{\mathcal{I}_j}\Big]. \tag{29}$$

We drive a lower bound for $R_{\mathcal{I}_j}$ for $j$ that $g_n \notin \mathcal{I}_j$. Note that since we are looking for the lower bound for the cost, we consider particular subgoal structures where subgoals are all of "OR" type:

$$R_{\mathcal{I}_j} = \sum_{g_{\text{sel}} \in (\Phi \setminus \mathcal{I}_j)} p_{\mathcal{I}_j, \mathcal{I}_j \cup \{g_{\text{sel}}\}} \Big[ wT(|\mathcal{I}_j| + 2) + \sum_{g'' \in N(\mathcal{I}_j \cup \{g_{\text{sel}}\})} wT'(|\mathcal{I}_j| + 2) \Big] \tag{30}$$

$$\geq \sum_{g_{\text{sel}} \in (\Phi \setminus \mathcal{I}_j)} p_{\mathcal{I}_j, \mathcal{I}_j \cup \{g_{\text{sel}}\}} \Big[ wT(|\mathcal{I}_j| + 2) \Big] \tag{31}$$

$$= wT(|\mathcal{I}_j| + 2), \tag{32}$$

where the last equality is because the sum of the probabilities should equal 1.

Now for each $k$:

$$\sum_{j \text{ where } g_n \notin \mathcal{I}_j} \mathbb{E}\Big[q_{1j}^{(k)} R_{\mathcal{I}_j}\Big]$$

$$\geq \sum_{j \text{ where } g_n \notin \mathcal{I}_j} \mathbb{E}[q_{1j}^{(k)}] wT(|\mathcal{I}_j| + 2)$$

$$= wT(k + 2) \sum_{j \text{ where } g_n \notin \mathcal{I}_j} \mathbb{E}[q_{1j}^{(k)}], \tag{33}$$

which means that we only need to compute $\sum_{j \text{ where } g_n \notin \mathcal{I}_j} \mathbb{E}[q_{1j}^{(k)}]$, which is the sum of the expected probability of the paths (in MDP $\mathcal{M}$) that start from the state $\{\}$ and reach to a state $\mathcal{I}_j$ that does not include the final subgoal after $k$ transitions ($g_n \notin \mathcal{I}_j$ and $|\mathcal{I}_j| = k$). We know that,

$$\sum_{j \text{ where } g_n \in \mathcal{I}_j} \mathbb{E}[q_{1j}^{(k)}] + \sum_{j \text{ where } g_n \notin \mathcal{I}_j} \mathbb{E}[q_{1j}^{(k)}] = 1. \tag{34}$$

Now consider each $\mathcal{I}_j$ where $g_n \notin \mathcal{I}_j$ and $|\mathcal{I}_j| = k$. We can create a new set $\mathcal{I}_{j_i}$ by replacing $g_i \in \mathcal{I}_j$ with $g_n$ (see Figure 8). Let us denote this new set with $\mathcal{I}_{j_i}$ (the index of this new set is denoted by $j_i$).

*Figure 8.* Left: The gray nodes represent elements of the set $\mathcal{I}_j$ with size $k$, where subgoal $g_n \notin \mathcal{I}_j$. Right: A transformation of the set $\mathcal{I}_j$ to a new set $\mathcal{I}_{j_i}$, where one element (in this case, node $g_i = 3$) in $\mathcal{I}_j$ is replaced by $g_n$, as shown by the gray node $g_n$ on the right.

Based on Lemma E.3 we have $\mathbb{E}[q_{1j_i}^{(k)}] \leq \mathbb{E}[q_{1j}^{(k)}]$. Hence, for each such $\mathcal{I}_j$ there exist $k$ number of $\mathcal{I}_{j_i}$s, where $\mathbb{E}[q_{1j_i}^{(k)}] \leq \mathbb{E}[q_{1j}^{(k)}]$. Summing all of these inequalities gives:

$$\sum_{g_i \in \mathcal{I}_j} \mathbb{E}[q_{1j_i}^{(k)}] \leq k \mathbb{E}[q_{1j}^{(k)}]. \tag{35}$$

Note that each $\mathcal{I}_{j_i}$ can be created from $(n - k)$ different $\mathcal{I}_j$s. Hence, summing all the inequalities from 35 for different $j$s, each $\mathbb{E}[q_{1j_i}^{(k)}]$ (or the corresponding $\mathbb{E}[q_{1j}^{(k)}]$ where $g_n \in \mathcal{I}_j$ ) will be counted $(n - k)$ times:

$$(n - k) \sum_{j \text{ where } g_n \in \mathcal{I}_j} \mathbb{E}[q_{1j}^{(k)}] \leq k \sum_{j \text{ where } g_n \notin \mathcal{I}_j} \mathbb{E}[q_{1j}^{(k)}]. \tag{36}$$

By adding $\sum_{j \text{ where } g_n \in \mathcal{I}_j} \mathbb{E}[q_{1j}^{(k)}]$ to the both sides and rearranging the terms:

$$\sum_{j \text{ where } g_n \in \mathcal{I}_j} \mathbb{E}[q_{1j}^{(k)}] \leq \frac{k}{n - k} \sum_{j \text{ where } g_n \notin \mathcal{I}_j} \mathbb{E}[q_{1j}^{(k)}]. \tag{37}$$

Therefore,

$$\sum_{j \text{ where } g_n \notin \mathcal{I}_j} \mathbb{E}[q_{1j}^{(k)}] + \sum_{j \text{ where } g_n \in \mathcal{I}_j} \mathbb{E}[q_{1j}^{(k)}] \leq \sum_{j \text{ where } g_n \notin \mathcal{I}_j} \mathbb{E}[q_{1j}^{(k)}] + \frac{k}{n - k} \sum_{j \text{ where } g_n \notin \mathcal{I}_j} \mathbb{E}[q_{1j}^{(k)}]. \tag{38}$$

The left-hand side is 1 and the right hand side is $\frac{n}{n-k} \sum_{j \text{ where } g_n \notin \mathcal{I}_j} \mathbb{E}[q_{1j}^{(k)}]$ , hence:

$$1 \leq \frac{n}{n - k} \sum_{j \text{ where } g_n \notin \mathcal{I}_j} \mathbb{E}[q_{1j}^{(k)}]. \tag{39}$$

Thus,

$$\sum_{j \text{ where } g_n \notin \mathcal{I}_j} \mathbb{E}[q_{1j}^{(k)}] \geq 1 - \frac{k}{n}. \tag{40}$$

Now we compute the total cost using (29) as follows:

$$\mathbb{E}\left[C_{g_n}[\{\}]\right] = \mathbb{E}\left[R_{\mathcal{I}_1}\right] + \sum_{k=1}^{n-1} \sum_{j \text{ where } g_n \notin \mathcal{I}_j} \mathbb{E}\left[q_{1j}^{(k)} R_{\mathcal{I}_j}\right]$$

$$\overset{(a)}{\geq} 2wT + wcT' \log(n) + \sum_{k=1}^{n-1} \left[ wT(k+2) \times \sum_{j \text{ where } g_n \notin \mathcal{I}_j} \mathbb{E}[q_{1j}^{(k)}] \right]$$

$$\overset{(b)}{\geq} 2wT + wcT' \log(n) + \sum_{k=1}^{n-1} \left[ wT(k+2)(1 - \frac{k}{n}) \right]$$

$$\geq 2wT + wcT' \log(n) + \frac{wT}{n} \frac{n(n-1)(3n+5)}{6}.$$

$(a)$ We used inequality in 33 in the third term.
$(b)$ We used inequality in 40 in the third term.

Therefore, the complexity of HRC$_b$ algorithm is $\Omega(n^2)$. In the following, we analyze the complexity of our suggested targeted strategies for this type of random graph.

**Upper bound for semi-Erdős–Rényi** $G(n, p)$ **where** $p = \frac{c \log(n)}{(n-1)}$ **for HRC$_h$** Due to the assumptions made in Section 7, our ranking rules in Section 6 intervene on all ancestors of the final subgoal in the worst-case scenario. Hence, we first obtain an upper bound on the expected number of nodes that are ancestors of the final subgoal.

For a given node $i$, we define the event $A_i$ as the probability that $i$ is the ancestor of node $n$. The probability of $A_i$ is as follows:

$$P(A_i) = \underbrace{P(i \text{ connected to } n)}_{(a)} + \underbrace{P(i \text{ not connected to } n)}_{(b)} \underbrace{P\left( \bigcup_{j=i+1}^{n-1} (j \text{ is a child of } i) \cap A_j \right)}_{(c)} \tag{41}$$

$$\leq p + (1-p) \underbrace{\sum_{j=i+1}^{n-1} p \times P(A_j)}_{(d)} \tag{42}$$

$$= p + (1-p)p \sum_{j=i+1}^{n-1} P(A_j), \tag{43}$$

where $(a)$, represents the probability that node $i$ is directly connected to node $n$ (which is $p$), and $(b)$ is the probability that node $i$ is not connected to node $n$ (which is $1 - p$). If $i$ is not directly connected to $n$, it must be connected to at least one node $(j > i + 1)$ that is itself an ancestor of $n$. This probability is represented in $(c)$ and upper-bounded by union bound in term $(d)$. We define $q_i = \sum_{j=i}^{n-1} P(A_j)$. The goal is to find an upper bound on $q_1 = \sum_{j=i}^{n-1} P(A_j)$ which is an upper bound of the expected number of ancestors of $n$. Therefore, we can rewrite the above equation in a recurrence form as follows:

$$q_i - q_{i+1} \leq p + (1-p)p q_{i+1}. \tag{44}$$

Let $a = p - p^2 + 1$. The recurrence then becomes:

$$q_i \leq p + a q_{i+1}.$$

To solve the recurrence, let us begin with $i = n - 1$. For this case, we have:

$$q_{n-1} = \sum_{j=n-1}^{n-1} P(A_j) = P(A_{n-1}) = p,$$

where the last equality holds because node $n-1$ is an ancestor of node $n$ if and only if $n$ is the direct child of $n-1$. In general, for any $q_i$, $1 \le i \le n-1$, we can derive:

$$q_i \le p \sum_{k=0}^{n-1-i} a^k.$$

To find $q_1$, we sum the series from $k = 0$ to $k = n - 2$:

$$q_1 \le p \frac{a^{n-1} - 1}{a - 1}.$$

Finally, we substitute $a = p - p^2 + 1$ and $p = \frac{c \log n}{n-1}$ into the above equation:

$$q_1 \le \frac{\left( \frac{c \log n}{n-1} - \left( \frac{c \log n}{n-1} \right)^2 + 1 \right)^{n-1} - 1}{\left( 1 - \frac{c \log n}{n-1} \right)} \le \frac{\left( \frac{c \log n}{n-1} + 1 \right)^{n-1} - 1}{\left( 1 - \frac{c \log n}{n-1} \right)} \le \frac{\left( \frac{c \log n}{n-1} + 1 \right)^{n-1}}{\left( 1 - \frac{c \log n}{n-1} \right)}, \tag{45}$$

where the last two inequalities are because the terms $-\left( \frac{c \log n}{n-1} \right)^2$ and $-1$ are negative. The term $\left( \frac{c \log n}{n-1} + 1 \right)^{n-1}$ is upper bounded as follows:

$$\left( \frac{c \log n}{n-1} + 1 \right)^{n-1} \le e^{c \log n} = n^c.$$

Substituting back into the (45):

$$q_1 \le \frac{n^c}{1 - \frac{c \log n}{n-1}}.$$

For $n > 4$, $c < 1$, we have: $\frac{c \log n}{n-1} < \frac{1}{2}$. Hence:

$$q_1 \le 2n^c.$$

As we mentioned earlier, $q_1$ is the upper bound on the expected number of the nodes that are ancestors of the node $n$. Hence:

$$\mathbb{E}[\mathrm{ANC}_n(G)] \le \sum_{j=1}^{n-1} P(A_j) = q_1 \le 2n^c, \tag{46}$$

where $G$ is the generated random graph and $\mathrm{ANC}_n(G)$ represents the number of ancestors of the node $n$ in the graph $G$. The total cost is then computed as the expected sum of the transition costs:

$$\mathbb{E}\left[ \mathbb{E}\left[ \sum_{k=1}^{\mathrm{ANC}_n(G)} \left[ T(k+2)w + N_k^G \times T'(k+2)w \right] \Big| G \right] \right], \tag{47}$$

where the inner expectation is because of over the orders of adding ancestors to the interventions. Let $N_k^G$ represent the set of reachable subgoals when the $k$-th ancestor is added to the intervention set. We define $N(G)$ as the maximum degree of the graph $G$. Hence, for every $k$, $N_k^G \leq N(G)$. With further simplification:

$$\mathbb{E}\left[\mathbb{E}\left[\sum_{k=1}^{\text{ANC}_n(G)}\left[T(k+2)w + N_k^G \times T'(k+2)w\right] \,\Big|\, g\right]\right] \leq \mathbb{E}\left[\sum_{k=1}^{\text{ANC}_n(G)}[T(k+2)w + N(G)T'(k+2)w]\right]. \quad (48)$$

We can partition the last expectation into two terms:

$$\mathbb{E}\left[\sum_{k=1}^{\text{ANC}_n(G)}[T(k+2)w + N(G)T'(k+2)w]\right] = \mathbb{E}\left[\sum_{k=1}^{\text{ANC}_n(G)}T(k+2)w\right] + \mathbb{E}\left[N(G)\sum_{k=1}^{\text{ANC}_n(G)}T'(k+2)w\right]. \quad (49)$$

We now analyze the second term as it is the dominant term. We partition it into two expectations based on the following

event $N(G) < (1 + \delta)c\log(n)$:

$$\mathbb{E}\left[N(G)\sum_{k=1}^{\text{ANC}_n(G)} T'(k+2)w\right] \tag{50}$$

$$= \mathbb{E}\left[\left(N(G)\sum_{k=1}^{\text{ANC}_n(G)} T'(k+2)w\right)\mathbb{1}_{N(G)<(1+\delta)c\log(n)}\right]$$

$$+ \mathbb{E}\left[\left(N(G)\sum_{k=1}^{\text{ANC}_n(G)} T'(k+2)w\right)\mathbb{1}_{N(G)\geq(1+\delta)c\log(n)}\right] \tag{51}$$

$$\overset{(a)}{\leq} (1+\delta)c\log(n)\mathbb{E}\left[\left(\sum_{k=1}^{\text{ANC}_n(G)} T'(k+2)w\right)\mathbb{1}_{N(G)<(1+\delta)c\log(n)}\right]$$

$$+ n\mathbb{E}\left[\left(\sum_{k=1}^{\text{ANC}_n(G)} T'(k+2)w\right)\mathbb{1}_{N(G)\geq(1+\delta)c\log(n)}\right] \tag{52}$$

$$\overset{(b)}{\leq} (1+\delta)c\log(n)\mathbb{E}\left[\left(\sum_{k=1}^{\text{ANC}_n(G)} T'(k+2)w\right)\right]$$

$$+ n\mathbb{E}\left[\left(\sum_{k=1}^{\text{ANC}_n(G)} T'(k+2)w\right)\mathbb{1}_{N(G)\geq(1+\delta)c\log(n)}\right] \tag{53}$$

$$\overset{(c)}{=} (1+\delta)c\log(n)\mathbb{E}\left[\left(\sum_{k=1}^{\text{ANC}_n(G)} T'(k+2)w\right)\right]$$

$$+ n\mathbb{E}\left[\sum_{k=1}^{\text{ANC}_n(G)} T'(k+2)w \,\middle|\, N(G) \geq (1+\delta)c\log(n)\right]P\left(N(G) \geq (1+\delta)c\log(n)\right) \tag{54}$$

$$\overset{(d)}{=} wT'(1+\delta)c\log(n)\mathbb{E}\left[\sum_{k=1}^{\text{ANC}_n(G)}(k+2)\right]$$

$$+ wT'n\mathbb{E}\left[\sum_{k=1}^{\text{ANC}_n(G)}(k+2) \,\middle|\, N(G) \geq (1+\delta)c\log(n)\right]P\left(N(G) \geq (1+\delta)c\log(n)\right) \tag{55}$$

$$\overset{(e)}{\leq} wT'(1+\delta)c\log(n)\mathbb{E}\left[\sum_{k=1}^{\text{ANC}_n(G)}(k+2)\right]$$

$$+ wT'n(\frac{n^2+5n}{2})P\left(N(G) \geq (1+\delta)c\log(n)\right) \tag{56}$$

$(a)$ We move the max of $N(G)$ outside of the expectations.

$(b)$ Because $\mathbb{E}\left[\left(\sum_{k=1}^{\text{ANC}_n(G)} T'(k+2)w\right)\mathbb{1}_{N(G)<(1+\delta)c\log(n)}\right] \leq \mathbb{E}\left[\left(\sum_{k=1}^{\text{ANC}_n(G)} T'(k+2)w\right)\right]$.

$(c)$ Expanding the second expectation based on the law of total expectation.

$(d)$ Moving constants out of the expectations.

$(e)$ Replacing $\text{ANC}_n(G)$ with its max value $(n)$ and computing the expectation.

Now we should compute an upper bound on $\mathbb{E}\left[\sum_{k=1}^{\text{ANC}_n(G)}(k+2)\right]$ and $P\left(N(G) \geq (1+\delta)c\log(n)\right)$. First, we obtain an upper bound on the term $\mathbb{E}\left[\sum_{k=1}^{\text{ANC}_n(G)}(k+2)\right]$:

$$\mathbb{E}\left[\sum_{k=1}^{\text{ANC}_n(G)}(k+2)\right] = \mathbb{E}\left[\frac{\text{ANC}_n^2(G)+5\text{ANC}_n(G)}{2}\right] = \frac{\mathbb{E}[\text{ANC}_n^2(G)]}{2} + \frac{5}{2}\mathbb{E}[\text{ANC}_n(G)].$$

We derived an upper bound on $\mathbb{E}[\text{ANC}_n(G)]$ in (46). Hence we only need to find an upper bound on $\mathbb{E}[\text{ANC}_n^2(G)]$:

$$
\begin{aligned}
\mathbb{E}\left[\text{ANC}_n^2(G)\right] &= \mathbb{E}\left[\text{ANC}_n^2(G) \mid \text{ANC}_n(G) \geq h\mathbb{E}[\text{ANC}_n(G)]\right]P\left(\text{ANC}_n(G) \geq h\mathbb{E}[\text{ANC}_n(G)]\right) \\
&\quad + \mathbb{E}\left[\text{ANC}_n^2(G) \mid \text{ANC}_n(G) < h\mathbb{E}[\text{ANC}_n(G)]\right]P\left(\text{ANC}_n(G) < h\mathbb{E}[\text{ANC}_n(G)]\right) \\
&\stackrel{(a)}{\leq} \mathbb{E}\left[\text{ANC}_n^2(G) \mid \text{ANC}_n(G) \geq h\mathbb{E}[\text{ANC}_n(G)]\right]\frac{1}{h} \\
&\quad + \mathbb{E}\left[\text{ANC}_n^2(G) \mid \text{ANC}_n(G) < h\mathbb{E}[\text{ANC}_n(G)]\right] \\
&\stackrel{(b)}{\leq} \frac{n^2}{h} + h^2\mathbb{E}[\text{ANC}_n(G)]^2 \\
&\leq \frac{n^2}{h} + 4h^2n^{2c}.
\end{aligned}
$$

$(a)$ Using Markov's inequality $P\left(\text{ANC}_n(G) > h\mathbb{E}[\text{ANC}_n(G)]\right) \leq \frac{1}{h}$.

$(b)$ For each expectation we replace $\text{ANC}_n(G)$ with its maximum value and take it outside the expectation.

The critical point for $\frac{n^2}{h} + 4h^2n^{2c}$ is $h = \left(\frac{n^{2-2c}}{8}\right)^{1/3}$. By substituting this into the above inequality we obtain:

$$\mathbb{E}[\text{ANC}_n^2(G)] \leq \frac{n^2}{\left(\frac{n^{2-2c}}{8}\right)^{1/3}} + 4\left(\frac{n^{2-2c}}{8}\right)^{2/3}n^{2c} = 3n^{\frac{4}{3}+\frac{2}{3}c}.$$

Thus, $\mathbb{E}\left[\sum_{k=1}^{\text{ANC}_n(G)}(k+2)\right]$ can be upper bounded by:

$$
\begin{aligned}
\mathbb{E}\left[\sum_{k=1}^{\text{ANC}_n(G)}(k+2)\right] &= \frac{\mathbb{E}[\text{ANC}_n^2(G)]}{2} + \frac{5}{2}\mathbb{E}[\text{ANC}_n(G)] \\
&\leq \frac{3}{2}n^{\frac{4}{3}+\frac{2}{3}c} + 5n^c.
\end{aligned}
\tag{57}
$$

Now, we only need to find the upper bound on the probability $P\left(N(G) \geq (1+\delta)c\log(n)\right)$ in the (56). To obtain an upper bound, we analyze the case in which each node $i$ can have edges to all other $n-1$ nodes. Let $D_i$ denote the out-degree of node $i$. Therefore, each $D_i$ follows Binomial$(n-1, p)$ with expected value:

$$\mu_i = \mathbb{E}[D_i] = (n-1)p = c\log(n).$$

Using Chernoff bound in Lemma E.2:

$$P\left(D_i \geq (1+\delta)c\log(n)\right) \leq \exp\left(-\frac{\delta^2 c\log(n)}{2+\delta}\right) = \exp\left(-\frac{\delta^2 c\log(n)}{2+\delta}\right) = \frac{1}{n^{\frac{\delta^2}{2+\delta}c}}.$$

The event that the maximum out-degree exceeds $(1 + \delta)c \log(n)$ is:

$$\{N(G) \geq (1 + \delta)c \log(n)\} = \bigcup_{i=1}^{n} \{D_i \geq (1 + \delta)c \log(n)\}.$$

Thus using union bound:

$$P\left(N(G) \geq (1 + \delta)c \log(n)\right) = P\left(\bigcup_{i=1}^{n} \{D_i \geq (1 + \delta)c \log(n)\}\right) \leq \sum_{i=1}^{n} P\left(D_i \geq (1 + \delta)c \log(n)\right) = \frac{n}{n^{\frac{\delta^2}{2+\delta}c}}. \quad (58)$$

We substitute equations (57) and (58) into equation (56):

$$wT'(1+\delta)c \log(n)\left[\frac{3}{2}n^{\frac{4}{3}+\frac{2}{3}c} + 5n^c\right] + wT'n(\frac{n^2 + 5n}{2})P\left(N(G) \geq (1 + \delta)c \log(n)\right) \quad (59)$$

$$\leq wT'(1+\delta)c \log(n)\left[\frac{3}{2}n^{\frac{4}{3}+\frac{2}{3}c} + 5n^c\right] + wT'n\left[\frac{n^2 + 5n}{2}\right]\frac{n}{n^{\frac{\delta^2}{2+\delta}c}}. \quad (60)$$

For any $0 < c < 1$, the first term in the above equation above is of order $n^{\frac{4}{3}+\frac{2}{3}c} \log(n)$ and remains less than $n^2$. For the second term, the exponent of $n$ is $4 - \frac{\delta^2}{2+\delta}c$, which we desire to be less than $\frac{4}{3} + \frac{2}{3}c$. For this purpose, $\delta$ should satisfy the following equation:

$$4 - \frac{\delta^2}{2+\delta}c \leq \frac{4}{3} + \frac{2c}{3},$$

which holds when

$$\delta \geq \frac{8 - 2c + \sqrt{-44c^2 + 160c + 64}}{6c}.$$

Note that for $0 < c < 1$, the expression under the square root, $-44c^2 + 160c + 64$, is positive. This ensures that $\delta$ has real solutions. Consequently, the dominant term in the above equation will be $n^{\frac{4}{3}+\frac{2}{3}c} \log(n)$. Therefore, the complexity of our ranking rules will be $O(n^{\frac{4}{3}+\frac{2}{3}c} \log(n))$.

### E.4. Auxiliary Lemma

**Lemma E.2** (Chernoff Bound, see (Mitzenmacher & Upfal, 2017)). *Let $X = \sum_{i=1}^{n} X_i$ be the sum of $n$ independent Bernoulli random variables, where each $X_i$ takes value $1$ with probability $p$ and $0$ otherwise. Let $\mu = \mathbb{E}[X] = np$. Then, for any $\delta > 0$,*

$$\Pr(X \geq (1 + \delta)\mu) \leq \exp\left(-\frac{\delta^2 \mu}{2 + \delta}\right).$$

**Lemma E.3.** *Under the setting outlined in Section E.1 and E.3, let $G(n, p)$ be a semi-Erdős–Rényi graph where $p = \frac{c \log(n)}{n-1}$ and $0 < c < 1$. For each $\mathcal{I}_j$ where $g_n \notin \mathcal{I}_j$ and $|\mathcal{I}_j| = k$, we create a new set $\mathcal{I}_{j_i}$ by replacing $g_i \in \mathcal{I}_j$ with $g_n$ (see Figure 8). Let us denote this new set with $\mathcal{I}_{j_i}$. Then:*

$$\mathbb{E}[q_{1j_i}^{(k)}] \leq \mathbb{E}[q_{1j}^{(k)}].$$

*Proof.* We define $\mathcal{I}_{j^{(i)}}$ as a new set by replacing $g_i \in \mathcal{I}_j$ with the node $g_{i+1}$. If $g_{i+1} \in \mathcal{I}_j$, $\mathbb{E}[q_{1j^{(i)}}^{(k)}] = \mathbb{E}[q_{1j}^{(k)}]$. Otherwise, we prove that

$$\mathbb{E}[q_{1j^{(i)}}^{(k)}] \leq \mathbb{E}[q_{1j}^{(k)}]. \quad (61)$$

Let $e$ denote the event that the edge $(i, i+1)$ exists in the graph (i.e., there is an edge from $g_i$ to $g_{i+1}$), which occurs with probability $p$, and let $\neg e$ denote the event that it does not exist, which occurs with probability $1 - p$. Then, we can write:

$$\mathbb{E}\left[q_{1j^{(i)}}^{(k)}\right] = (1 - p) \times \mathbb{E}\left[q_{1j^{(i)}}^{(k)} \mid \neg e\right] + p \times \mathbb{E}\left[q_{1j^{(i)}}^{(k)} \mid e\right].$$

For the first term, under the event $\neg e$, from the perspective of other nodes, $g_i$ and $g_{i+1}$ are the same. Therefore:

$$\mathbb{E}\left[q_{1j^{(i)}}^{(k)} \mid \neg e\right] = \mathbb{E}\left[q_{1j}^{(k)} \mid \neg e\right]. \tag{62}$$

For the second term, under the event $e$, we consider additional events. Let $e_i$ be the event that there is at least one edge from the set $\mathcal{I}_j \setminus \{g_i\}$ to $g_i$, and $e_{i+1}$ be the event that there is at least one edge from the set $\mathcal{I}_j \setminus \{g_i\}$ to $g_{i+1}$. Moreover, consider that $k' = |\{g_r | g_r \in \mathcal{I}_j \text{ and } r < i\}|$.

We can partition $\mathbb{E}\left[q_{1j^{(i)}}^{(k)} \mid e\right]$ into four cases based on $e_i$ and $e_{i+1}$:

Using the law of total expectation, we rewrite $\mathbb{E}\left[q_{1j^{(i)}}^{(k)} \mid e\right]$ as follows:

$$
\begin{aligned}
\mathbb{E}\left[q_{1j^{(i)}}^{(k)} \mid e\right] = {} & P(\neg e_{i+1}, e_i) \times \mathbb{E}\left[q_{1j^{(i)}}^{(k)} \mid e, \neg e_{i+1}, e_i\right] \\
& + P(e_{i+1}, \neg e_i) \times \mathbb{E}\left[q_{1j^{(i)}}^{(k)} \mid e, e_{i+1}, \neg e_i\right] \\
& + P(\neg e_{i+1}, \neg e_i) \times \mathbb{E}\left[q_{1j^{(i)}}^{(k)} \mid e, \neg e_{i+1}, \neg e_i\right] \\
& + P(e_{i+1}, e_i) \times \mathbb{E}\left[q_{1j^{(i)}}^{(k)} \mid e, e_{i+1}, e_i\right]
\end{aligned}
\tag{63}
$$

For each of the above cases, we show that the expectation is less than or equal to the corresponding expectation for $\mathcal{I}_j$.

Case 1: $e_i$ occurs and $e_{i+1}$ does not occur. In this case, there is at least one edge from $\mathcal{I}_j \setminus \{g_i\}$ to $g_i$, but there are no edges from $\mathcal{I}_j \setminus \{g_i\}$ to $g_{i+1}$. In this case, $P(\neg e_{i+1}, e_i) = \left[1 - (1 - p)^{k'}\right] \cdot (1 - p)^{k'}$. To add the node $g_{i+1}$ to the intervention set, $g_i$ must first be added. Hence, $\mathbb{E}\left[q_{1j^{(i)}}^{(k)} \mid e, \neg e_{i+1}, e_i\right] = 0$. Therefore,

$$\mathbb{E}\left[q_{1j^{(i)}}^{(k)} \mid e, \neg e_{i+1}, e_i\right] = 0 \leq \mathbb{E}\left[q_{1j}^{(k)} \mid e, \neg e_{i+1}, e_i\right].$$

Case 2: $e_i$ does not occur and $e_{i+1}$ occurs. Here, there is at least one edge from $\mathcal{I}_j \setminus \{g_i\}$ to $g_{i+1}$, but no edges from $\mathcal{I}_j \setminus \{g_i\}$ to $g_i$. In this case, $P(e_{i+1}, \neg e_i) = (1 - p)^{k'} \cdot \left[1 - (1 - p)^{k'}\right]$. If we merge nodes $g_i$ and $g_{i+1}$ into a dummy node, from the perspective of the other nodes, the dummy node in cases 1 and 2 can be treated similarly except that in case 1, paths in $\mathcal{M}$ that go to $g_{i+1}$ must include $g_i$, (thus $\mathbb{E}[q_{1j^{(i)}}^{(k)} \mid e, \neg e_{i+1}, e_i] = 0$) while in case 2, the opposite does not hold for $g_i$ and $\mathbb{E}[q_{1j}^{(k)} \mid e, e_{i+1}, \neg e_i] \neq 0$. Therefore, the expectation of visiting the set $\mathcal{I}_j$ in case 1 is bigger than the expectation of visiting $\mathcal{I}_{j^{(i)}}$ in case 2.

$$\mathbb{E}\left[q_{1j^{(i)}}^{(k)} \mid e, e_{i+1}, \neg e_i\right] \leq \mathbb{E}\left[q_{1j}^{(k)} \mid e, \neg e_{i+1}, e_i\right].$$

Note that this comparison is possible because $P(\neg e_{i+1}, e_i) = P(e_{i+1}, \neg e_i)$.

Case 3: Neither $e_i$ nor $e_{i+1}$ occur. In this case, there are no edges from $\mathcal{I}_j \setminus \{g_i\}$ to either $g_i$ or $g_{i+1}$. In this case, $P(\neg e_{i+1}, \neg e_i) = (1 - p)^{2k'}$.

As in Case 1, to add $g_{i+1}$ to the intervention set, $g_i$ must first be added. Hence,

$$\mathbb{E}\left[q_{1j^{(i)}}^{(k)} \mid e, \neg e_{i+1}, \neg e_i\right] = 0.$$

Therefore,

$$\mathbb{E}\left[q_{1j^{(i)}}^{(k)} \mid e, \neg e_{i+1}, \neg e_i\right] = 0 \leq \mathbb{E}\left[q_{1j}^{(k)} \mid e, \neg e_{i+1}, \neg e_i\right].$$

Case 4: Both $e_i$ and $e_{i+1}$ occur. Here, there is at least one edge from $\mathcal{I}_j \setminus \{g_i\}$ to both $g_i$ and $g_{i+1}$. In this case, $P(e_{i+1}, e_i) = \left[ 1 - (1-p)^{k'} \right]^2$ and both $g_i$ and $g_{i+1}$ have incoming edges from other nodes. Similar to case 2 we merge nodes $g_i$ and $g_{i+1}$ into a single dummy node.

However, some paths in $\mathcal{M}$ with length $k$ that include $g_{i+1}$, include $g_i$ too, thus will not be counted in $\mathbb{E}[q_{1j^{(i)}}^{(k)} \mid e, e_{i+1}, e_i]$) while the paths leading to $\mathcal{I}_j$ do not rely on $g_{i+1}$. Hence,

$$\mathbb{E}\left[ q_{1j^{(i)}}^{(k)} \mid e, e_{i+1}, e_i \right] \leq \mathbb{E}\left[ q_{1j}^{(k)} \mid e, e_{i+1}, e_i \right].$$

If we use all the inequality achieved in all the above cases,

$$
\begin{aligned}
\mathbb{E}\left[ q_{1j^{(i)}}^{(k)} \mid e \right] = & \underbrace{P(\neg e_{i+1}, e_i) \times 0}_{\text{case 1}} + \underbrace{P(e_{i+1}, \neg e_i) \times \mathbb{E}\left[ q_{1j^{(i)}}^{(k)} \mid e, e_{i+1}, \neg e_i \right]}_{\text{case 2}} \\
& + \underbrace{P(\neg e_{i+1}, \neg e_i) \times 0}_{\text{case 3}} + \underbrace{P(e_{i+1}, e_i) \times \mathbb{E}\left[ q_{1j^{(i)}}^{(k)} \mid e, e_{i+1}, e_i \right]}_{\text{case 4}} \\
\leq & \underbrace{P(\neg e_{i+1}, e_i) \times \mathbb{E}\left[ q_{1j}^{(k)} \mid e, \neg e_{i+1}, e_i \right]}_{\text{case 1 and 2}} + P(e_{i+1}, \neg e_i) \times \mathbb{E}\left[ q_{1j}^{(k)} \mid e, e_{i+1}, \neg e_i \right] \\
& + \underbrace{P(\neg e_{i+1}, \neg e_i) \times \mathbb{E}\left[ q_{1j}^{(k)} \mid e, \neg e_{i+1}, \neg e_i \right]}_{\text{case 3}} + \underbrace{P(e_{i+1}, e_i) \times \mathbb{E}\left[ q_{1j}^{(k)} \mid e, e_{i+1}, e_i \right]}_{\text{case 4}} = \mathbb{E}\left[ q_{1j}^{(k)} \mid e \right].
\end{aligned}
$$

Hence, the expectations in (63) are less than or equal to the corresponding expectations for $\mathcal{I}_j$. This, along with the (62), results in:

$$\mathbb{E}\left[ q_{1j^{(i)}}^{(k)} \right] \leq \mathbb{E}\left[ q_{1j}^{(k)} \right].$$

To construct the set $\mathcal{I}_{j_i}$, we perform a sequence of successive single swaps. We begin with the set $\mathcal{I}_{j^{(r)}}$, where $r = \max\{z : g_z \in \mathcal{I}_j\}$ represents the maximum index among nodes in set $\mathcal{I}_j$. The construction process is as follows:

1. First, we replace node $g_r$ with $g_{r+1}$ and construct $\mathcal{I}_{j^{(r)}}$.
2. Then replace $g_{r+1}$ in $\mathcal{I}_{j^{(r)}}$ by $g_{r+2}$.
3. Continue this process until $g_n$ is in the set.

At this point, we have constructed $\mathcal{I}_{j_r}$. If $i = r$, the process is complete. Otherwise, we continue a similar replacement process starting from set $\mathcal{I}_{j_r}$. In $\mathcal{I}_{j_r}$, we replace the node $g_{r'}$ where $r'$ is the second maximum index among nodes in set $\mathcal{I}_j$ ($r' = \max\{z : g_z \in \mathcal{I}_j \setminus \{g_r\}\}$) with its successive node $g_{r'+1}$ until we reach the node $r$. If $i = r'$ the process is complete, otherwise we repeat the process similarly.

Through this sequence of single node replacements, we can successfully construct the desired set $\mathcal{I}_{j_i}$. Since each replacement operation results in a set with lower expectation, when we construct $\mathcal{I}_{j_i}$, the following inequality holds:

$$\mathbb{E}\left[ q_{1j_i}^{(k)} \mid e \right] \leq \mathbb{E}\left[ q_{1j}^{(k)} \mid e \right] \qquad\qquad \square$$

## F. Ranking Rules

### F.1. Graph search algorithm

In the following, we describe the graph search algorithm (Algorithm 6) and also refer to the corresponding parts in HRC algorithm. The graph search algorithm initiates with empty sets CS and IS, along with $root\_subgoals$, which are the

subgoals with no parents in the subgoal structure $\mathscr{G}$. Sets IS and CS represent intervention and controllable sets, respectively. In line 1 (corresponding to line 2 of Algorithm 1), we add the $root\_subgoals$ to the controllable set CS. At each iteration, a subgoal $g_{\text{sel}}$, is selected from CS (corresponding to line 4 of Algorithm 1). Then, in line 4, $g_{\text{sel}}$ will be added to IS and removed from CS (corresponding to line 5 of Algorithm 1). Once the children of the selected subgoal $g_{\text{sel}}$ are determined using the estimated subgoal structure $\hat{\mathscr{G}}$ (corresponding to lines 5 to 8 of Algorithm 1), set CS is updated in line 7 to include these reachable subgoals (corresponding to lines 9 to 11 of Algorithm 1).

---

**Algorithm 6** Graph Search Algorithm

---

**Require:** Final subgoal $g_n$, empty set IS $= \{\}$ and CS $= \{\}$, $root\_subgoals$.
1: CS $\leftarrow root\_subgoals$
2: **while** $g_n \notin$ IS **do**
3:     Select a subgoal $g_{\text{sel}}$ from CS
4:     Remove $g_{\text{sel}}$ from CS and add it to IS
5:     **for all** subgoals $g' \in (CH^{\mathscr{G}}_{g_{\text{sel}}} \setminus (\text{IS} \cup \text{CS}))$ **do**
6:         {reachable subgoals become controllable}
7:         Add $g'$ to list CS
8:     **end for**
9: **end while**
10: **return** $2 * length(\text{IS}) + length(\text{CS})$

---

We define the cost of the graph search algorithm to be proportional to the training cost of Algorithm 1. In Section E.1, equation (8) shows the expected transition cost if $\mathbb{E}[w^{\tau_j}_{\text{int},g'}] = \mathbb{E}[w^{\tau_j}_{\text{exp}}] = \mathbb{E}[w^{\tau_j}_{\text{train},g''}] = w$. In this equation, if we assume $|\mathcal{I}| = 0$ (i.e., there is no cost to intervene on subgoals in $\mathcal{I}$), $T = T'$, and $U = 2wT$, the equation will be as follows:

$$U + \sum_{g'' \in N(\mathcal{I} \cup \{g_{\text{sel}}\})} U. \tag{64}$$

If we consider $U$ as a unit of cost, during each transition, the above equation counts 1 unit for Intervention Sampling and $N(\mathcal{I} \cup g_{\text{sel}}\})$ units for Subgoal Training steps. In other words, this shows the number of additions to the intervention set and controllable set, during the run of the algorithm 1, hence we define the cost of running the graph search algorithm after the termination as follows:

$$c = 2 \times |\text{IS}| + |\text{CS}|, \tag{65}$$

The cost is calculated as the total number of additions to sets IS and CS, during executing the graph search algorithm, which is equal to twice the number of nodes in the intervention set IS plus the number of nodes in the controllable set CS, at the time of termination. To guarantee the validity of our ranking rules in sections 6, we assume that Assumption 7.1 holds and that there are no errors in estimating $ECE^{\Delta}_{t^*}$ (in Section 6) and h is perfect heuristic according to (Pearl, 1984) (in Section 6). It is important to note that while these assumptions are necessary to provide formal guarantees, the ranking rules can still function without them.

### F.2. A* Search for Weighted Shortest Path

The A* search algorithm finds the shortest path from a start node to a goal node in a weighted graph. The description of A* search is as follows:

1. Initialize two node sets: a closed set (already evaluated nodes) and an open set (yet to be evaluated, starting with the initial node). We define a function parent_pointer : $\Phi \rightarrow \Phi$, which assigns the corresponding parent of a node $g_i$. Nodes are evaluated based on three key functions:
   - g$(g_i)$: the cost from the start node to node $g_i$,
   - h$(g_i)$: an admissible heuristic estimating the cost to the goal from $g_i$,
   - f$(g_i) = $ g$(g_i) + $ h$(g_i)$: the total estimated cost through node $g_i$.
2. While the open set is not empty:
   - Select node $g_i$ with the lowest f$(g_i)$ from the open set.
   - If $g_i$ is the goal, backtrack to construct the path.

- Move $g_i$ to the closed set and evaluate its neighbors:
  - For each neighbor $g_m$ of $g_i$, if $g_m$ is not in the closed set, add it to the open set if not already present.
  - Calculate tentative $\mathsf{g}(g_m) = \mathsf{g}(g_i) + \text{weight}(g_i, g_m)$.
  - If tentative $\mathsf{g}(g_m)$ improves upon the known $\mathsf{g}(g_m)$, update $\mathsf{g}(g_m)$, set the $\mathsf{parent\_pointer}(g_m) = g_i$, and recalculate $\mathsf{f}(g_m)$.
3. If no path to the goal is found and the open set is empty, conclude no path exists.

## F.3. Backtrack function

We define the function $\mathsf{back\_track}(g_i)$ that backtracks through the parent pointers of the node $g_i$ and returns the set of ancestors of $g_i$ during the backtracking process:

$$\mathsf{back\_track}(g_i) = \begin{cases} \{g_i\} & \text{if } \mathsf{parent\_pointer}(g_i) = \varnothing, \\ \mathsf{back\_track}(\mathsf{parent\_pointer}(g_i)) \cup \{g_i\} & \text{otherwise.} \end{cases}$$

## F.4. Admissible Heuristic

A heuristic is admissible if, for every node $g_i$, the heuristic estimate $\mathsf{h}(g_i)$ is less than or equal to the minimum cost from $g_i$ to the goal. Mathematically, this is expressed as:

$$\mathsf{h}(g_i) \leq \mathsf{h}^*(g_i),$$

where $\mathsf{h}^*(g_i)$ is the actual minimum cost from $g_i$ to the goal.

## F.5. Consistent Heuristic

This is a stronger condition than admissibility. A heuristic is consistent if, for every node $g_i$ and each of its children $g_i'$, the following condition holds:

$$\mathsf{h}(i) \leq w(g_i, g_i') + \mathsf{h}(g_i'),$$

where $w(g_i, g_i')$ is the weight from $g_i$ to $g_i'$.

## F.6. Conditions for Optimality Guarantee

If a heuristic is admissible (and we allow revisiting closed nodes) or consistent, A* search is guaranteed to find the optimal path to the goal. Using an admissible heuristic ensures that A* prioritizes paths that are potentially closer to the goal, and explores them first. If the heuristic is also consistent, it preserves the order of the nodes as they are expanded, and ensures that the first time a node $i$ is expanded (moved to closed set), it is guaranteed to have the shortest possible cost $\mathsf{g}(i)$.

## F.7. Shortest Path Ranking Rule

After the Causal Discovery step, we define the set $E$ of the children of $g_{\mathrm{sel,\,t}}$ in $\hat{\mathscr{G}}$ as follows:

$$E = \left\{ g_i \mid g_i \notin \mathrm{IS}_t, \ \ g_i \in CH_{g_{\mathrm{sel,\,t}}}^{\hat{\mathscr{G}}_t} \right\}.$$

Then, for each $g_i \in E$, we update the cost of getting from the start node (root subgoal) to it, i.e., $\mathsf{g}(g_i)$, as follows:

$$\mathsf{g}(g_i) \leftarrow \min \left( \mathsf{g}(g_i), \mathsf{g}(g_{\mathrm{sel,\,t}}) + \left| CH_{g_{\mathrm{sel,\,t}}}^{\hat{\mathscr{G}}_t} \setminus \mathsf{back\_track}(g_{\mathrm{sel,\,t}}) \right| + 1 \right),$$

where $\mathsf{back\_track}(g_{\mathrm{sel,\,t}})$ is the set of subgoals on the path from root subgoals to $g_{\mathrm{sel,\,t}}$ (see the exact definition in Appendix F.2). In the above equation, the term $\left| CH_{g_{\mathrm{sel,\,t}}}^{\hat{\mathscr{G}}_t} \setminus \mathsf{back\_track}(g_{\mathrm{sel,\,t}}) \right| + 1$, shows the expansion cost of the node $g_j$ in the context of A* algorithm. Specifically, the term $\left| CH_{g_{\mathrm{sel,\,t}}}^{\hat{\mathscr{G}}_t} \setminus \mathsf{back\_track}(g_{\mathrm{sel,\,t}}) \right|$ represents the number of new subgoals needed to be trained in Subgoal Training step and 1 denotes the number of subgoals added to the intervention set (i.e., intervening on it in Intervention Sampling).

To update $h(g_i)$, for each $g_i \in E$, we propose a dynamic version of Dijkstra's algorithm that computes the weighted shortest path from $g_i$ to $g_n$ within $\hat{\mathscr{G}}_t$. In our approach, the edge weights in $\hat{\mathscr{G}}_t$ are determined during the algorithm's execution. For each $g_i \in E$, perform the following:

1. **Initialization:** Set the tentative distance $\text{dist}(g_i) = 0$ and $\text{dist}(g) = \infty$ (the distance of $g_i$ to $g$) for all $g \in E \setminus \{g_i\}$. We define a function $\text{parent\_pointer} : \Phi \to \Phi$, which assigns the corresponding parent of a node $g_i$.
2. **Priority Queue:** Insert all nodes into a priority queue $Q$ based on their tentative distances.
3. **Algorithm Execution:**
    - While $Q$ is not empty:
        - Extract the node $g_u$ with the smallest $\text{dist}(g_u)$ from $Q$.
        - For each neighbor $g_v$ of $g_u$ in $\hat{\mathscr{G}}_t$:
            * Set $w(g_u, g_v) = |CH_{g_u}^{\hat{\mathscr{G}}_t} \setminus \text{back\_track}(g_u)| + 1$.
            * If $\text{dist}(g_v) > \text{dist}(g_u) + w(g_u, g_v)$, then: update $\text{dist}(g_v) = \text{dist}(g_u) + w(g_u, g_v)$ and $\text{parent\_pointer}(g_v) = g_u$.
4. **Heuristic Update:** After computing the shortest path, set

$$h(g_i) = \text{dist}(g_n).$$

### F.8. Intuitive Examples

In Figure 9a, to achieve the final subgoal $g_8$, all the green subgoals must be achieved and these are precisely the subgoals with a non-zero causal effect on $g_8$. Similarly in Figure 9b, we achieve the final subgoal $g_8$ if we only achieve the green subgoals during the execution of the algorithm. Under the conditions explained in Section F.1, Causal Effect and Shortest Path ranking rules will yield the minimum training cost in scenarios of Figure 9a and Figure 9b, respectively.

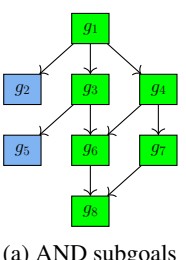

(a) AND subgoals

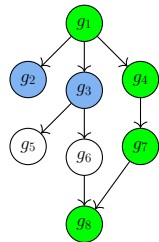

(b) OR subgoals

*Figure 9.* The causal effect and shortest path ranking rules aim to include only the green nodes to the intervention set, as shown in Figures (a) and (b), respectively.

### F.9. Hybrid Ranking Rule

In this rule, we integrate the two ranking rules described in Section 6. In Initialization step of Algorithm 1, we initialize a list *selected_subgoals* that keeps the subgoal to be selected as $g_{\text{sel}, t}$ on top of it. At each iteration, if *selected_subgoals* is not empty we pop the top of *selected_subgoals* and choose the returned subgoal as $g_{\text{sel}, t}$, otherwise, we do the following steps before popping the top of *selected_subgoals*:

(i) For each subset $S \subseteq \text{CS}_{t-1}$, we calculate the $\widehat{ECE}_{t^*}^{\Delta}(S, \text{CS}_{t-1} \setminus S, g_n)$. This measures the impact of $S$ on the final subgoal $g_n$. We keep subsets, where the corresponding $\widehat{ECE}_{t^*}^{\Delta}$ is not zero, and put them into a collection $\mathcal{S}$.

(ii) Next, for every $S \in \mathcal{S}$, we define the function $F(S)$ as follows:

$$F(S) = G(S) + H(S),$$

where:

- $G(S)$ represents the cumulative sum of the costs of all subgoals in $S$ to be added to CS. It is defined as follows:

$$G(S) = \sum_{g_j \in PA_S^{\hat{\mathscr{G}}_{t-1}}} \mathsf{g}(g_j) + \left| CH_{g_j}^{\hat{\mathscr{G}}_{t-1}} \setminus \mathrm{IS}_{t-1} \right| + 1,$$

where,

$$PA_S^{\hat{\mathscr{G}}_{t-1}} = \bigcup_{g_i \in S} PA_{g_i}^{\hat{\mathscr{G}}_{t-1}}.$$

$PA_S^{\hat{\mathscr{G}}_{t-1}}$ denotes the set of parents of the nodes in $S$ in the graph $\hat{\mathscr{G}}_{t-1}$.

- $H(S)$ is the estimated cost from $S$ to the final subgoal $g_n$. The heuristic function $H(S)$ is computed as the number of distinct nodes that appear on the paths from each $g_i \in S$ to the final subgoal $g_n$, based on the adjacency matrix of $\hat{\mathscr{G}}_{t-1}$, i.e.,

$$H(S) = \sum_{g_j \in \Phi} \mathbf{1}_{g_j \in \mathrm{path}_{\hat{\mathscr{G}}_{t-1}}(S \to g_n)},$$

where $\mathrm{path}_{\hat{\mathscr{G}}_{t-1}}(S \to g_n)$ represents all the paths from any subgoal $g_i \in S$ to $g_n$ including $g_i$ and $g_n$ based on $\hat{\mathscr{G}}_{t-1}$.

We select the subset $S$ that minimizes $F(S)$. Then, we add all the subgoals in $S$ to the *selected_subgoals* in an arbitrary order.

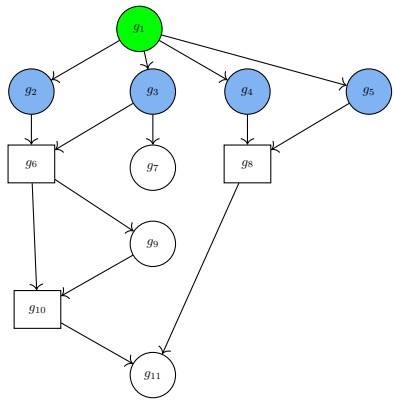

*Figure 10.* A stage of the algorithm where $g_1$ is in the intervention set and $g_2, g_3, g_4, g_5$ are controllable. Our Hybrid heuristic aims to select $g_4$ and $g_5$ as $g_{\mathrm{sel}}$ for the next steps.

For instance, in Figure 10, consider that we are in line 4 of Algorithm 1 at iteration $t$ where $\mathrm{IS}_{t-1} = \{g_1\}$ and $\mathrm{CS}_{t-1} = \{g_2, g_3, g_4, g_5\}$ are the intervention set (represented by green nodes) and controllable set (represented by blue nodes) at the end of the iteration $t-1$, respectively. Suppose that $\widehat{ECE}_{t^*}^{\Delta}$ is accurate in this example, meaning that if $\widehat{ECE}_{t^*}^{\Delta}$ is zero, the corresponding $ECE_{t^*}^{\Delta}$ is also zero, and vice versa. Now apply the steps of Hybrid heuristic: (i) We compute the estimate of the expected causal effect ($\widehat{ECE}_{t^*}^{\Delta}$) for every subset of CS. Subsets with non-zero $\widehat{ECE}_{t^*}^{\Delta}$ will be added to the collection $\mathcal{S}$. For instance, if we select subset $\{g_2, g_3\}$ from $\mathrm{CS}_{t-1}$, $\widehat{ECE}_{t^*}^{\Delta}(\{g_2, g_3\}, \{g_4, g_5\}, g_8)$ will be non-zero. Furthermore, if we select $\{g_3, g_4\}$, $\widehat{ECE}_{t^*}^{\Delta}(\{g_3, g_4\}, \{g_2, g_5\}, g_8)$ will be zero. With further evaluations, we identify three candidate sets with non-zero $\widehat{ECE}_{t^*}^{\Delta}$s: $S_1 = \{g_2, g_3\}$, $S_2 = \{g_4, g_5\}$, and $S_3 = \{g_2, g_3, g_4, g_5\}$, resulting in the collection $\mathcal{S} = \{S_1, S_2, S_3\}$. (ii) In this simple example, $G(S_1) = G(S_2) = G(S_3) = 4 + 1 = 5$. Now, we compute the $H$ function for each of these sets: $H(S_1) = |\{g_2, g_3, g_6, g_9, g_{10}\}| = 5$, $H(S_2) = |\{g_4, g_5, g_8\}| = 3$ and $H(S_3) = |\{g_2, g_3, g_6, g_9, g_{10}, g_4, g_5, g_8\}| = 8$. Therefore, $F(S_2) < F(S_1)$ and $F(S_2) < F(S_3)$, and we choose $S_2 = \{g_4, g_5\}$ for the next selections of $g_{\mathrm{sel}}$. Therefore, we add all the subgoals in $S_2$ to the *selected_subgoals*.

### F.10. Key differences with CDHRL

If HRC uses the random strategy in line 4 and applies the causal discovery method described in Ke et al. (2019) in line 7, it becomes similar to CDHRL framework in Hu et al. (2022). CDHRL does not consider the concept of $g_{\text{sel}}$ and simply appends the entire controllable set to the intervention set ($\text{IS}_t = \text{CS}_{t-1} \cup \text{CS}_{t-1}$). In contrast, we prioritize the controllable subgoals (explained in the next sections) to guide the exploration of the subgoal structure. Another difference is that CDHRL treats the action variable as a subgoal and initializes the intervention set with the action variable. In contrast, we make no such assumption and allow the intervention set to start by any root subgoal. Moreover, we design a new causal discovery algorithm with theoretical guarantees while CDHRL applies the causal discovery in Ke et al. (2019) where there is no guarantee on what it can learn.

## G. Causal Discovery

### G.1. Proof of the proposition 8.3

The SCM that satisfies Assumption 4.2 can be written as follows:

$$
X_i^{t+1} = f_i(\mathbf{X}_{EV}^t, \mathbf{X}^t, A^t, \epsilon_i^{t+1}) = \begin{cases} 1, & \text{if } X_i^t = 1, \\ 0 & \text{if } X_i^t = 0 \text{ and } \theta_i(\mathbf{X}^t) = 0, \qquad 1 \le i \le n, \\ h_i(\mathbf{X}_{EV}^t \backslash \mathbf{X}^t, A^t, \epsilon_i^{t+1}), & \text{if } X_i^t = 0 \text{ and } \theta_i(\mathbf{X}^t) = 1, \end{cases}
$$

where,

$$
\theta_i(\mathbf{X}^t) = \begin{cases} \bigwedge_{g_j \in PA_{g_i}} X_j^t, & \text{if } g_i \text{ is an AND subgoal}, \\ \bigvee_{g_j \in PA_{g_i}} X_j^t, & \text{if } g_i \text{ is an OR subgoal}, \end{cases}
$$

$h_i$ is the causal mechanism of $X_i$ when $\theta_i(\mathbf{X}_t) = 1$, and $\mathbf{X}_{EV}^t \backslash \mathbf{X}^t$ represents the vector of all the variables in $\mathcal{X}_{EV} \setminus \mathcal{X}$ at time $t$. Now, we show that we can identify the subgoal structure up to the discoverable parents.

**OR Subgoal:**

Suppose $g_i$ is an OR subgoal, and $g_j \in PA_{g_i}$ is an *undiscoverable* parent. If $g_j$ is not discoverable one of the following occurs:

1. There is no valid assignment $\mathbf{X} \in \{0, 1\}^n$ such that

$$
\forall\, g_k \in PA_{g_i}, \quad X_k = 0.
$$

   This implies that in all possible assignments, at least one parent of $g_i$ is always 1. Therefore, the OR operation can be replaced by 1:

$$
\theta_i(\mathbf{X}^t) = \bigvee_{g_j \in PA_{g_i}} X_j^t = 1.
$$

   Thus, the SCM simplifies to:

$$
X_i^{t+1} = f_i(\mathbf{X}_{EV}^t, \mathbf{X}^t, A^t, \epsilon_i^{t+1}) = h_i(\mathbf{X}_{EV}^t \backslash \mathbf{X}^t, A^t, \epsilon_i^{t+1}) \vee X_i^0.
$$

   Now consider an alternative SCM where $g_i$ has no parents and is only dependent on initial state and causal mechanism $h_i$:

$$
X_i^{t+1} = h_i(\mathbf{X}_{EV}^t \backslash \mathbf{X}^t, A^t, \epsilon_i^{t+1}) \vee X_i^0.
$$

   Therefore, the original SCM is the same as the alternative one, making them indistinguishable.

2. If there is no valid assignment $\mathbf{X} \in \{0,1\}^n$ such that

$$X_j = 1, \quad \forall g_k \in PA_{g_i} \setminus \{g_j\}, \quad X_k = 0.$$

This means that in the collected data, if $X_j$ is one, at least of some other parent $X_k$ is also one.
Now consider an alternative SCM where $g_j$ is not a parent of $g_i$:

$$\theta_i'(\mathbf{X}^t) = \bigvee_{g_k \in PA_{g_i} \setminus \{g_j\}} X_k^t.$$

Both the original SCM and the alternative have the same structural assignments on $\mathbf{X}_{EV}^t$ given the condition that $X_j$ and at least some other parent of $X_i$ are one at all time $t$. Therefore, from the collected data, we cannot distinguish between these two models and $g_j$ is not discoverable.

**AND Subgoal:**

Suppose $g_i$ is an AND subgoal, and $g_j$ is an undiscoverable parent. This means that one of the following situations occurs:

1. There is no valid assignment $\mathbf{X} \in \{0,1\}^n$ such that

$$\forall g_k \in PA_{g_i}, \quad X_k = 1.$$

Thus, the AND operator equals to 0:

$$\theta_i(\mathbf{X}^t) = \bigwedge_{g_j \in PA_{g_i}} X_j^t = 0.$$

The SCM simplifies to:

$$X_i^{t+1} = X_i^0.$$

Now consider an alternative SCM where $g_i$ is isolated and only dependent on its initial state, thus it has no parents and no dependency to $\mathbf{X}_{EV}^t \setminus \mathbf{X}^t, A^t$, and $\epsilon_i^{t+1}$ through mechanism $h_i$ (i.e., $h_i(\mathbf{X}_{EV}^t \setminus \mathbf{X}^t, A^t, \epsilon_i^{t+1}) = 0$). Hence:

$$X_i^{t+1} = X_i^0.$$

Again, both SCMs have the same structural assignment on $\mathbf{X}_{EV}^t$ given the condition that the parents of $g_i$ are not achieved at all time $t$.

2. There is no valid assignment where

$$X_j = 0, \quad \forall g_k \in PA_{g_i} \setminus \{g_j\}, \quad X_k = 1.$$

This means that if $X_j$ is zero, at least of some other parent $X_k$ is one. Now consider an alternative SCM where $g_j$ is not a parent of $g_i$:

$$\theta_i'(\mathbf{X}^t) = \bigwedge_{g_k \in PA_{g_i} \setminus \{g_j\}} X_k^t.$$

Both the original SCM and the alternative one have the same structural assignments on $\mathbf{X}_{EV}^t$ given the condition that at least some parent (other than $g_j$) is one at all time $t$.

This shows that when a parent is undiscoverable, alternative SCMs without that parent produce the same observable behavior, making them indistinguishable. This justifies the claim in Proposition 8.3 that the subgoal structure is identifiable up to discoverable parents.

### G.2. Example of Undiscoverable Parents

In this section, we provide an example to illustrate how certain edges in the subgoal structure may not be detectable.

Consider the subgoal structure depicted in Figure 11b corresponding to the time series with the one-step causal relationships shown in Figure 11a. $X_1$ and $X_2$ are OR subgoals, and $X_3$ is an AND subgoal. Both $X_1$ and $X_2$ are parents of $X_3$, and

additionally, $X_1$ is a parent of $X_2$. Subgoal $X_3$ cannot be achieved at some time step $t$ unless both $X_1$ and $X_2$ have been achieved at some prior time step $t' < t$. Moreover, to achieve $X_2$, $X_1$ must be achieved at an earlier time step. Now, if Assumption 4.2 holds, $X_1$ is always one when $X_2$ is equal to one. Therefore, we could not find two valid assignments in the observed time series satisfying the condition in Definition 8.2.

In this simple example, the following conditional independence holds in distribution $P$ over the time series data $X_i^t$:

$$(X_3^{t+1} \perp X_1^t \mid X_2^t = 1).$$

Therefore, constraint-based causal discovery methods cannot detect the edge from $X_1^t$ to $X_3^{t+1}$ in 11a

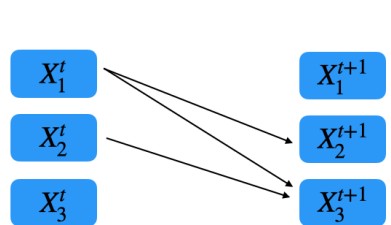

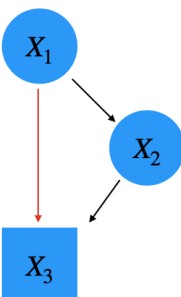

(a) The one-step causal relationships in time series.

(b) The subgoal structure.

*Figure 11.* An example of undiscoverable parent ($X_1$). The subgoal structure of the time series shows the red edge cannot be detected.

### G.3. Proof of Theorem 8.4

**Theorem G.1.** *(Formal Statement of Theorem 8.4)*

*Consider an SCM where the value of $X_i^{t+1}$ is determined as a function of the variables in the system at time $t$ and an error term $\epsilon_i^{t+1}$:*

$$X_i^{t+1} = \theta(\mathbf{X}^t) \oplus \epsilon_i^{t+1},$$

*where $g_i$ is either AND or an OR operation defined as*

$$\theta_i(\mathbf{X}^t) = \begin{cases} \bigwedge_{X_j \in PA_{X_i}} X_j^t & \text{if AND operation,} \\ \bigvee_{X_j \in PA_{X_i}} X_j^t & \text{if OR operation,} \end{cases}$$

*and $\oplus$ denotes the XOR operation. Moreover, the error term $\epsilon_i^{t+1}$ has Bernoulli distribution with parameter $\rho < 1/2$.*

*For a variable $X_i^{t+1}$ and $\boldsymbol{\beta} \in \mathbb{R}^n$, consider $S$ as:*

$$S(\mathbf{X}^t, \boldsymbol{\beta}) = \sum_j \beta_j X_j^t + \beta_0. \tag{66}$$

*Let $\hat{X}_i^{t+1} = \mathbb{1}\{S(\mathbf{X}^t, \boldsymbol{\beta}) > 0\}$ be an estimate of $X_i^{t+1}$. For any vector $\boldsymbol{\beta}$, consider the following loss function:*

$$\mathcal{L}(\boldsymbol{\beta}) = \mathbb{E}[(\hat{X}_i^{t+1} - X_i^{t+1})^2] + \lambda\|\boldsymbol{\beta}\|_0. \tag{67}$$

*There exists a $\lambda > 0$ such that for any optimal solution $\boldsymbol{\beta}^*$ minimizing the loss function in (67), the positive coefficients in $\boldsymbol{\beta}^*$ correspond to the parents of $X_i^{t+1}$ in $\mathbf{X}^t$.*

*Proof of Theorem G.1:* In the proof, for ease of representation, we denote $X_i^{t+1}$ by $y$ and drop superscript $t$ in $\mathbf{X}^t$ and $t+1$ in $\epsilon_i^{t+1}$. Also, we might use the notation of environment variable and its associated subgoal interchangeably. Moreover, for any given $\boldsymbol{\beta}$, we also denote $\hat{X}_i^{t+1}$ by $\hat{y}_{\boldsymbol{\beta}}$.

Let $PA_y$ be the parent set of $y$. We define $P_y = \{j | X_j \in PA_y\}$.

**Proof for OR subgoal:** We show that any optimal solution $\boldsymbol{\beta}^*$ should satisfy the following conditions:

1. For $j \notin P_y$, $\beta_j^* = 0$,
2. For $j \in P_y$, $\beta_j^* > 0$.

Let us rewrite the loss function in (67) for any $\boldsymbol{\beta}$ satisfying the above conditions as follows:

$$\mathcal{L}(\boldsymbol{\beta}) = \mathbb{E}[(y - \hat{y}_{\boldsymbol{\beta}})^2] + \lambda \|\boldsymbol{\beta}\|_0, \tag{68}$$

$$\overset{(a)}{=} \sum_{\mathbf{X} \in \{0,1\}^n} P(\mathbf{X}) \mathbb{E}[(y - \hat{y}_{\boldsymbol{\beta}})^2 | \mathbf{X}] + \lambda |P_y| \tag{69}$$

$$\overset{(b)}{=} \sum_{\mathbf{X} \in \{0,1\}^n} P(\mathbf{X}) \rho + \lambda |P_y| \tag{70}$$

$$= \rho + \lambda |P_y|. \tag{71}$$

$(a)$ We marginalize over the set of all possible $\mathbf{X}$s. $|P_y|$ denotes the cardinality of the set $P_y$.
$(b)$ $|y - \hat{y}_{\boldsymbol{\beta}}|$ is equal to one if $\epsilon_i = 1$ which occurs with probability $\rho$. Otherwise, there is no error in prediction.

Now, we show that if any vector $\boldsymbol{\beta}$ fails to satisfy either of the aforementioned conditions, it will result in an increase in loss compared to 71. To prove this, we consider the following three cases: 1- $\boldsymbol{\beta}$ does not satisfy condition 1 but it satisfies condition 2. 2- $\boldsymbol{\beta}$ satisfies condition 1 but does not condition 2. 3- $\boldsymbol{\beta}$ does not satisfy both conditions 1 and 2.

Case 1: As $\boldsymbol{\beta}$ does not satisfy condition 1, the non-parent variables might also have non-zero coefficients in $\boldsymbol{\beta}$. We form the vector $\boldsymbol{\beta}^*(\boldsymbol{\beta})$ by setting zero to the entries of $\boldsymbol{\beta}$ corresponding to the non-parent variables. Moreover, we define $A_y$ as a set of indices of non-parent variables that have non-zero coefficients in $\boldsymbol{\beta}$, i.e., $A_y = \{j | j \notin P_y, \beta_j \neq 0\}$.

Now we compute the difference in the losses for $\boldsymbol{\beta}$ and $\boldsymbol{\beta}^*(\boldsymbol{\beta})$:

$$\mathcal{L}(\boldsymbol{\beta}) - \mathcal{L}(\boldsymbol{\beta}^*(\boldsymbol{\beta})) = \left[ \mathbb{E}[(y - \hat{y}_{\boldsymbol{\beta}})^2] - \mathbb{E}[(y - \hat{y}_{\boldsymbol{\beta}^*(\boldsymbol{\beta})})^2] \right] + \lambda(\|\boldsymbol{\beta}\|_0 - \|\boldsymbol{\beta}^*(\boldsymbol{\beta})\|_0), \tag{72}$$

$$= \left[ \mathbb{E}[(y - \hat{y}_{\boldsymbol{\beta}})^2] - \rho \right] + \lambda(\|\boldsymbol{\beta}\|_0 - |P_y|), \tag{73}$$

$$\overset{(a)}{\geq} \lambda(\|\boldsymbol{\beta}\|_0 - |P_y|), \tag{74}$$

$$\overset{(b)}{=} \lambda(|P_y| + |A_y| - |P_y|), \tag{75}$$

$$= \lambda |A_y| \overset{(c)}{>} 0. \tag{76}$$

$(a)$ It is due to the fact that for any $\boldsymbol{\beta} \in \mathbb{R}^n$, $\mathbb{E}[(y - \hat{y}_{\boldsymbol{\beta}})^2] \geq \rho$.
$(b)$ As condition 2 holds $\|\boldsymbol{\beta}\|_0 = |P_y| + |A_y|$.
$(c)$ We assume that $|A_y| > 0$.

This indicates that $\boldsymbol{\beta}$ has a higher cost compared with $\boldsymbol{\beta}^*(\boldsymbol{\beta})$ and it cannot be an optimal solution.

Case 2: Now consider $\boldsymbol{\beta}$ fails to satisfy condition 2 (but it satisfies condition 1). This implies that there exists $j$ belonging to the parent node set $P_y$ such that its corresponding coefficient is less than or equal to 0. In particular, we define the set $M_y = \{j | j \in P_y, \beta_j \leq 0\}$. Please note that we also suppose that $|M_y| > 0$. We form the vector $\boldsymbol{\beta}^*(\boldsymbol{\beta})$ by setting the entries of $\boldsymbol{\beta}$ corresponding to the set $M_y$ to a value greater than zero. Moreover, we define $Z_y$, $D_{M_y}$, and $D_{Z_y}$ as follows:

- $Z_y = \{j | j \in P_y, \beta_j = 0\}$
- $D_{M_y} = \{\mathbf{X} \in \{0,1\}^n | \exists j \in M_y, X_j = 1 \text{ and } X_i = 0 \text{ for all } i \in P_y \text{ s.t. } i \neq j\}$,

where $D_{M_y}$ corresponds to the samples, where only one parent in $M_y$ is equal to one and other parents in the system are zero. Let us compare the loss function for $\boldsymbol{\beta}$ and $\boldsymbol{\beta}^*(\boldsymbol{\beta})$:

$$\mathcal{L}(\boldsymbol{\beta}) - \mathcal{L}(\boldsymbol{\beta}^*(\boldsymbol{\beta})) = \left[ \mathbb{E}[(y - \hat{y}_{\boldsymbol{\beta}})^2] - \mathbb{E}[(y - \hat{y}_{\boldsymbol{\beta}^*(\boldsymbol{\beta})})^2] \right] + \lambda(\|\boldsymbol{\beta}\|_0 - \|\boldsymbol{\beta}^*(\boldsymbol{\beta})\|_0), \tag{77}$$

$$= \sum_{\mathbf{X} \in D_{M_y}} P(\mathbf{X}) \left[ \mathbb{E}[(y - \mathbb{1}_{S(\mathbf{X},\boldsymbol{\beta})})^2 | \mathbf{X}] - \mathbb{E}[(y - \mathbb{1}_{S(\mathbf{X},\boldsymbol{\beta}^*(\boldsymbol{\beta}))})^2 | \mathbf{X}] \right] \tag{78}$$

$$+ \sum_{\mathbf{X} \notin D_{M_y}} P(\mathbf{X}) \left[ \mathbb{E}[(y - \mathbb{1}_{S(\mathbf{X},\boldsymbol{\beta})})^2 | \mathbf{X}] - \mathbb{E}[(y - \mathbb{1}_{S(\mathbf{X},\boldsymbol{\beta}^*(\boldsymbol{\beta}))})^2 | \mathbf{X}] \right] \tag{79}$$

$$+ \lambda(\|\boldsymbol{\beta}\|_0 - \|\boldsymbol{\beta}^*(\boldsymbol{\beta})\|_0), \tag{80}$$

$$\overset{(a)}{\geq} \sum_{\mathbf{X} \in D_{M_y}} P(\mathbf{X})(1 - 2\rho) + \lambda(\|\boldsymbol{\beta}\|_0 - \|\boldsymbol{\beta}^*(\boldsymbol{\beta})\|_0) \tag{81}$$

$$\overset{(b)}{=} \sum_{\mathbf{X} \in D_{M_y}} P(\mathbf{X})(1 - 2\rho) - \lambda |Z_y| > 0, \tag{82}$$

$(a)$ For $\mathbf{X} \in D_{M_y}$, $S(\mathbf{X}, \boldsymbol{\beta}) < S(\mathbf{X}, \boldsymbol{\beta}^*(\boldsymbol{\beta}))$ and $\hat{y}_{\boldsymbol{\beta}}$ is set to 0 while $\hat{y}_{\boldsymbol{\beta}^*(\boldsymbol{\beta})}$ is set to 1 since we assigned values greater than zero for the coefficients in $\boldsymbol{\beta}^*(\boldsymbol{\beta})$ corresponding to parent nodes. Thus, for any $\mathbf{X} \in D_{M_y}$, the model with coefficient vectors $\boldsymbol{\beta}$ and $\boldsymbol{\beta}^*(\boldsymbol{\beta})$ predicts $y$ correctly with probability $1 - \rho$ and $\rho$, respectively. Consequently,

$$\sum_{\mathbf{X} \in D_{M_y}} P(\mathbf{X}) \left[ \mathbb{E}[(y - \mathbb{1}_{S(\mathbf{X},\boldsymbol{\beta})})^2 | \mathbf{X}] - \mathbb{E}[(y - \mathbb{1}_{S(\mathbf{X},\boldsymbol{\beta}^*(\boldsymbol{\beta}))})^2 | \mathbf{X}] \right] \tag{83}$$

$$= \sum_{\mathbf{X} \in D_{M_y}} P(\mathbf{X}) \left[ (1 - \rho) - \rho \right] \tag{84}$$

$$= \sum_{\mathbf{X} \in D_{M_y}} P(\mathbf{X})(1 - 2\rho). \tag{85}$$

Moreover, the term

$$\sum_{\mathbf{X} \notin D_{M_y}} P(\mathbf{X}) \left[ \mathbb{E}[(y - \mathbb{1}_{S(\mathbf{X},\boldsymbol{\beta})})^2 | \mathbf{X}] - \mathbb{E}[(y - \mathbb{1}_{S(\mathbf{X},\boldsymbol{\beta}^*(\boldsymbol{\beta}))})^2 | \mathbf{X}] \right] \geq 0,$$

since $\mathbb{E}[(y - \mathbb{1}_{S(\mathbf{X},\boldsymbol{\beta}^*(\boldsymbol{\beta}))})^2 | \mathbf{X}] = \rho$ and for any $\boldsymbol{\beta} \in \mathbb{R}^n$, $\mathbb{E}[(y - \mathbb{1}_{S(\mathbf{X},\boldsymbol{\beta})})^2 | \mathbf{X}] \geq \rho$.

$(b)$ There are $|Z_y|$ more nonzero terms in $\boldsymbol{\beta}^*(\boldsymbol{\beta})$ compared with $\boldsymbol{\beta}$. Moreover, $\sum_{\mathbf{X} \in D_{M_y}} P(\mathbf{X})(1 - 2\rho) \geq |D_{M_y}| \times \min_{\mathbf{X} \in D_{M_y}} [P(\mathbf{X})(1-2\rho)] \geq |Z_y| \times \min_{\mathbf{X} \in D_{M_y}} [P(\mathbf{X})(1-2\rho)]$. Therefore, for $\lambda < \min_{\mathbf{X} \in D_{M_y}} [P(\mathbf{X})(1-2\rho)]$, the loss of $\boldsymbol{\beta}$ is greater than $\boldsymbol{\beta}^*(\boldsymbol{\beta})$.

Case 3: In this case, $\boldsymbol{\beta}$ fails to satisfy both conditions 1 and 2. This implies that some non-parent variables have non-zero coefficients in $\boldsymbol{\beta}$. Moreover, $\beta_0$ is not necessarily equal to zero, and for any $j$ belonging to the parent node set $P_y$, their corresponding coefficient can be less than or equal to 0. In particular, first, we form the vector $\boldsymbol{\beta}'(\boldsymbol{\beta})$ by setting the entries of $\boldsymbol{\beta}$ corresponding to the set $M_y$ to a value bigger than zero, and second, we form $\boldsymbol{\beta}^*(\boldsymbol{\beta})$ by setting zero the entries of $\boldsymbol{\beta}'(\boldsymbol{\beta})$ corresponding to the non-parent variables. The reasoning about the loss change $\mathcal{L}(\boldsymbol{\beta}) - \mathcal{L}(\boldsymbol{\beta}'(\boldsymbol{\beta}))$ is the same as case 2, except that for $\mathbf{X} \notin D_{M_y}$, one can claim that having $S(\mathbf{X}, \boldsymbol{\beta}) < S(\mathbf{X}, \boldsymbol{\beta}'(\boldsymbol{\beta}))$ might lead to a decrease in the $\mathbb{E}[(y - \hat{y}_{\boldsymbol{\beta}})^2 | \mathbf{X}]$. This means that for a given $\mathbf{X}$, $\hat{y}_{\boldsymbol{\beta}'(\boldsymbol{\beta})}$ is wrongly set to 1 while $\hat{y}_{\boldsymbol{\beta}}$ is correctly set to zero resulting in a lower squared loss in $\mathbb{E}[(y - \hat{y}_{\boldsymbol{\beta}})^2 | \mathbf{X}]$ compared to $\mathbb{E}[(y - \hat{y}_{\boldsymbol{\beta}'(\boldsymbol{\beta})})^2 | \mathbf{X}]$. Since the actual class of $y$ is zero, then all $X_j \in PA_y$ must be equal to 0. Hence, changing the corresponding coefficients of the parents would not affect the value of $S$. This implies that for this specific $\mathbf{X}$, $S(\mathbf{X}, \boldsymbol{\beta}) = S(\mathbf{X}, \boldsymbol{\beta}'(\boldsymbol{\beta}))$, which contradicts the given assumption that $S(\mathbf{X}, \boldsymbol{\beta}) < S(\mathbf{X}, \boldsymbol{\beta}'(\boldsymbol{\beta}))$. Therefore,

$$\sum_{\mathbf{X} \notin D_{M_y}} P(\mathbf{X}) \left[ \mathbb{E}[(y - \mathbb{1}_{S(\mathbf{X},\boldsymbol{\beta})})^2 | \mathbf{X}] - \mathbb{E}[(y - \mathbb{1}_{S(\mathbf{X},\boldsymbol{\beta}'(\boldsymbol{\beta}))})^2 | \mathbf{X}] \right] \geq 0.$$

The reasoning about the change in loss, $\mathcal{L}(\boldsymbol{\beta}'(\boldsymbol{\beta})) - \mathcal{L}(\boldsymbol{\beta}^*(\boldsymbol{\beta}))$, is the same as case 1. Therefore, the total loss change would be as follows:

$$\mathcal{L}(\boldsymbol{\beta}) - \mathcal{L}(\boldsymbol{\beta}^*(\boldsymbol{\beta})) = \left[ \mathbb{E}[(y - \hat{y}_{\boldsymbol{\beta}})^2] - \mathbb{E}[(y - \hat{y}_{\boldsymbol{\beta}^*(\boldsymbol{\beta})})^2] \right] + \lambda(\|\boldsymbol{\beta}\|_0 - \|\boldsymbol{\beta}^*(\boldsymbol{\beta})\|_0), \tag{86}$$

$$= \sum_{\mathbf{X} \in D_{M_y}} P(\mathbf{X}) \left[ \mathbb{E}[(y - \mathbb{1}_{S(\mathbf{X},\boldsymbol{\beta})})^2 | \mathbf{X}] - \mathbb{E}[(y - \mathbb{1}_{S(\mathbf{X},\boldsymbol{\beta}^*(\boldsymbol{\beta}))})^2 | \mathbf{X}] \right] \tag{87}$$

$$+ \sum_{\mathbf{X} \notin D_{M_y}} P(\mathbf{X}) \left[ \mathbb{E}[(y - \mathbb{1}_{S(\mathbf{X},\boldsymbol{\beta})})^2 | \mathbf{X}] - \mathbb{E}[(y - \mathbb{1}_{S(\mathbf{X},\boldsymbol{\beta}^*(\boldsymbol{\beta}))})^2 | \mathbf{X}] \right] \tag{88}$$

$$+ \lambda(\|\boldsymbol{\beta}\|_0 - \|\boldsymbol{\beta}^*(\boldsymbol{\beta})\|_0), \tag{89}$$

$$\geq \sum_{\mathbf{X} \in D_{M_y}} P(\mathbf{X})(1 - 2\rho) + \lambda(|P_y| - |Z_y| + |A_y| - |P_y|), \tag{90}$$

$$= \sum_{\mathbf{X} \in D_{M_y}} P(\mathbf{X})(1 - 2\rho) + \lambda(-|Z_y| + |A_y|). \tag{91}$$

Similar to case 2, we select $\lambda < \min_{\mathbf{X} \in D_{M_y}}[P(\mathbf{X})(1 - 2\rho)]$ to ensure that the calculated difference remains positive. It is important to note that the term $P(\mathbf{X})(1 - 2\rho)$ may not be applicable to all $\mathbf{X}$ in $D_{M_y}$. Specifically, for those $\mathbf{X}$ where $\sum_{j \notin P_y} \beta_j X_j$ is positive within the score $S(\mathbf{X}, \boldsymbol{\beta})$. However, the cost $P(\mathbf{X}')(1 - 2\rho)$ applies to the sample $\mathbf{X}'$, which is identical to $\mathbf{X}$ except that the corresponding parent value is 0. Therefore, we can conclude that any optimal solution to the loss function given in (67) must satisfy both conditions 1 and 2, thereby completing the proof.

**Proof for AND subgoal:** We show that any optimal solution $\boldsymbol{\beta}^*$ should satisfy the following conditions (we consider that $\beta_0 < 0$):

1. For $j \notin P_y, \beta_j^* = 0$.
2. $\forall U \subsetneq P_y$ :

$$\sum_{j \in U} \beta_j^* \leq -\beta_0 \tag{92}$$

$$\sum_{j \in P_y} \beta_j^* > -\beta_0, \tag{93}$$

The value of the loss function in (67) for any $\boldsymbol{\beta}$ satisfying the above conditions is equal to 71. Now, we show that if some $\boldsymbol{\beta}$ fails to satisfy either of the aforementioned conditions, it will result in an increase in loss compared to 71. To prove this, we consider the following three cases: 1- $\boldsymbol{\beta}$ does not satisfy condition 1 but it satisfies condition 2. 2- $\boldsymbol{\beta}$ satisfies condition 1 but does not condition 2. 3- $\boldsymbol{\beta}$ does not satisfy both conditions 1 and 2.

1- This part is similar to OR subgoal, case 1.

2- Consider $\boldsymbol{\beta}$ satisfies condition 1 but fails to satisfy condition 2. In this case, we first show that if there exists some $j$ belonging to the parent node set $P_y$ such that its corresponding coefficient is less than or equal to 0, at least one of the conditions in (92) or (93) will be violated which in turn results in a higher loss. To prove this, we define the set $M_y = \{j | j \in P_y, \beta_j \leq 0\}$ and $Z_y = \{j | j \in P_y, \beta_j = 0\}$ and form the vector $\boldsymbol{\beta}^*(\boldsymbol{\beta})$ by setting the entries of $\boldsymbol{\beta}$ corresponding to the set $M_y$ to a value greater than zero such that both (92) and (93) are satisfied. Note that this operation is feasible since we can adjust the coefficients corresponding to non-parent nodes and parent nodes independently. Now we show that the $\mathcal{L}(\boldsymbol{\beta})$ is greater than or equal to $\mathcal{L}(\boldsymbol{\beta}^*(\boldsymbol{\beta}))$. We can define the following sample set:

$$D_{M_y} = \{\mathbf{X}_{j,0} \mid j \in M_y\},$$

where $[\mathbf{X}_{j,0}]_i = \begin{cases} 1 & \text{if } i \in P_y \text{ and } i \neq j, \\ 0 & \text{if } i = j \end{cases}$. We also define the vector $\mathbf{X}_1$ where $[\mathbf{X}_1]_i = 1$ for $i \in P_y$. We can compute the difference in the losses for $\boldsymbol{\beta}$ and $\boldsymbol{\beta}^*(\boldsymbol{\beta})$ as follows:

$$\mathcal{L}(\boldsymbol{\beta}) - \mathcal{L}(\boldsymbol{\beta}^*(\boldsymbol{\beta})) = \left[\mathbb{E}[(y - \hat{y}_{\boldsymbol{\beta}})^2] - \mathbb{E}[(y - \hat{y}_{\boldsymbol{\beta}^*(\boldsymbol{\beta})})^2]\right] + \lambda(\|\boldsymbol{\beta}\|_0 - \|\boldsymbol{\beta}^*(\boldsymbol{\beta})\|_0), \tag{94}$$

$$= \sum_{\mathbf{X} \in D_{M_y} \cup \{\mathbf{X}_1\}} P(\mathbf{X}) \left[\mathbb{E}[(y - \mathbb{1}_{S(\mathbf{X},\boldsymbol{\beta})})^2|\mathbf{X}] - \mathbb{E}[(y - \mathbb{1}_{S(\mathbf{X},\boldsymbol{\beta}^*(\boldsymbol{\beta}))})^2|\mathbf{X}]\right]$$

$$+ \sum_{\mathbf{X} \notin D_{M_y} \cup \{\mathbf{X}_1\}} P(\mathbf{X}) \left[\mathbb{E}[(y - \mathbb{1}_{S(\mathbf{X},\boldsymbol{\beta})})^2|\mathbf{X}] - \mathbb{E}[(y - \mathbb{1}_{S(\mathbf{X},\boldsymbol{\beta}^*(\boldsymbol{\beta}))})^2|\mathbf{X}]\right]$$

$$+ \lambda(\|\boldsymbol{\beta}\|_0 - \|\boldsymbol{\beta}^*(\boldsymbol{\beta})\|_0). \tag{95}$$

Now, we derive a lower bound on (95). It is clear that $S(\mathbf{X}_1, \boldsymbol{\beta}) \leq S(\mathbf{X}_{j_0}, \boldsymbol{\beta})$. We show that at least one of the terms corresponding to $\mathbf{X}_{j_0}$ or $\mathbf{X}_1$ results in an increase in the loss compared to the one of $\boldsymbol{\beta}^*(\boldsymbol{\beta})$. In particular, if $S(\mathbf{X}_{j_0}, \boldsymbol{\beta}) \leq 0$, then $S(\mathbf{X}_1, \boldsymbol{\beta}) \leq 0$ and as a result, $\hat{y}_{\boldsymbol{\beta}}$ is set to 0 which yields the expected loss of $1 - \rho$ given observing the sample $\mathbf{X}_1$. Moreover, if $S(\mathbf{X}_{j_0}, \boldsymbol{\beta}) > 0$, then $\hat{y}_{\boldsymbol{\beta}}$ is set to 1 which also yields the expected loss of $1 - \rho$ given the sample $\mathbf{X}_{j_0}$. Thus, we have an increase of either $P(\mathbf{X}_{j_0})(1 - 2\rho)$ or $P(\mathbf{X}_1)(1 - 2\rho)$ in the loss compared with $\mathcal{L}(\boldsymbol{\beta}^*(\boldsymbol{\beta}))$. Therefore there exists a sample in $D_{M_y} \cup \{\mathbf{X}_1\}$ for which $\left[\mathbb{E}[(y - \mathbb{1}_{S(\mathbf{X},\boldsymbol{\beta})})^2] - \mathbb{E}[(y - \mathbb{1}_{S(\mathbf{X},\boldsymbol{\beta}^*(\boldsymbol{\beta}))})^2]\right]$ is $(1 - 2\rho)$ and we denote it by $\tilde{\mathbf{X}}$.

Note that

$$\sum_{\mathbf{X} \in D_{M_y} \cup \{\mathbf{X}_1\}} P(\mathbf{X}) \left[\mathbb{E}[(y - \mathbb{1}_{S(\mathbf{X},\boldsymbol{\beta})})^2] - \mathbb{E}[(y - \mathbb{1}_{S(\mathbf{X},\boldsymbol{\beta}^*(\boldsymbol{\beta}))})^2]\right] \geq 0,$$

since $\mathbb{E}[(y - \mathbb{1}_{S(\mathbf{X},\boldsymbol{\beta}^*(\boldsymbol{\beta}))})^2] = \rho$ and $\mathbb{E}[(y - \mathbb{1}_{S(\mathbf{X},\boldsymbol{\beta})})^2]$ cannot be less than $\rho$. Hence, (95) can be lower bounded as follows:

$$\mathcal{L}(\boldsymbol{\beta}) - \mathcal{L}(\boldsymbol{\beta}^*(\boldsymbol{\beta})) = \left[\mathbb{E}[(y - \hat{y}_{\boldsymbol{\beta}})^2] - \mathbb{E}[(y - \hat{y}_{\boldsymbol{\beta}^*(\boldsymbol{\beta})})^2]\right] + \lambda(\|\boldsymbol{\beta}\|_0 - \|\boldsymbol{\beta}^*(\boldsymbol{\beta})\|_0), \tag{96}$$

$$= \sum_{\mathbf{X} \in D_{M_y} \cup \{\mathbf{X}_1\}} P(\mathbf{X}) \left[\mathbb{E}[(y - \mathbb{1}_{S(\mathbf{X},\boldsymbol{\beta})})^2|\mathbf{X}] - \mathbb{E}[(y - \mathbb{1}_{S(\mathbf{X},\boldsymbol{\beta}^*(\boldsymbol{\beta}))})^2|\mathbf{X}]\right] \tag{97}$$

$$+ \sum_{\mathbf{X} \notin D_{M_y} \cup \{\mathbf{X}_1\}} P(\mathbf{X}) \left[\mathbb{E}[(y - \mathbb{1}_{S(\mathbf{X},\boldsymbol{\beta})})^2|\mathbf{X}] - \mathbb{E}[(y - \mathbb{1}_{S(\mathbf{X},\boldsymbol{\beta}^*(\boldsymbol{\beta}))})^2|\mathbf{X}]\right] \tag{98}$$

$$+ \lambda(\|\boldsymbol{\beta}\|_0 - \|\boldsymbol{\beta}^*(\boldsymbol{\beta})\|_0), \tag{99}$$

$$\geq P(\tilde{\mathbf{X}})(1 - 2\rho) + \lambda(\|\boldsymbol{\beta}\|_0 - \|\boldsymbol{\beta}^*(\boldsymbol{\beta})\|_0) \tag{100}$$

$$\overset{(i)}{=} P(\tilde{\mathbf{X}})(1 - 2\rho) - \lambda|Z_y| > 0, \tag{101}$$

where $(i)$ is due to the fact that there are $|Z_y|$ additional nonzero entries in $\boldsymbol{\beta}^*(\boldsymbol{\beta})$ compared to $\boldsymbol{\beta}$. To ensure that the last inequality holds, we need to choose a value for $\lambda$ such that it is less than $\frac{1}{|Z_y|}P(\tilde{\mathbf{X}})(1 - 2\rho)$. We observe that:

$$\min\left[\min_{j \in P_y}(P(\mathbf{X}_{j_0})(1 - 2\rho)), P(\mathbf{X}_1)(1 - 2\rho)\right] \leq P(\tilde{\mathbf{X}})(1 - 2\rho),$$

and

$$|Z_y| < n.$$

Which result in:

$$\frac{1}{n}\min[\min_{j \in P_y}(P(\mathbf{X}_{j_0})(1 - 2\rho)), P(\mathbf{X}_1)(1 - 2\rho)] \leq \frac{1}{|Z_y|}P(\tilde{\mathbf{X}})(1 - 2\rho).$$

Therefore, if we choose $\lambda$ such that $\lambda < \frac{1}{n}\min[\min_{j \in P_y}(P(\mathbf{X}_{j_0})(1 - 2\rho)), P(\mathbf{X}_1)(1 - 2\rho)]$, we guarantee that the loss of $\boldsymbol{\beta}$ is greater than $\boldsymbol{\beta}^*(\boldsymbol{\beta})$.

Now we study the cases where $\beta_j > 0$ for all $j \in P_y$, yet (92) or (93) are not satisfied. In the following, we will cover these cases and show that there is an increase in loss. It is noteworthy that in these cases, there is no change in the regularization term $\|\boldsymbol{\beta}\|_0$.

- (a) Consider the case,

$$\exists U \subsetneq P_y, \sum_{j \in U} \beta_j > -\beta_0 \tag{102}$$

$$\sum_{j \in P_y} \beta_j > -\beta_0. \tag{103}$$

In this case, if we have positivity assumption, there exists a sample such as $\tilde{\mathbf{X}}$ where $[\tilde{\mathbf{X}}]_i = \begin{cases} 1 & \text{if } i \in U \\ 0 & \text{if } i \notin U \end{cases}$ which

wrongly be classified as 1 and the loss will increase at least $P(\tilde{\mathbf{X}})(1 - 2\rho)$ compared to the $\boldsymbol{\beta}^*(\boldsymbol{\beta})$.
- (b) Consider the case,

$$\forall U \subsetneq P_y, \sum_{j \in U} \beta_j \leq -\beta_0 \tag{104}$$

$$\sum_{j \in P_y} \beta_j \leq -\beta_0. \tag{105}$$

In this case, if we have positivity assumption, there exists a sample such as $\tilde{\mathbf{X}}$ where $[\tilde{\mathbf{X}}]_i = \begin{cases} 1 & \text{if } i \in P_y \\ 0 & \text{if } i \notin P_y \end{cases}$, which

will wrongly be classified as 0 and the loss will increase at least $P(\tilde{\mathbf{X}})(1 - 2\rho)$ compared to the $\boldsymbol{\beta}^*(\boldsymbol{\beta})$.
- (c) Consider the case,

$$\exists U \subsetneq P_y, \sum_{j \in U} \beta_j > -\beta_0 \tag{106}$$

$$\sum_{j \in P_y} \beta_j \leq -\beta_0. \tag{107}$$

In this case, both arguments made in (a) and (b) can be used to show that there is an increase in loss (at least $P(\tilde{\mathbf{X}})(1 - 2\rho)$) compared to the $\boldsymbol{\beta}^*(\boldsymbol{\beta})$.

3- In this case, $\boldsymbol{\beta}$ fails to satisfy both conditions 1 and 2. Similar to the OR subgoals, we form the vector $\boldsymbol{\beta}^*(\boldsymbol{\beta})$ by setting the entries of $\boldsymbol{\beta}$ corresponding to the set $M_y$ to a value bigger than zero such that condition 2 is satisfied. Moreover, we set the entries corresponding to the non-parent variables to zero. The difference in the loss for $\boldsymbol{\beta}$ and $\boldsymbol{\beta}^*(\boldsymbol{\beta})$ is as follows:

$$\mathcal{L}(\boldsymbol{\beta}) - \mathcal{L}(\boldsymbol{\beta}^*(\boldsymbol{\beta})) = \left[ \mathbb{E}[(y - \hat{y}_{\boldsymbol{\beta}})^2] - \mathbb{E}[(y - \hat{y}_{\boldsymbol{\beta}^*(\boldsymbol{\beta})})^2] \right] + \lambda(\|\boldsymbol{\beta}\|_0 - \|\boldsymbol{\beta}^*(\boldsymbol{\beta})\|_0), \tag{108}$$

$$= \sum_{\mathbf{X} \in D_{M_y} \cup \{\mathbf{X}_1\}} P(\mathbf{X}) \left[ \mathbb{E}[(y - \mathbb{1}_{S(\mathbf{X}, \boldsymbol{\beta})})^2 | \mathbf{X}] - \mathbb{E}[(y - \mathbb{1}_{S(\mathbf{X}, \boldsymbol{\beta}^*(\boldsymbol{\beta}))})^2 | \mathbf{X}] \right] \tag{109}$$

$$+ \sum_{\mathbf{X} \notin D_{M_y} \cup \{\mathbf{X}_1\}} P(\mathbf{X}) \left[ \mathbb{E}[(y - \mathbb{1}_{S(\mathbf{X}, \boldsymbol{\beta})})^2 | \mathbf{X}] - \mathbb{E}[(y - \mathbb{1}_{S(\mathbf{X}, \boldsymbol{\beta}^*(\boldsymbol{\beta}))})^2 | \mathbf{X}] \right] \tag{110}$$

$$+ \lambda(\|\boldsymbol{\beta}\|_0 - \|\boldsymbol{\beta}^*(\boldsymbol{\beta})\|_0), \tag{111}$$

$$\geq P(\tilde{\mathbf{X}})(1 - 2\rho) + \lambda(\|\boldsymbol{\beta}\|_0 - \|\boldsymbol{\beta}^*(\boldsymbol{\beta})\|_0) \tag{112}$$

$$= P(\tilde{\mathbf{X}})(1 - 2\rho) + \lambda(|A_y| - |Z_y|) > 0. \tag{113}$$

with a similar argument, $|A_y|$ appears in cases 1 and $\tilde{\mathbf{X}}$ and $\lambda$ are chosen based on the arguments made in case 2. Therefore, the last inequality is greater than zero, implying that any optimal solution of the loss function (67) should satisfy both conditions 1 and 2, and the proof is complete.

# H. Details of Experiments

## H.1. Experiments setup for synthetic dataset

Figures 5(a), 5(b), and 5(c) illustrate the costs for semi-Erdős–Rényi graph $G(n, p)$ for different values of $c$. For each semi-Erdős–Rényi graph, we randomly choose a node (subgoal) to be either AND or OR. For each node size illustrated, we generated 100 graphs, and for each graph, we conducted 10 trials of the methods. Each point in the plots represents the mean cost of these runs. Figure 5(d) illustrates the cost of $\text{HRC}_\text{c}$ for a tree $G(n, b)$ where the branching factor is set to 3. Each point in the plot of Figure 5(d) shows the logarithm of the mean cost over 100 runs on the generated graph. In our experiments with synthetic data, we considered a relaxed version of Assumption 7.1 with a specific form of error ($\frac{1}{1+t}$) in recovering new edges where $t$ is the iteration number. We considered a 95% confidence interval for all the experiments.

## H.2. Experiments setup for Minecraft

CDHRL is the most relevant method, proposed by Hu et al. (2022), which utilizes the causal discovery method introduced in Ke et al. (2019). Hierarchical Actor-Critic (HAC) is a well-known hierarchical reinforcement learning (HRL) framework (Levy et al., 2017) that enables hierarchical agents to jointly learn a hierarchy of policies, thereby significantly accelerating learning. However, compared to our targeted strategy, HAC exhibits poorer performance. Similar to the setup in Hu et al. (2022), Oracle HRL (OHRL) is implemented as a two-level Deep Q-Network (DQN) with Hindsight Experience Replay (HER) introduced in Andrychowicz et al. (2017) and an "oracle goal space". In order to imitate the oracle goal space, we manually remove a subset of $\Phi$ that is useless in achieving the final subgoal. We conducted experiments in Minecraft using a two-level Deep Q-Network (DQN) with Hindsight Experience Replay (HER) (Andrychowicz et al., 2017). HER allows sample-efficient learning by reusing the unsuccessful trajectories to achieve the goal by relabeling goals in the previous trajectory with different goals. Additionally, we experimented in Minecraft using VACERL, introduced in Nguyen et al. (2024), by using their publicly available code. However, in our experiments, we were unable to train this method successfully. The success rate and reward remained consistently at zero, similar to the PPO curve in Figure 6 even after 4 million system probes. Due to its poor performance, we excluded it from Figure 6. There are other methods in the literature that showed poorer performance compared to those we included (see CDHRL in Hu et al. (2022)) in our comparison; therefore, we did not include them in our experiments.

## H.3. Expected causal effect

Consider an estimate of the causal model. This model takes the input $\mathbf{X}$ and returns $\mathbf{X}$ at the next time step. In practice, to calculate an estimate of $\mathbb{E}[X_n^{t^*+\Delta} \mid \text{do}(X_A^{t^*} = \alpha), \text{do}(X_B^{t^*} = 0)]$ in Definition 3.4, we set $t^* = 0$ and $\Delta = 20$. We start by initializing $\mathbf{X}$ where the $X_i$s associated with the subgoals in $A$ are set to $\alpha$ and those in $B$ to 0 as the input to the causal model. All the other EVs are initialized with zero, except those associated with the subgoals in the intervention set IS. Then we take the output and use it as the next input. Each time we feed the causal model with $\mathbf{X}$, we ensure that the $X_i$s associated with the subgoals in $A$ are set to $\alpha$ and those in $B$ to 0. We repeat this process $\Delta$ times and then record the outcome for the final subgoal ($X_n$). We do this procedure multiple times to estimate the average of $X_n$.

## H.4. Sensitivity Analysis

In Figure 12 we show the performance of HRC algorithm under missing ratio of 20% of EVs. As you can see, the performance of $\text{HRC}_\text{h}$ (SSD) is slightly affected by the missing EVs but it still shows relatively robust and stable performance. $\text{HRC}_\text{b}$ (SSD) is more affected by missing EVs. This result indicates that our targeted strategy was able to adapt to missing information very well.

## H.5. Hyperparameters

We utilized a Linux server with Intel Xeon CPU E5-2680 v3 (24 cores) operating at 2.50GHz with 377 GB DDR4 of memory and Nvidia Titan X Pascal GPU. The computation was distributed over 48 threads to ensure a relatively efficient run time. In the following table, we provide the fine-tuned parameters for each algorithm. Batch sizes are considered the same for all algorithms. For hyperparameter tuning, we performed a grid search, systematically exploring a predefined range of values for each parameter. To ensure robustness, we repeated the process multiple times with different random seeds. Table 3 shows the key hyperparameters used in our setup. Additional hyperparameter values can be found in the supplementary materials.

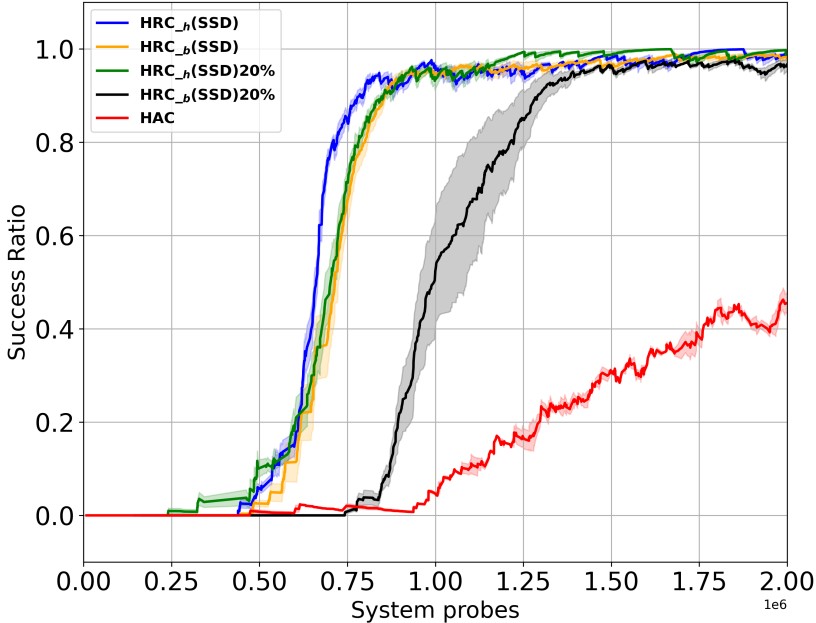

*Figure 12.* Sensitivity analysis under missing ratio of 20% of EVs.

*Table 3.* More hyperparameters of different methods used in the experiment.

| METHOD | SUCCESS RATIO THRESHOLD | INTERVENTION SAMPLING BATCH SIZE | HRL LR | TRAINING BATCH SIZE | LASSO L1-RATIO /OTHERS |
|---|---|---|---|---|---|
| HRC_H (SSD) | 0.5 | 32 | 0.0001 | 128 | 0.0001 |
| HRC_B (SSD) | 0.5 | 32 | 0.0001 | 128 | 0.0001 |
| HAC | - | - | 0.0001 | 128 | - |
| CDHRL | 0.5 | 32 | 0.0001 | 128 | func-lr=5e-3 struct-lr=5e-2 |
| OHRL/ HER | - | - | 0.0001 | 128 | - |
| PPO | - | - | 3E-4 | 128 | ENTROPY COEF=0.005 |

## H.6. Computational Complexity

Regarding computational complexity, our experiments indicate that although our method introduces some additional computation compared to some HRL baselines, this is offset by a significant reduction in the number of system probes and overall training cost. Below, we provide a runtime comparison between various methods to reach a success rate of 0.5 in Figure 6(a):

Table 4. Runtime comparison (minutes) required to reach a success rate of 0.5.

| Method | Average Time (min) |
|---|---|
| HER | 249.1 |
| HAC | 185.5 |
| OHRL | 133.8 |
| CDHRL | 352.3 |
| HRC$_h$ (ours) | 33.7 |
| HRC$_b$ (ours) | 38.1 |

## H.7. More Experimental Comparisons

For more empirical comparison, we conducted experiments on the CraftWorld environment (non-binary) provided by (Wang et al., 2024) (SkiLD). In Figure 9 of their paper, they report achieving a maximum success rate of approximately 0.6 after training the downstream task (note that they need pretraining in their method, and environment steps for pretraining are not plotted in this Figure). However, when we attempted to reproduce their results using the provided code, we were unable to match their reported performance. Additionally, the code ran extremely slowly on our GPU server, requiring about 10 hours to complete just 100k environment steps. Due to time constraints, we contacted the authors to clarify the number of pretraining steps. We received two responses—10 million and 20 million steps—but without a definitive confirmation. Based on Figure 9 of their paper, it appears that at least 12 million environment steps are needed to reach a success rate of 0.5. In contrast, the performance of our method on this environment is shown in Figure 13. Each unit on the x-axis represents 10 million environment steps. As illustrated, our method surpasses a 0.5 success rate with just 5 million environment steps, whereas SkiLD requires at least 12 million environment steps to reach the same level. Furthermore, our approach achieves a maximum success rate of approximately 0.8 , compared to about 0.6 for SkiLD.

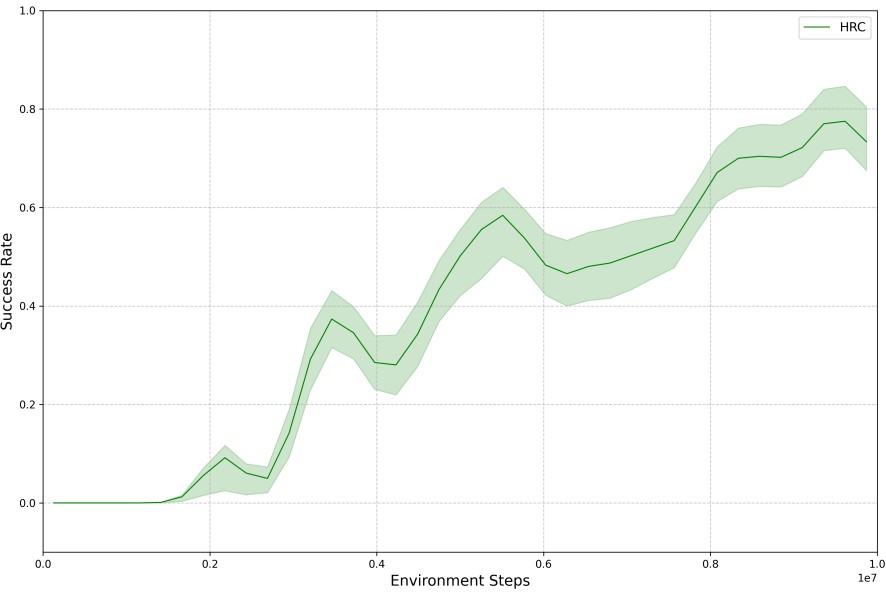

Figure 13. The performance of HRC on CraftWorld environment

