# OpenReview forum: "Hierarchical Reinforcement Learning with Targeted Causal Interventions"
_ICML.cc/2025/Conference — ICML 2025 poster_

### Official Review · Reviewer_5Zcn · 2025-03-06

**Overall Recommendation:** 3

**Summary:**

This paper considers Hierarchical Reinforcement Learning (HRL) by leveraging causal discovery to improve training efficiency in long-horizon tasks with sparse rewards. In particular, the subgoal structure is modeled as a causal graph and an algorithm to learn this hierarchy is introduced. Instead of random subgoal interventions during exploration, the proposed method prioritizes interventions based on their causal importance in achieving the final goal. This targeted intervention strategy significantly reduces training costs. The paper provides theoretical analysis, showing improvements for tree-structured and Erdős-Rényi random graphs. Experiments show that the proposed framework outperforms existing HRL methods in terms of training efficiency.

**Claims And Evidence:**

There is reasonable evidence for the main claims, both in terms of theory and experiments.

**Essential References Not Discussed:**

not that I am aware of.

**Experimental Designs Or Analyses:**

The consider grid world setup with subgoals is quite standard test environment for hierarchical RL. It could be better explained how the hyper-parameter tuning is done, especially for the baseline algorithms. More description is need in the plots, e.g., there seem to be some confidence bounds that are not explained.

**Methods And Evaluation Criteria:**

The evaluation seems reasonable, but still limited to only one test environment. The consider grid world setup with subgoals is quite standard test environment for hierarchical RL.

**Other Comments Or Suggestions:**

none

**Other Strengths And Weaknesses:**

Strengths: The paper reads very well, and based on the results, the suggested approach significantly improves reasonably selected base line algorithms. The approach intuitively makes sense.

Weaknesses: The problem formulation is quite limiting, it seems to cover mostly these type of grid-world examples with subgroups. I cannot see this used in any type of realistic setting, or maybe I am wrong, but it would then be interesting if some more meaningful application than simple grid-world toy environments. Also, compared to the baseline algorithms, the considered algorithms are clearly more simple efficient, but there seem to be high computational burden in the sub-goal structure discovery, it would be useful to contrast this what is done in the baseline algorithms, and maybe compare, not only in number of system probes, but also considered other dimensions.

**Questions For Authors:**

none

**Relation To Broader Scientific Literature:**

Mostly seems reasonable. It seems that prior work on reward machines could be relevant, they also formalize subgoal structures and use them to guide reinforcement learning by providing a structured decomposition of tasks. Reward machines leverage automata-based representations to define subgoals and shape rewards, which could complement the proposed causal discovery approach by offering an alternative way to encode and utilize hierarchical structures in HRL.

Icarte, Rodrigo Toro, et al. "Reward machines: Exploiting reward function structure in reinforcement learning." Journal of Artificial Intelligence Research 73 (2022): 173-208.

**Theoretical Claims:**

The theoretical claims seem reasonable. Did not manage to read the proofs in the appendix in much detail, only skim through some main steps, and I did not see any issues.

---

> ### Author Rebuttal · Authors · 2025-03-31
>
> We thank the reviewer for their valuable feedback and address the concerns under the following headings.
> ___
> ## 1- Reviewer: The evaluation seems reasonable, but still...
> As the reviewer mentioned, grid-world environments are widely used in HRL research, as they capture key challenges such as subgoal discovery and hierarchical planning. In our work, we primarily focus on the 2D Minecraft environment as other environments (such as Mini-Behavior environments) are relatively simple and do not adequately showcase the strengths of our method.
> ___
> ## 2- Reviewer: It could be better explained how the hyper-parameter...
> Due to the space limitations in the main text, we have provided a detailed description of the experiments and hyper‑parameter tuning in Appendix G.
> ___
> ## 3- Reviewer: It seems that prior work on reward machines...
> We agree with the reviewer that the reward machine can be viewed as an alternative way to represent hierarchical structures in HRL and can be effectively used for training policies. As the reviewer noted, our work takes a causal approach, representing subgoal structures within the framework of structural causal models. In the revised version, we will include a discussion on the line of research about reward machines in the related work.
>
> ___
> ## 4- Applicability of the framework to realistic settings
> We acknowledge that our framework currently assumes an explicit definition of EVs, which may seem restrictive in some deep RL applications. However, in practice, many environments provide state vector–based observations that are inherently disentangled—meaning that each state dimension corresponds to an independent feature. This makes it significantly easier to extract EVs. Common benchmarks such as Atari, MuJoCo, and 2D‑Minecraft, all naturally possess this property. Moreover, many hierarchical RL methods—such as LESSON, DSC, and SkilD (mentioned by the reviewer)— already rely on such structured representations. In scenarios where EVs are not directly provided, one can leverage disentangled representation learning, which aims to reconstruct a latent, disentangled state representation from observations. Methods such as CausalVAE, DEAR, and  the reference [9] suggested by the reviewer q4Ho specifically focus on learning disentangled representations from raw visual input. Therefore, they enable our framework to be applied in such settings too. Note that, in the submitted version, we conducted a sensitivity analysis to evaluate the impact of missing EVs on performance. As shown in Figure 12 (Appendix G.4), HRCh (SSD) remains fairly robust even with up to 20\% of EVs missing.
> We would like to emphasize that one of the main messages of the current work is that learning the causal structure among subgoals enables more efficient training of hierarchical policies. By leveraging this structure to guide exploration, our approach can be adapted to a wide range of applications where hierarchical decision-making is required.
> ___
> ## 5- Reviewer: Also, compared to the baseline algorithms, the considered algorithms are clearly more simple efficient, but there seem to be high computational burden...
> Regarding computational complexity, our experiments indicate that although our method introduces some additional computation compared to some HRL baselines, this is offset by a significant reduction in the number of system probes and overall training cost. Below, we provide a runtime comparison between various methods to reach a success rate of 0.5 in Figure 6(a) of the paper:
> | **Metric**       | **Average Time (mins)** |
> |-------------------|-------------------------|
> | HER              | 249.1                   |
> | HAC              | 185.5                   |
> | OHRL             | 133.8                   |
> | CDHRL            | 352.3                   |
> | HRC_h (ours)     | 33.7                    |
> | HRC_b (ours)     | 38.1                    |
>
> **Table**: Runtime Comparison of HRL Algorithms

---

### Official Review · Reviewer_q4Ho · 2025-03-14

**Overall Recommendation:** 3

**Summary:**

This paper tackles the long-horizon RL tasks with hierarchical abstractions. Specifically, the authors propose Hierarcahical RL via Causality (HRC) which enables the agents to prioritize some causally impactful subgoals over the others. Among the HRC framework, the authors also develop a new subgoal-based causal discovery approach and derive the theoretical guarantees for it. Compared to the existing hierarchical causal RL baselines, HRC empirically outperforms them in both synthetic data and MineCraft tasks.

**Claims And Evidence:**

The major claim about the theoretical property (training cost bound in interventional causal discovery) and empirical performance of HRC are generally supported by Theorem 7.4, as well as Figure 5, 6 and Table 2, respectively.

**Essential References Not Discussed:**

In the Neuro-Symbolic Reinforcement learning, there are some related works that also use similar formulations of target tasks [1, 2].

There are also some works on causal discovery from interventional data, and also derive theoretical bounds over them [3, 4].

There are some some causal RL works [5, 6, 7, 8] that use either causal discovery or explicit hierarchical structure in the policy, and other causal RL works that use controllability-related state abstraction [9, 10], which is relevant to the task domain that the authors are tackling with.

> [1] Jiang, Zhengyao, and Shan Luo. "Neural logic reinforcement learning." ICML 2019
>
> [2] Kimura, Daiki, et al. "Neuro-symbolic reinforcement learning with first-order logic." ACL 2021
>
> [3] Yang, Karren, Abigail Katcoff, and Caroline Uhler. "Characterizing and learning equivalence classes of causal dags under interventions." ICML 2018.
>
> [4] Brouillard, Philippe, et al. "Differentiable causal discovery from interventional data." NeurIPS 2020
>
> [5] Scherrer, Nino, et al. "Learning neural causal models with active interventions." NeurIPS workshop on Causal Inference & Machine Learning, 2021
>
> [6] Wang, Zizhao, et al. "SkiLD: Unsupervised Skill Discovery Guided by Factor Interactions." NeurIPS 2024
>
> [7] Lin, Haohong, et al. "BECAUSE: Bilinear Causal Representation for Generalizable Offline Model-based Reinforcement Learning." NeurIPS 2024
>
> [8] Hu, Jiaheng, et al. "Disentangled unsupervised skill discovery for efficient hierarchical reinforcement learning." NeurIPS 2024
>
> [9] Zhang, Amy, et al. "Learning invariant representations for reinforcement learning without reconstruction." ICLR 2021
>
> [10] Wang, Tongzhou, et al. "Denoised mdps: Learning world models better than the world itself." ICML 2022

**Experimental Designs Or Analyses:**

The experiments compare different variants of the proposed algorithm (using various ranking rules and causal discovery methods) against several state-of-the-art HRL baselines using metrics like training cost (system probes), success ratio, and structural Hamming distance for recovered causal graphs under the synthetic experiments and 2D-MineCraft evaluation.

1. The evaluation protocol is comprehensive in evaluating both the success ratio and causal discovery quality.
2. The ablation study is not explicitly mentioned. Since the pipeline in Algorithm 1 is pretty long, it might be helpful to conduct throrough ablation studies and discuss how important each module is and how likely some assumptions would hold in practice, such as the assumption 7.1.
3. I'm not sure whether OR, AND could cover all the possible subgoal relationship in all the long-horizon decision making problems. The authors may include this in the limitation discussions.
4. It might be helpful if the authors could compare the HRC with other causal RL approaches listed in some of the essential references [5, 6, 7].

**Methods And Evaluation Criteria:**

**Method**: The paper’s method involves a hierarchical reinforcement learning framework that leverages causal discovery to learn the subgoal structure underlying long-horizon, sparse-reward tasks. Specifically, it introduces a new causal discovery algorithm tailored for HRL, and then uses targeted causal interventions—prioritizing subgoals via ranking rules (such as causal effect and shortest path ranking)—to guide exploration and improve training efficiency.

**Criteria**: The evaluation criteria include cost complexities for theoretical cost analysis. For the empirical performance on benchmark environment, the authors use metrics like success ratio for the task reward and structural Hamming distance (SHD), missing edges, and extra edges for causal graph accuracy.

The proposed methods and evaluation criteria make sense to me. They directly address the challenges of sparse rewards and inefficient exploration in HRL by exploiting causal relationships to guide subgoal selection, and the combination of theoretical and practical evaluation provides a well-rounded assessment of the framework’s efficacy in complex, long-horizon tasks.

**Other Comments Or Suggestions:**

The paper structure can be merged into a few major sections and some subsections. Now it has 10 sections, which may be too much for the audience to keep up with the authors stories.

**Other Strengths And Weaknesses:**

**Strengths**

1. The paper is generally well-written and easy to follow, the paper is well-structured and well-motivated.
2. The baseline comparison is comprehensive, and it considers both the task success ratio and causal discovery quality.
3. The theoretical derivation is generally sound and solid, which makes the results potential more generalizable in broader subgoal-based causal discovery applications. Moreover, the authors verify the theoretical results with empirical causal discovery performance in Figure 5.
4. The empirical performance demonstrate a superior performance than SDI in Structured Hamming Distance, and a better assymptotic performance and convergence speed compared to other HRL baselines.

**Weaknesses**

1. Additional causal RL baselines such as the ones in [5, 6, 7, 8] in additional references, would be helpful. Comparison to other HRL approaches with implicit causal abstraction (e.g. the causal abstraciton based on controllability and/or reward relevance [9, 10]  in additional references) could also be interesting.
2. If the cost complexity bound can be comparable with other literatures, please provide for a better reference.
3. The authors do not include any limitation discussions or future works in the main text.
4. The assumption of AND, OR subgoal abstraction may not be expressive enough for all the scalable applications in long-horizon real-world decision-making tasks, such as robot manipulation or autonomous driving tasks which require complex reasoning.

**Questions For Authors:**

See the above sections for more details.

**Relation To Broader Scientific Literature:**

Beyond the hierarchical RL and causality community, the Definition 4.1 and Assumption 7.1 remind me of the controllability of the subgoal seems to be quite similar to the reachability analysis [1] in stochastic control domain. It might be interesting to reveal the inner connection between HRC and the stochastic control methods.



> [1] Amin, Saurabh, et al. "Reachability analysis for controlled discrete time stochastic hybrid systems." Hybrid Systems: Computation and Control: 9th International Workshop, HSCC 2006, Santa Barbara, CA, USA, March 29-31, 2006. Proceedings 9. Springer Berlin Heidelberg, 2006.

**Theoretical Claims:**

The paper's main theoretical claims are that by exploiting the causal structure among subgoals using a novel, HRL-tailored causal discovery algorithm, the proposed HRC framework can significantly lower the training cost compared to random exploration. In particular,

1. Undet the assumption 4.2, the authors provide formal guarantees under certain assumptions, the subgoal structure is identifiable up to discoverable parents in Proposition 8.3 and Theorem 8.4.

2. Under the assumption 7.1, 7.2, 7.3, the targeted causal interventions based on ranking rulesyield lower cost complexities—in tree and semi‐Erdős–Rényi graph models—compared to naive exploration strategies in Theorem 7.4.

I reviewed the proofs provided for Theorem 7.4 (the cost complexity analysis) and Theorem 8.4 (the identifiability of the causal subgoal structure). The derivations appear logically sound and make appropriate use of standard techniques. While I did not perform a line‐by‐line verification, I did not find any obvious mathematical errors. However, in general deep RL applications, the assumptions may limit their generality in more practical tasks, e.g. the subgoal is not explicitly defined and is hard to abstract from pure observation, or the direct intervention is not practical in the simulator.

---

> ### Author Rebuttal · Authors · 2025-03-31
>
> We thank the reviewer for their valuable feedback and address the concerns under the following headings.
> ___
> ## 1- Reviewer: The ablation study...
> We already conducted several ablation studies (Figure 6) to evaluate our approach. To assess the impact of our targeted strategy, we compared three HRC algorithm variants: HRCh (SSD), HRCb (SSD), and HRC (optimal). As shown in Figure 6(a), HRCh (SSD) outperforms HRCb (SSD), highlighting the ranking rule's effectiveness.
> To examine the causal discovery component, we also compared two HRC versions: one using our SSD method and another using a standard method (SDI). Results show SSD significantly outperforms SDI, demonstrating the benefits of our tailored causal discovery.
> Lastly, a sensitivity analysis (Figure 12, Appendix G.4) shows HRCh (SSD) remains robust even with up to 20% of EVs missing.
>
> ___
> ## 2- Reviewer: However, in general deep RL applications...
>
> ***Regarding the comment "Generality to more practical applications" please see number "4" of our response to Reviewer 5Zcn***
>
> Regarding interventions, as noted in Remark 4.3, in environments where EVs correspond to specific resources or skills, we assume that once a resource or skill is acquired at some time step, it remains accessible in subsequent time steps (Assumption 4.2). Under this assumption, a subgoal being controllable by a policy $\pi$ (as defined in Definition 4.1) is equivalent to performing an intervention on the corresponding EV with policy $\pi$. This interpretation is applicable in both simulated and real-world environments.
>
> ___
> ## 3- Reviewer: I'm not sure whether OR, AND...
> As noted in lines 161–164, our proposed solutions can be readily extended to settings with non-binary domains for EVs, albeit at the cost of heavier notation. In fact, our experiments already evaluated the proposed methods in non-binary settings. The assumption of binary variables for resource EVs with AND/OR subgoals is made solely to facilitate the cost analysis in Section 7 and to provide theoretical guarantees for our causal discovery method in Section 8. We will clarify this further in the revised version.
>
> ___
> ## 4- Other references
>
> Thank you for pointing out these references. In Related Work section, we will add references [1, 2] regarding neuro-symbolic RL; [3–5] for the use of interventional data in learning causal structures; [6, 8] for subgoal/skill discovery; and [7, 9, 10] for causal representation learning/causal abstraction in RL. We will also discuss about connection with stochastic control methods.  References [3–5] pertain to learning causal structures among random variables in structural causal models while our causal discovery method is specifically tailored to the HRL setting, working with sequenced data. In our experiments (Figure 6(b)), we have already compared our method with SDI, a representative method from this class that has been used in prior HRL work as a subroutine to learn the causal structure.
>
> Regarding [9], this line of research on causal representation learning is more geared towards recovering EVs, which is not the focus of our paper. Similar to prior work, we assume that either the EVs are already available or that the environment provides state vector–based observations. As for prior work on skill discovery (e.g., [6,8]), please refer to our response "Comparison with skill discovery" to the reviewer T49j for more details.
>
>
> Regarding [1–2], ILP-based methods represent policies as logical rules to enhance interpretability.  Our work considers a multi-level policy, with a focus on recovering the hierarchy among subgoals (with our explicit hierarchical structure definition) for training the multi-level policy more efficiently. Herein, we do not impose any structural limitations on our policy. For our theoretical analysis, we assumed that the causal mechanism of each subgoal follows an AND/OR structure. This assumption pertains to the environment rather than for representing the policy using logical rules. We will clarify this in the related work.
> ___
> ## 5- Reviewer: If the cost complexity bound...
> To the best of our knowledge, this is the first work to rigorously model the HRL problem in SCM framework with a formal associated cost, and provide theoretical guarantees on the performance of the proposed methods. Analytical comparison with other related HRL approaches is challenging, as most existing methods are experimental in nature and lack a rigorous formalism. Nevertheless, in our analysis, we compared our approach with a baseline that uses random exploration for subgoal discovery, and our method outperforms it in both considered subgoal structures (tree and Erdős–Rényi graphs).
> Regarding  limitations, due to space constraints in the main text, we were unable to include a detailed discussion. However, in the revised version, we will add a section to discuss the limitations and the future work.

---

### Official Review · Reviewer_T49j · 2025-03-19

**Overall Recommendation:** 4

**Summary:**

The paper presents a novel approach to Hierarchical Reinforcement Learning by leveraging causal discovery to identify hierarchical structures among subgoals. The key contribution is a causal discovery algorithm that learns the subgoal structure, which is then used to guide interventions during exploration. This targeted intervention strategy improves training efficiency compared to random exploration. The paper provides a formal analysis of the method and demonstrates empirical improvements on synthetic datasets and a gridworld environment (2D-Minecraft). Results show that the proposed method outperforms existing HRL approaches.

## update after rebuttal
I am happy with the rebuttal content. I'm raising my score to 4.

**Claims And Evidence:**

Most of the claims are well-supported. The paper provides a formal analysis for its proposed method, which strengthens its theoretical grounding.

More ablation studies, particularly on the impact of different causal discovery techniques, would strengthen the claims regarding the proposed algorithm's effectiveness. In addition, the writing of this paper is not super clear, with complicated and sometimes unexplained notations scattered over a total of 9 different sections. It would be beneficial to organize the method sections with a clearer intuition of what each subsection is doing.

**Essential References Not Discussed:**

The overall idea of this paper seems to resemble previous works [1][2], in terms of using causal dependency as a way to guide skill learning. Can the author discuss the connection/difference between this work and the aforementioned ones, and possibly compare their performance?

[1] Chuck, Caleb, et al. "Granger-causal hierarchical skill discovery." TMLR 2023
[2] Wang, Zizhao, et al. "SkiLD: Unsupervised Skill Discovery Guided by Factor Interactions." NeurIPS 2024.

**Experimental Designs Or Analyses:**

Please see the "Evaluation Criteria" section above.

**Methods And Evaluation Criteria:**

The authors conduct experiments on both synthetic datasets and a real-world inspired HRL environment (2D-Minecraft), demonstrating the effectiveness of the approach.

While the method performs well on the relatively toy 2D-Minecraft domain, it is unclear how well it scales to more complex real-world tasks with high-dimensional state spaces. In particular, I’m worried about two assumptions. First, the proposed method relies on being able to detect whether a environment variable is controllable. While this may be relatievly easy when the variable is discrete (especially when binary), but seems to be more problematic when the variable is continuous. Second, the possible “preconditions” for achieving a subgoal can grow exponentially w.r.t the total number of environment variables. It’s unclear whether this method will still work with a large number of environment variables.

**Other Comments Or Suggestions:**

N/A

**Other Strengths And Weaknesses:**

N/A

**Questions For Authors:**

See the questions above.

**Relation To Broader Scientific Literature:**

This paper fits nicely into the causality-inspired skill discovery literature, and introduces a principled way to incorporate causal discovery into HRL. In particular, the causally-guided ranking system seems quite novel. However, the addition of some missing references (as detailed in the next section) would help assess the novelty of this work.

**Theoretical Claims:**

The proofs look correct to me.

---

> ### Author Rebuttal · Authors · 2025-03-31
>
> We thank the reviewer for their valuable feedback and address the concerns under the following headings.
> ___
> ## 1- Reviewer:  In particular, I’m worried about two assumptions...
> Regarding controllability, our current framework assumes discretized subgoals, which is a reasonable assumption in domains where acquiring specific resources is required. For environments with continuous variables, one can apply disentangled representation learning methods to high-dimensional continuous observations in order to extract categorical latent variables—such as using categorical VAEs. We view this as a natural extension of our current work. The assumption of binary variables for resource EVs is made solely to facilitate the cost analysis in Section 7 and to provide theoretical guarantees for our causal discovery method in Section 8. To the best of our knowledge, this is the first work to rigorously model the HRL problem in SCM framework with a formal associated cost, and provide theoretical guarantees on the performance of the proposed methods. We will clarify this further in the revised version.
>
> As for the exponential growth of preconditions, we agree that in the worst case, the number of potential combinations can be large. However, in practice, many environment variables (EVs) exhibit sparse dependencies, i.e., the parent set is small. As shown in Theorem 7.4, our method leverages the sparsity in the subgoal structure to guide exploration more efficiently toward the target goal. For example, in a tree-structures, our method achieves a logarithmic cost with respect to the number of EVs, whereas a random strategy incurs a quadratic cost. This efficiency enables our approach to scale effectively to environments with a large number of EVs in such structures.
> ___
> ## 2- Clarity and organization of the paper
> We will include a table in the appendix listing all notations along with references to their definitions in the text. Additionally, we will add a brief paragraph at the end of Section 1 to clarify the overall structure of the paper, particularly the methodology section. It would be helpful if the reviewer could refer any notation that they believe was not explicitly or adequately defined in the text.
> ___
> ## 3- Comparison with skill discovery
>
> The goal of skill discovery is to learn a diverse set of skills, which are later used to train a higher-level policy for a downstream task. In that context, a skill is conceptually similar to a subgoal in our work. However, there are key methodological differences in subgoal vs. skill discovery: 1- In skill discovery, the process of learning the skill set is often decoupled from the downstream task. In contrast, our work—framed within the context of HRL—conditions the learning process on a target goal. By leveraging the learned subgoal structure, our approach guides exploration toward only the relevant subgoals that contribute to achieving the target goal. In contrast, skill discovery methods often aim to learn as many diverse skills as possible, regardless of their relevance to the target goal in the downstream task. 2- In our framework, the lower-level policy is itself hierarchical and is trained according to defined hierarchical structure (where we defined it formally based on the discovered subgoal structure). This results in a significant reduction in sample complexity compared to standard policy training, which typically does not utilize such a structure. Beyond these methodological distinctions in subgoal discovery and training policies, to the best of our knowledge, our work is the first to rigorously study HRL within a causal framework. We formally defined the cost formulation, proposed subgoal discovery strategies (with our key measure ECE) with performance guarantees (Theorem 7.4), and provided theoretical bounds on the extent to which the subgoal structure can be learned in an HRL setting (Prop. 8.3). Furthermore, we introduce a causal discovery algorithm tailored to this setting, with provable guarantees on its correctness (Theorem 8.4).
> ### Empirical comparison
> For empirical comparison, we consider [2] as it showed superior performance to [1]. [2] conducted a similar experiment on a simplified Minecraft version with only 8 EVs and a 500-step horizon, achieving a 0.5 success rate after 2 million system probes. Our environment is much more complex with 21 EVs and a 100-step horizon, yet our method achieves a success rate approaching 1 (Figure 6). When testing [2]'s code on our Minecraft version, it consumed over 450 GB of memory before crashing. Therefore, we couldn't complete further testing of their approach, but we're currently evaluating our method on their Minecraft version for direct comparison.

---

> > ### Comment · Reviewer_T49j · 2025-04-03
> >
> > I thank the author for the rebuttal, which has addressed many of my concerns. I encourage the authors to add the aforementioned clarifications, the promised comparisons as well as the missing related works in the next version of this paper.

---

> > > ### Author Response · Authors · 2025-04-07
> > >
> > > We thank the reviewer for their supportive comment. We conducted experiments on the CraftWorld environment (non-binary) provided by [2] (SkiLD). In Figure 9 of their paper, they report achieving a maximum success rate of approximately 0.6 after training the downstream task (note that they need pretraining in their method, and environment steps for pretraining are not plotted in this Figure). However, when we attempted to reproduce their results using the provided code, we were unable to match their reported performance. Additionally, the code ran extremely slowly on our GPU server, requiring about 10 hours to complete just 100k environment steps. Due to time constraints, we contacted the authors to clarify the number of pretraining steps. We received two responses—10 million and 20 million steps—but without a definitive confirmation. Based on Figure 9 of their paper, it appears that at least 12 million environment steps are needed to reach a success rate of 0.5.
> > >
> > > In contrast, the **performance of our method on this environment is shown in the following plot: https://ibb.co/dsYvwWJR (alternative link: https://postimg.cc/tZgy9sBM)**. Each unit on the x-axis represents 10 million environment steps. As illustrated, our method surpasses a 0.5 success rate with just 5 million environment steps, whereas SkiLD requires at least 12 million environment steps to reach the same level. Furthermore, **our approach achieves a maximum success rate of approximately 0.8 , compared to about 0.6 for SkiLD**.
> > >
> > >
> > > We will ensure that the clarifications, comparisons, and related works mentioned in the rebuttal are thoroughly incorporated into the next version of the paper to improve its quality. If you find our work convincing and aligned with your expectations, we would greatly appreciate your support in recommending it for acceptance.

---

### Decision · Program_Chairs · 2025-05-01

**Decision:**

Accept (poster)

**Comment:**

This paper explores a novel approach to Hierarchical Reinforcement Learning (HRL) by incorporating causal discovery to enhance learning efficiency in environments with sparse rewards and long-horizon tasks. The key idea is to represent the subgoal hierarchy as a causal graph and introduce an algorithm that learns this structure. Rather than relying on random subgoal exploration, the method strategically selects interventions based on their causal relevance to the final objective, reducing training time. Empirical evaluations further confirm that the proposed framework consistently outperforms existing HRL techniques regarding training efficiency. Reviewers agreed on the interesting and valuable contributions of the work.